# TOWARDS PRINCIPLED EVALUATIONS OF SPARSE AUTOENCODERS FOR INTERPRETABILITY AND CONTROL

**Aleksandar Makelov**[*]
aleksandar.makelov@gmail.com

**Georg Lange**[*]
mail@georglange.com

**Neel Nanda**
neelnanda27@gmail.com

## ABSTRACT

Disentangling model activations into human-interpretable features is a central problem in interpretability. Sparse autoencoders (SAEs) have recently attracted much attention as a scalable unsupervised approach to this problem. However, our limited understanding of ground-truth features in realistic scenarios makes it difficult to measure the success of SAEs. To address this challenge, we propose to evaluate SAEs on specific tasks by comparing them to supervised feature dictionaries computed with knowledge of the concepts relevant to the task.

Specifically, we suggest that it is possible to (1) compute supervised sparse feature dictionaries that disentangle model computations for a specific task; (2) use them to evaluate and contextualize the degree of disentanglement and control offered by SAE latents on this task. Importantly, we can do this in a way that is agnostic to whether the SAEs have learned the exact ground-truth features or a different but similarly useful representation.

As a case study, we apply this framework to the indirect object identification (IOI) task using GPT-2 Small, with SAEs trained on either the IOI or OpenWebText datasets. We find that SAEs capture interpretable features for the IOI task, and that more recent SAE variants such as Gated SAEs and Top-K SAEs are competitive with supervised features in terms of disentanglement and control over the model. We also exhibit, through this setup and toy models, some qualitative phenomena in SAE training illustrating feature splitting and the role of feature magnitudes in solutions preferred by SAEs.

## 1 INTRODUCTION

While large language models (LLMs) have demonstrated impressive (Vaswani et al., 2017; Devlin et al., 2019; Radford et al., 2019; Brown et al., 2020; OpenAI, 2023) results, the mechanisms behind their successes and failures largely remain a mystery (Olah, 2023). A prominent bottom-up approach to this problem is taken by mechanistic interpretability (MI), which aims to disentangle model activations into units with faithful, human-interpretable roles in an LLM's computation (Olah, 2022). Recently, the MI community has focused on sparse autoencoders (SAEs) as a promising approach to this problem (Olshausen & Field, 1997; Faruqui et al., 2015; Goh, 2016; Arora et al., 2018; Yun et al., 2021; Cunningham et al., 2023; Bricken et al., 2023). If successful at scale, this research could provide significant scientific and practical value, enabling enhanced model robustness, controllability, interpretability, and debugging (Gandelsman et al., 2023; Nanda et al., 2023; Marks et al., 2024; Conmy & Nanda, 2024).

However, realistic ground-truth evaluations of the SAE paradigm have been lacking, which hinders progress in this area. While recent works have proposed various metrics to evaluate SAE quality (detailed in Section 2), they all rely on indirect metrics only conjecturally correlated with recovering the 'true' features used by LLMs.

To address this challenge, we propose the first principled method for evaluating SAEs on realistic linguistic tasks. Our method directly compares SAEs to *supervised* feature dictionaries computed using knowledge of task-relevant concepts. By comparing SAEs to objects of the same 'type signature', we provide a fair comparison where the supervised dictionaries serve as a canonical skyline

---

[*]Joint contribution. Correspondence to aleksandar.makelov@gmail.com

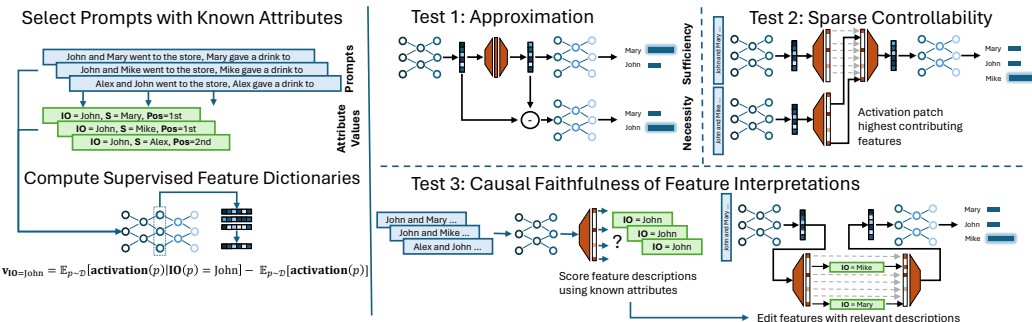

Figure 1: Overview of our evaluation pipeline. We begin by selecting a specific model capability (IOI task, Section 3) and then disentangling model activations into capability-relevant features using supervision (Section 4.2). Then, we evaluate a given feature dictionary w.r.t. this capability, using the supervised features as a benchmark (Section 4.3). We test the extent to which (1) the feature dictionary's reconstructions of the activations are necessary and sufficient for the capability (Section 5.3), (2) the features can be used to edit capability-relevant information in internal model representations agnostic of feature interpretations (Section 5.4), and (3) the features can be interpreted w.r.t. the capability in a manner consistent with their causal role (Section 5.5).

for SAE performance. The IOI task (Wang et al., 2022) – a non-trivial natural language task that has been extensively studied in the mechanistic interpretability literature – serves as a natural case study on which we base this paper. However, we stress that our framework is applicable to other tasks, as we illustrate in Appendix 7.1. Our contributions are outlined as follows:

- We propose a principled method to compute *supervised* sparse feature dictionaries w.r.t. a specific task an LLM can do, using supervision via task-relevant prompt attributes.
- We apply this method to the IOI task, demonstrating that these dictionaries exhibit three desirable properties: (1) dictionaries' activation reconstructions are both sufficient and necessary for the task; (2) attributes relevant to the task are disentangled in a way that allows precise control over model behavior; and (3) the features are interpretable in a way consistent with their causal role in the model's computation.
- We use these feature dictionaries to design and contextualize evaluations of any feature dictionaries along the same three axes, in a way agnostic to whether they use the same latent directions.
- We apply this methodology to SAEs trained on either the IOI dataset (*task-specific SAEs*) or the LLM's pre-training dataset (*full-distribution SAEs*), finding that both types contain interpretable latents for the task, and that some task-specific SAE variants allow us to edit attributes about as effectively as supervised dictionaries
- We also briefly remark on two qualitative phenomena we observed in SAE training and reproduced in toy models: a tendency for SAEs to learn higher-magnitude features, and a tendency to 'over-split' an intuitively single concept into many features. Due to space constraints, we defer a full discussion to Appendix 7.2.

Our results suggest that more detailed and controlled SAE evaluations are possible and informative, and that SAEs may hold promise for disentangling model computations in realistic scenarios. A limitation of our approach is that it requires potentially substantial per-task effort to independently identify the relevant attributes, which is proportional to the complexity of the task and the number of attributes we want to consider. However, as we show in Appendix 7.1, our framework allows for the targeted evaluation of a few attributes in a few locations of the model, which can substantially reduce this effort.

## 2 RELATED WORK

**Most related to this paper.** Several recent works propose controlled SAE evaluations related to our approach. Karvonen et al. used a board game model as a testbed, comparing SAE latents to

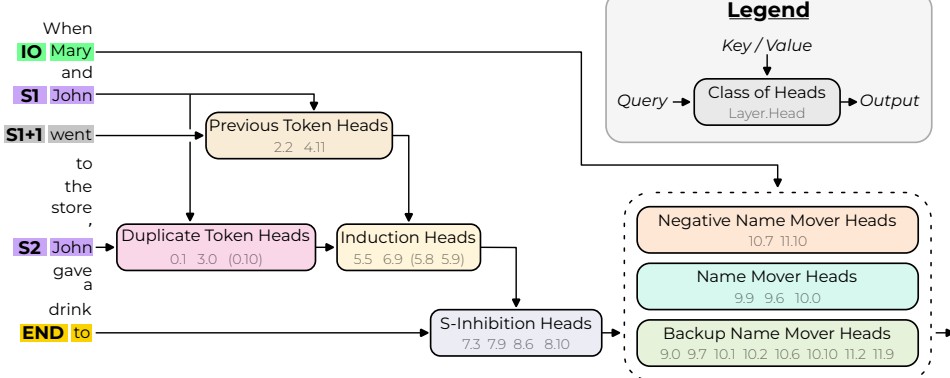

Figure 2: A reproduction of Figure 2 from Wang et al. (2022), showing the internal structure of the IOI circuit. Original caption: *The input tokens on the left are passed into the residual stream. Attention heads move information between residual streams: the query and output arrows show which residual streams they write to, and the key/value arrows show which residual streams they read from.*

supervised features based on atomic board state components and simple strategic configurations. Huang et al. (2024); Chaudhary & Geiger (2024) evaluated SAEs for disentanglement of factual attributes in a templated dataset, while Gao et al. (2024) proposed evaluating SAE latents as binary classifiers over human-defined tasks (e.g., sentiment classification). A concern with the choice of 'ground-truth' concepts in these works is that, while meaningful from a human perspective, they may not align with internal model computations. We also differ from these works in that we focus on linguistic tasks in pre-trained LLMs.

**Other SAE evaluations.** Prior works have also proposed various indirect metrics as proxies for assessing recovery of 'ground-truth' features. These include automated interpretability methods based on highly activating examples (Bills et al., 2023; Choi et al., 2024; Juang et al., 2024), though using maximum activating examples has been criticized as potentially misleading (Bolukbasi et al., 2021). Other approaches use the trade-off between reconstruction error and sparsity (Bricken et al., 2023; Cunningham et al., 2023; Rajamanoharan et al., 2024), toy models (Elhage et al., 2022b), computational proxies (Bricken et al., 2023; Templeton et al., 2024), or geometric measures like mean maximum cosine similarity between different SAEs' features (Sharkey et al., 2023).

## 3 PRELIMINARIES

**The linear representation hypothesis and sparse autoencoders.** A central hypothesis in interpretability is the *linear representation hypothesis*. A strong variant of this hypothesis posits that model activations can be approximately decomposed into meaningful features using a *sparse feature dictionary*: given a location in the model (e.g., an attention head output), there exists a set of vectors $\{\mathbf{u}_i\}_{i=1}^m$ such that each activation $\mathbf{a}$ at this location can be approximated as a sparse linear combination of the $\mathbf{u}_j$ with non-negative coefficients. In particular, recent work suggests that $n$-dimensional activations $\mathbf{a} \in \mathbb{R}^n$ may be best described by $m \gg n$ such features in *superposition* (Elhage et al., 2022a; Gurnee et al., 2023). Recently, SAEs have been proposed as a way to disentangle these features. Following the setup of Bricken et al. (2023) here and in the rest of this work, a sparse autoencoder (SAE) is an unsupervised model which learns to reconstruct activations $\mathbf{a} \in \mathbb{R}^n$ as a weighted sum of $m$ features with non-negative weights. Specifically, the autoencoder computes a hidden representation $\mathbf{f} = \mathrm{ReLU}\left(W_{enc}\left(\mathbf{a} - \mathbf{b}_{dec}\right) + \mathbf{b}_{enc}\right)$ and a reconstruction

$$\widehat{\mathbf{a}} = W_{dec}\mathbf{f} + \mathbf{b}_{dec} = \sum_{j=1}^m \mathbf{f}_j(W_{dec})_{:,j} + \mathbf{b}_{dec} \qquad (1)$$

where $W_{enc} \in \mathbb{R}^{m \times n}$, $W_{dec} \in \mathbb{R}^{n \times m}$, $\mathbf{b}_{dec} \in \mathbb{R}^n$, $\mathbf{b}_{enc} \in \mathbb{R}^m$ are learned parameters. The rows of $W_{enc}$ are the *encoder directions*, and the columns of $W_{dec}$ are the *decoder directions*. Similarly, $\mathbf{b}_{enc}$ is the encoder bias and $\mathbf{b}_{dec}$ is the decoder bias. The decoder directions determine

the features we decompose the activations into, while the encoder directions compute the coefficients of these features for a given activation. The decoder directions are constrained to have unit norm: $\|(W_{dec})_{:,i}\|_2 = 1$. In the simplest setup (Bricken et al., 2023), the training objective over examples $\{\mathbf{a}^{(k)}\}_{k=1}^N$ is the sum of the MSE between the activations $\mathbf{a}^{(k)}$ and their reconstructions $\widehat{\mathbf{a}}^{(k)}$, and the $\ell_1$ regularization term $\lambda \sum_{k=1}^N \|\mathbf{f}^{(k)}\|_1$, where $\lambda$ is the $\ell_1$ regularization coefficient.

**The IOI task.** In Wang et al. (2022), the authors analyze how the decoder-only transformer language model GPT-2 Small (Radford et al., 2019) performs the Indirect Object Identification (IOI) task. In this task, the model is given sentences of the form 'When Mary and John went to the store, John gave a book to' (with the intended completion in this case being ' Mary'). We refer to the repeated name (John) as **S** (the subject) and the non-repeated name (Mary) as **IO** (the indirect object). For each choice of the **IO** and **S** names, there are two patterns the sentence can have: one where the **IO** name comes first (we call these 'ABB examples'), and one where it comes second (we call these 'BAB examples'). We refer to this binary attribute as the **Pos** attribute (short for position). Additional details on the data distribution, model and task performance are given in Appendix 7.6.

Wang et al. (2022) discover several classes of attention heads in GPT2-Small that collectively form the *IOI circuit* solving the IOI task (Figure 2; see Appendix 7.3 for more details on the circuit structure). Specifically, Wang et al. (2022) provide multiple lines of evidence that the circuit approximately implements the algorithm:

1. detect the (i) position in the sentence and (ii) identity of the repeated name in the sentence (i.e., the **S** name). This information is computed and moved by *duplicate token/induction* and *S-Inhibition* heads;

2. based on the two signals (i) and (ii), exclude this name from the attention of the *name mover heads*, so that they copy the remaining name (i.e., the **IO** name) to the output.

**The logit difference metric.** To discover the IOI circuit, Wang et al. (2022) used the logit difference: the difference in log-probabilities assigned by the model to the **IO** and **S** names. This metric is more sensitive than accuracy, which makes possible the detection of individual model components (or interventions thereof) with a consistent but non-pivotal role in the task. Accordingly, we also use the logit difference throughout this work to evaluate the causal effect of fine-grained model interventions.

**Measuring interpretability with the $F_1$ score.** Following prior work (Bricken et al., 2023), we assign interpretability scores for SAE latents using precision and recall, which we combine in the $F_1$ score. Specifically, given a set of examples $S$ used for evaluation, an SAE latent $f$ active on a subset $F \subset S$ of examples, and a potential interpretation in the form of a subset $A \subset S$ (e.g., the **IO** name being 'Mary'), we can evaluate the interpretation's precision and recall with respect to $f$: $\mathrm{recall}(F, A) = |A \cap F| / |A|$ and $\mathrm{precision}(F, A) = |A \cap F| / |F|$ We combine them into a single number using the $F_1$ score (see Appendix 7.4 for some discussion of this metric):

$$F_1(F, A) = \frac{2\,\mathrm{precision}(F, A)\,\mathrm{recall}(F, A)}{\mathrm{precision}(F, A) + \mathrm{recall}(F, A)}.$$

## 4 COMPUTING AND EVALUATING SUPERVISED FEATURE DICTIONARIES

This section is a self-contained presentation of our supervised feature dictionary methods and results. In the next section 5, we will extend these methods to SAEs.

### 4.1 PARAMETRIZING THE IOI TASK VIA ATTRIBUTES

Given a distribution over IOI prompts $p \sim \mathcal{D}$, we can form the sets $\mathrm{Names_{IO}}, \mathrm{Names_S}$ of names that appear as, respectively, the **IO** and **S** name in the prompts from $\mathrm{supp}(\mathcal{D})$. In a slight abuse of notation, we define functions

$$\mathbf{IO}: p \mapsto \mathrm{Names_{IO}}, \ \mathbf{S}: p \mapsto \mathrm{Names_S}, \ \mathbf{Pos}: p \mapsto \{\mathrm{ABB}, \mathrm{BAB}\}$$

that assign the values of **IO**, **S** and **Pos** to each prompt ('ABB' means the **IO** name appears first, 'BAB' means it appears second). Together, these three attributes define a parametrization of IOI prompts.

**Why use these attributes?** Not every set of attributes will result in a good approximation of the model's internal activations; in fact, we find that the choice of attributes is a crucial modeling decision. We chose the **IO**, **S** and **Pos** attributes because they are the ones found to be key to the IOI circuit in Wang et al. (2022). We experimented with other choices of attributes, but did not find them to be more successful in our tests (see Appendix 7.10 for details[1]). We emphasize that there are many other imaginable choices of attributes; see Appendix 7.12 for further discussion.

## 4.2 COMPUTING SUPERVISED FEATURE DICTIONARIES

Motivated by the linear representation hypothesis, we conjecture that model activations can be linearly decomposed as sums of features corresponding to the attributes. Specifically, given activations $\mathbf{a}(p) := \mathbf{a}_{\mathcal{C}}(p) \in \mathbb{R}^d$ of a given model component $\mathcal{C}$ (e.g., outputs of an attention head) for IOI prompts $p \sim \mathcal{D}$, there exist vectors (specific to the component $\mathcal{C}$, which we omit from the notation for brevity) $\mathbf{u}_{\mathbf{IO}=x}, \mathbf{u}_{\mathbf{S}=y}, \mathbf{u}_{\mathbf{Pos}=z} \in \mathbb{R}^d$ for each possible value $x, y, z$ of the respective attributes, such that

$$\mathbf{a}(p) \approx \mathbb{E}_{p \sim \mathcal{D}}\left[\mathbf{a}(p)\right] + \mathbf{u}_{\mathbf{IO}=\mathbf{IO}(p)} + \mathbf{u}_{\mathbf{S}=\mathbf{S}(p)} + \mathbf{u}_{\mathbf{Pos}=\mathbf{Pos}(p)} := \widehat{\mathbf{a}}(p) \tag{2}$$

where $\widehat{\mathbf{a}}(p)$ is the *reconstruction* of $\mathbf{a}(p)$. Note that Equation 2 is quite similar, but less expressive, than SAE reconstructions (Equation 1), as it effectively requires a given feature to always appear with the same coefficient in reconstructions. This is a reasonable assumption in settings like IOI, where we expect the relevant features to behave as binary, on/off switches, as opposed to having continuous degrees of activation[2]. The expectation of $\mathbf{a}(p)$ is analogous to the decoder bias in an SAE.

To compute optimal values for the attribute vectors $\mathbf{u}_{\mathbf{IO}=x}, \ldots$, we considered several approaches[3]. Our most effective method for the IOI task is straightforward: to compute e.g. $\mathbf{u}_{\mathbf{IO}=x}$, we take the conditional expectation over the event $\mathbf{IO}(p) = x$ and subtract the overall expectation. Formally,

$$\mathbf{u}_{\mathbf{IO}=x} = \mathbb{E}_{p \sim \mathcal{D}}\left[\mathbf{a}(p) \mid \mathbf{IO}(p) = x\right] - \mathbb{E}_{p \sim \mathcal{D}}\left[\mathbf{a}(p)\right].$$

We refer to this method as **mean feature dictionaries**. One can prove that, in the limit of infinite data, the mean features for an attribute not linearly detectable in the activations will converge to zero (Appendix 7.7).

## 4.3 EVALUATING SUPERVISED FEATURE DICTIONARIES

**Intervention notation and locations.** As above, we denote by $\mathbf{a}(p) := \mathbf{a}_{\mathcal{C}}(p)$ the activation of a single model component $\mathcal{C}$ (e.g., output of a single attention head) that we omit for brevity. We denote an intervention that runs the model on a prompt $p$ but replaces the activation $\mathbf{a}(p)$ with $\mathbf{v}$ as $\mathbf{a}(p) \leftarrow \mathbf{v}$ and continues the forward pass (often called *activation patching* (Vig et al., 2020)). Note that such interventions can be applied simultaneously to multiple model components, but upstream interventions may be 'overwritten' by downstream ones. In what follows, given an intervention intv, we denote the logit difference of the model after the intervention as $\mathrm{logitdiff}\,(\mathrm{intv})$.

In all evaluations, we focus on cross-sections of the IOI circuit as identified by Wang et al. (2022). These cross-sections are groups of attention heads that serve a shared computational role, such as detecting name repetitions or moving information between positions. They represent key information bottlenecks in the model's computation where variables are passed between different parts of the circuit (described in more detail in Appendix 7.3; also see Figure 2). We intervene on entire cross-sections instead of just single heads in order to observe a stronger causal effect.

**Sufficiency and necessity of reconstructions.** The first set of evaluations measures the faithfulness of activation reconstructions $\widehat{\mathbf{a}}(p)$. It is closely related to prior work (e.g., the 'cross-entropy loss recovered' metric used in many SAE works (Bricken et al., 2023; Rajamanoharan et al., 2024)). Here, we consider natural IOI-specific versions of this metric, and complement it with a measure of how much information is lost when reconstructions are removed.

---

[1]In particular, we found that another set of attributes, though more expressive in principle, learns to approximate the features we get using the **S**, **IO** and **Pos** attributes through a change-of-variables-like transformation.

[2]See also the discussion of 'features as directions' vs 'features as points' in Tamkin et al. (2023).

[3]See Appendix 7.8 for more details on other approaches we considered.

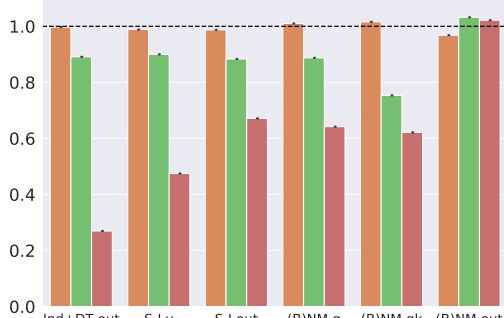 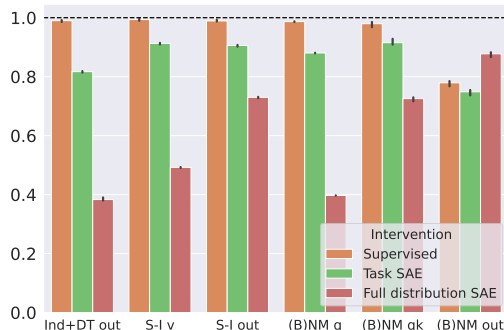

Figure 3: Sufficiency (left) and necessity (right) evaluations of reconstructions of cross-sections of the IOI circuit computed using supervised feature dictionaries, task- and full-distribution SAEs. **Left**: average logit difference when replacing activations in cross-sections of the IOI circuit with their reconstructions, normalized by the average logit difference over the data distribution in the absence of intervention (a $y$-axis value of 1 is best). **Right**: drop in logit difference when deleting reconstructions, normalized by the respective drop when performing mean-ablation, and linearly rescaled so that values close to 1 are best. See Appendix 7.5 for details.

*Sufficiency.* We check if reconstructions are sufficient to do the task using the intervention $\mathbf{a}(p) \leftarrow \widehat{\mathbf{a}}(p)$, i.e., replacing activations $\mathbf{a}(p)$ with their reconstructions $\widehat{\mathbf{a}}(p)$ at circuit cross-sections. To normalize values, we report the metric

$$\text{sufficiencyscore} := \frac{\mathbb{E}_{p \sim \mathcal{D}}\left[\text{logitdiff}\left(\mathbf{a}(p) \leftarrow \widehat{\mathbf{a}}(p)\right)\right]}{\mathbb{E}_{p \sim \mathcal{D}}\left[\text{logitdiff}\left(\mathbf{id}\right)\right]}$$

where $\text{logitdiff}\left(\mathbf{id}\right)$ is the logit difference of the model in the absence of intervention, with expectation $\approx 3.3$ over our dataset. Results are shown in Figure 3 (left, orange bars)[4].

*Necessity.* We check if the reconstructions are necessary for the model to do the task. To do this, we intervene by replacing activations $\mathbf{a}(p)$ with their average plus the SAE error term:

$$\text{necessity-intv} := \mathbf{a}(p) \leftarrow \mathbb{E}_{p \sim \mathcal{D}}\left[\mathbf{a}(p)\right] + \underbrace{\left(\mathbf{a}(p) - \widehat{\mathbf{a}}(p)\right)}_{\text{error term}}$$

We measure the drop in logit difference that results from this intervention compared to no intervention, and compare to the corresponding drop when using the mean-ablation intervention to normalize[5]:

$$\text{necessity score} = 1 - \frac{\left|\mathbb{E}_{p \sim \mathcal{D}}\left[\text{logitdiff}\left(\text{necessity-intv}\right) - \text{logitdiff}\left(\mathbf{a} \leftarrow \mathbb{E}_{p \sim \mathcal{D}}\left[\mathbf{a}(p)\right]\right)\right]\right|}{\left|\mathbb{E}_{p \sim \mathcal{D}}\left[\text{logitdiff}\left(\mathbf{id}\right) - \text{logitdiff}\left(\mathbf{a} \leftarrow \mathbb{E}_{p \sim \mathcal{D}}\left[\mathbf{a}(p)\right]\right)\right]\right|}$$

Results are shown in Figure 3 (right, orange bars), rescaled linearly versus the logit difference in the absence of intervention.

**Control.** This evaluation measures the degree to which the supervised feature dictionaries disentangle the different attributes [6]. Specifically, supervised dictionaries suggest a straightforward way to edit the model's internal representation of the **IO**, **S** and **Pos** attributes via feature arithmetic. For

---

[4]We note that we report fractions of average logit differences; this ignores the distribution of logit differences across prompts, and raises the possibility that a few examples with very high logit differences may dominate the final score. However, in practice we observe that logit differences are concentrated close to their means.

[5]The intuition for this is that mean-ablation over the IOI distribution keeps shared information like grammar, syntax and non-IOI-specific semantics intact, which keeps the model somewhat on-distribution, but erases IOI-specific data. If the error term has useful information for the task, we expect the logit difference to drop less when we intervene with the error term than when we intervene with the mean-ablation.

[6]Note that, to meaningfully conclude disentanglement of attributes, they must be 'entangled' in the first place. This is true in the IOI task, where many circuit locations represent the **S** and **Pos** information simultaneously. We also find that one particular location, the queries of the L10H0 name mover head, represent all three attributes **IO**, **S** and **Pos**.

example, to change the value of the **IO** attribute in some activation $\mathbf{a}(p)$, we can define the edited activation

$$\mathbf{a}_{\mathbf{IO}\leftarrow x}(p) := \mathbf{a}(p) + \left(\mathbf{u}_{\mathbf{IO}=x} - \mathbf{u}_{\mathbf{IO}=\mathbf{IO}(p)}\right). \tag{3}$$

We want to check the extent to which this edit changes the **IO** attribute and only the **IO** attribute. We measure this in two ways:

- **probing accuracy (correlational effect)**: we train linear probes to predict the **IO**, **S** and **Pos** attributes from the activations $\mathbf{a}(p)$ of a single circuit location, and then we check whether their predictions are consistent with the edits. We find these probes have high accuracy overall w.r.t. both the attribute being edited-in, as well as the attributes that should be fixed (see Appendix Figure 16 for the results).

- **edit accuracy (causal effect)**: we measure the fraction of edits where an intervention such as $\mathbf{a}(p) \leftarrow \mathbf{a}_{\mathbf{IO}\leftarrow x}(p)$ results in the same next-token prediction as the 'ground truth' intervention $\mathbf{a}(p) \leftarrow \mathbf{a}(p_{\mathbf{IO}\leftarrow x})$. Here $p_{\mathbf{IO}\leftarrow x}$ is the prompt $p$ where the **IO** attribute is set to $x$, but all other data is kept the same. This is 'ground truth' because it directly patches the 'correct' activation our edit attempts to approximate. Results are shown in Figure 4 (red shapes, x-axis); some additional results are given in Appendix Figure 28 (orange bars).

**Sparsity of supervised feature interactions.** Supervised feature dictionaries are in a way emph-tautologically interpretable: they were defined to have a single feature activating for each possible value of each of the 3 attributes. In particular, the $F_1$ score of a supervised feature with respect to its ground-truth interpretation (e.g., **IO**='John') will be 1 by definition. Accordingly, we performed a more demanding test of interpretability: decomposing internal model computations in terms of interactions between individual features. We find that pre-softmax attention scores and composition between heads can be decomposed in terms of feature-level interactions, such that many interactions are close to zero, and the few non-zeros correspond to those expected based on the high-level IOI circuit description from Wang et al. (2022); see Appendix 7.9 for details, and especially Figures 8 and 9.

Having established our supervised feature dictionary framework and demonstrated its effectiveness on the IOI task, we now turn to the challenge of evaluating SAEs using this methodology.

## 5 EVALUATING SPARSE AUTOENCODERS

### 5.1 SAE TRAINING METHODOLOGY

We trained SAEs using a range of architectures and datasets. A central distinction in our experiments is between **task SAEs** trained on IOI dataset activations, and **full-distribution SAEs** trained on activations from the LLM's pre-training dataset OpenWebText (Gokaslan & Cohen, 2019).

For most experiments reported in this paper, we used the 'vanilla' SAE architecture following Bricken et al. (2023) with task SAEs. On the whole, we found vanilla task SAEs good at sufficiency/necessity, but struggling to control the model effectively.

This prompted us to experiment with more recent SAE architectures such as gated (Rajamanoharan et al., 2024) and topK (Gao et al., 2024) SAEs, as well as to better optimize how we choose SAEs for control. The specific details of these architectures are not relevant for this paper, and we refer the reader to the original papers for more details. For our purposes, all SAE architectures used come with a notion of reconstruction as a bias term plus a linear combination of a subset of SAE latents. For the training methodology of these newer variants, we adopted standard practices from recent works (Bricken et al., 2023; Rajamanoharan et al., 2024; Gao et al., 2024). Exhaustive details on SAE training algorithms and hyperparameters are given in Appendix 7.13. We found that these new architectures significantly increased performance in the sparse control evaluation.

### 5.2 CHALLENGES IN SAE EVALUATION

Our goal now is to extend the methodologies used to evaluate supervised feature dictionaries from Subsection 4.3 to deal with SAE latents. This involves two main challenges:

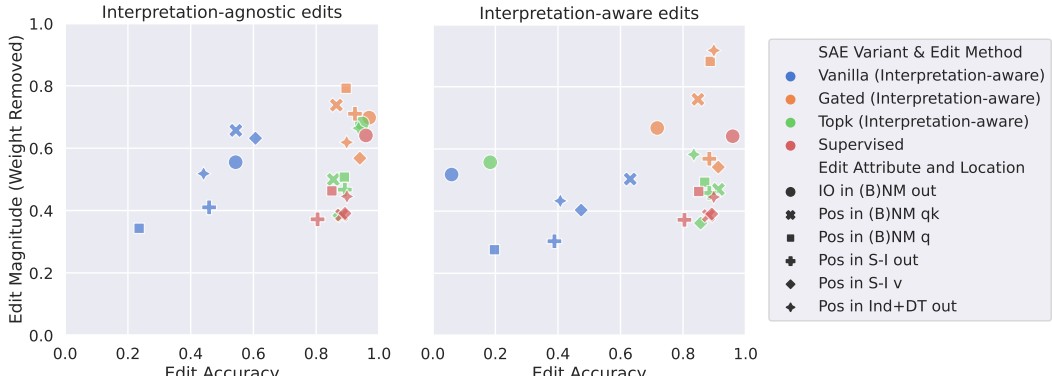

Figure 4: Activation editing results, showing the trade-offs between edit accuracy and edit magnitude (better is down and to the right). **Left**: results in the interpretation-agnostic editing regime, where we edit using the SAE latents best for each individual edit. **Right**: results in the interpretation-aware editing regime, where we edit using SAE latents with high F1 score for the attribute values being edited. Each color represents an SAE variant, and each shape a cross-section of the IOI circuit. See subsection 5.4 for more details on the two editing regimes.

**No built-in interpretations**: unlike our supervised features, which by definition correspond to values of the IOI attributes, SAE latents do not come with a built-in interpretation. This means that we must perform some kind of labeling/search over SAE latents when interested in e.g. using SAE latents to control the model;

**Multiple good solutions may exist**: even if the SAE latents do not correspond in a 1-to-1 manner to the supervised features, they may still be useful for approximating and controlling the IOI circuit w.r.t. the attributes **IO**, **S** and **Pos** we have chosen. A qualitative discussion of some concrete ways in which this might occur is given in Appendix 7.12. This means that we should not compare SAE latents to supervised features in a head-to-head manner, but rather allow for some flexibility in how SAE latents can be leveraged to achieve the outcomes we want.

With these challenges in mind, we now generalize the evaluations from Subsection 4.3 in the next sections.

## 5.3 SUFFICIENCY/NECESSITY

We note that the sufficiency/necessity evaluation applies as-is to SAE latents, as it only depends on the unambiguous notion of reconstruction. Results are shown in Figure 3 (green and red bars) for vanilla SAEs. We find that vanilla task SAEs are good at sufficiency/necessity, but full-distribution SAEs are not. We hypothesize this is because the task SAEs' training dataset (i.e., the IOI dataset) is a much better fit for the IOI task than the full pre-training dataset; this is consistent with our other observations that full-distribution SAEs do worse on the control evaluations.

## 5.4 SPARSE CONTROL VIA SAE LATENTS

**Expressing activation editing as an optimization problem over SAE latents.** To generalize the supervised edit from Equation (3) to the SAE setting, we need a way to pick SAE latents to remove/add to the clean activation $\mathbf{a}(p)$. In the supervised setting, we *always* have a *single* feature for e.g. **IO**='John' which we always add to the activation to change the **IO** attribute.

However, in the SAE setting, it may be too demanding to expect a single feature with this property to exist. For example, multiple SAE latents might be relevant to representing **IO**='John', but each might activate only in specific contexts.

Thus, to remain agnostic to the exact decomposition of the SAE latents, we propose to allow more flexibility to the SAE latents in the control evaluation. Suppose we have two prompts $p_s, p_t$ where

$p_s$ is the 'source' prompt that we want to intervene on, and $p_t$ is the 'target' prompt which expresses the edit we want to make; for example, we could have $p_t = (p_s)_{\textbf{IO} \leftarrow x}$ if we want to change the **IO** attribute to $x$. Let the SAE have a dictionary of decoder vectors $\{\mathbf{u}_j\}_{j=1}^m$, and the original and counterfactual activations $\mathbf{a}(p_s), \mathbf{a}(p_t)$ have reconstructions respectively $\widehat{\mathbf{a}}(p_s) = \sum_{i \in S} \alpha_i \mathbf{u}_i + \mathbf{b}_{dec}, \widehat{\mathbf{a}}(p_t) = \sum_{i \in T} \beta_i \mathbf{u}_i + \mathbf{b}_{dec}$ for $S, T \subset \{1, \ldots, m\}$ and $\alpha_i, \beta_i > 0$. We frame activation editing as the optimization problem

$$\min_{R \subset S, A \subset T, |R \cup A| \le k} \left\| \mathbf{a}(p_s) - \sum_{i \in R} \alpha_i \mathbf{u}_i + \sum_{i \in A} \beta_i \mathbf{u}_i - \mathbf{a}(p_t) \right\|_2 \tag{4}$$

In words, this problem asks for at most $k$ features to remove ($R$) from and/or add ($A$) to the original activation to bring it as close as possible to the counterfactual activation, where the features to add are taken directly from the counterfactual one (we also experimented with using the reconstructions instead of the actual activations in this algorithm, but results were worse). In general, this problem has no polynomial-time (in $k, |S|, |T|$) solution, as the NP-hard problem SUBSETSUM reduces to it; instead, we use a greedy algorithm to find a solution.

We stress that this formulation is heuristic, but can be readily replaced with a more principled one; we chose it because it is intuitive and cheap to optimize. Another natural choice would be to optimize the logit difference downstream of the intervention to match that of patching the true counterfactual activation directly.

**Preventing complete activation overwrites by measuring edit magnitude.** When editing activations, we must ensure our edits are minimal and targeted. A concerning failure mode would be to completely overwrite the original activation's features with those from the target, which would demonstrate no true disentanglement of attributes. Furthermore, depending on the SAE's structure, this situation can arise even if we remove 1 latent and add 1 latent back [7].

To measure the magnitude of the edit, we compare the contribution of the changed features to the reconstruction against the analogous quantity for our 'ideal' supervised feature dictionary. Namely, for each summand $f_i \mathbf{u}_i$ in a reconstruction $\widehat{\mathbf{a}}$, we assign a measure of its contribution

$$\text{weight}(i) := (f_i \mathbf{u}_i)^\top (\widehat{\mathbf{a}} - \mathbf{b}_{dec}) / \|\widehat{\mathbf{a}} - \mathbf{b}_{dec}\|_2^2 \tag{5}$$

so that $\sum_{i=1}^k \text{weight}(i) = 1$ [8]. Note that weights are additive in the features, so that the sum of weights of some subset of features is the weight for these features' total contribution to the reconstruction. We then measure the magnitude of an edit by the total weight of the features removed during the edit.

**Results.** We instantiate the optimization objective from Equation (4) in two main ways:

*Interpretation-agnostic*: here, we optimize the objective over all SAE latents active in the source and target activations. This is the most flexible regime.

*Interpretation-aware*: here, we pick SAE latents in the order of F1 score w.r.t. the relevant attribute value. For instance, if we want to change activation $\mathbf{a}(p_s)$ with **IO**='John' to activation $\mathbf{a}(p_t)$ with **IO**='Mary', we first order the SAE latents active in $\mathbf{a}(p_s)$ (resp. $\mathbf{a}(p_t)$) by their F1 score for **IO**='John' (resp. **IO**='Mary'), and then remove the top $k/2$ features from $\mathbf{a}(p_s)$ and add the top $k/2$ features from $\mathbf{a}(p_t)$.

Results on the trade-off between edit accuracy and edit magnitude for $k = 2$ are shown in Figure 4 for task SAE variants. Note that $k = 2$ is the smallest meaningful value we can use, and it has the same expressive power as our supervised edits, leading to a more fair comparison. Furthermore, topK SAEs allow us to directly set the sparsity level of the latents, and we chose 3 active latents per activation for this evaluation. This further matches the supervised feature dictionaries, which also have 3 'active' features per activation. Results for probing accuracy with topK SAEs ($k = 3$) are shown in Appendix Figure 16.

---

[7] As a simple but extreme example, consider an SAE which has a dedicated latent for each possible combination of values of the **IO**, **S** and **Pos** attributes (this can be achieved with $\sim 10^5$ features in our setup, far more than any SAE we train). Such an SAE can represent every IOI example perfectly (modulo random variation beyond the attributes) with just 1 active feature.

[8] While weights can in general take any real value, we find that in practice they are almost always approximately in $[0, 1]$; see Appendix 7.18 for empirical details.

We find that vanilla SAEs are markedly worse at this evaluation than supervised dictionaries, but gated and especially topK SAEs are competitive with supervised dictionaries, especially in the interpretation-agnostic regime. Further results for vanilla task SAEs with higher $k$ are shown in Appendix Figure 28 (for task SAEs). We find that higher $k$ values lead to only marginally better performance. Full-distribution vanilla SAEs do even worse on this task (Appendix Figure 27). As a control condition, we also consider running the same experiment with an SAE where the decoder vectors were frozen at initialization; this leads to significantly worse performance (Appendix Figure 26), indicating that the SAE latents are indeed learning useful information for the task.

## 5.5 Interpretability

To evaluate SAE latent interpretability, we frame an 'interpretation' as a subset $A$ of the IOI distribution's support, assigning interpretability scores via the $F_1$ score between a latent's active set $F$ and $A$. To avoid trivial interpretations where each latent simply matches its own active set, we restrict $A$ to meaningful human-defined subsets, including our primary attributes (**IO**, **S**, **Pos**), their intersections, and unions of attribute values. Given a latent, we pick the interpretation among these with the highest $F_1$ score. Details on the interpretation methodology are given in Appendix 7.14.

Our analysis reveals that many SAE latents have clear interpretations ($F_1 > 0.8$) consistent with their circuit locations, as shown in Appendix Figure 15 for full-distribution SAEs and in 29 (among others) for task SAEs. We note that these experiments are of a rather exploratory and qualitative nature; the main conclusion we draw is that there is a wide variety of interpretations for the SAE latents.

## 6 Discussion, Limitations and Conclusion

**Discussion.** We have taken steps towards more principled and objective evaluations of the usefulness of sparse feature dictionaries for disentangling LLM activations. In particular, we have demonstrated that simple supervised methods can be used as a principled way to compute high-quality feature dictionaries in a task-specific context, and that these supervised dictionaries can be used as 'skylines' to evaluate and contextualize the performance of unsupervised methods, such as SAEs. Moreover, while we have focused on the IOI task because of how well-studied it is, we have shown a proof of concept that our methods can be generalized straightforwardly to new tasks and models, as well as new distributions (Appendix 7.1).

**Limitations and Future Work.** A central limitation of our approach is that we impose the set of attributes we want to evaluate SAEs against. This influences many of our evaluations, in particular the 'sparse control evaluation' from Section 5.4, which relies on these attributes to compute the counterfactual activations used to evaluate edit accuracy.[9] This is somewhat mitigated by the fact that we use attributes shown to be consistent with the computation of the IOI circuit, and that we aim to be agnostic to the precise SAE representation of the task. We believe this assumption (conditioned on the supervised dictionaries successfully passing our tests) is a reasonable middle ground between assessing the usefulness of SAEs for model behaviors humans understand and wish to control, and evaluating SAEs fairly.

Another major limitation of our framework is that it deals with quite templatic tasks. We hope that future work can bridge this gap. For example, we find the approach of Gao et al. (2024) for evaluating SAE latents as probes for binary classification tasks to be a promising direction for building more fine-grained supervised feature dictionaries.

Finally, a third limitation is that it takes work to identify in a way independent of SAEs what good attributes for a given task are. We show through two case studies in Appendix 7.1 that, by targeting only key attributes in a single activation location, we can reduce this effort substantially. This comes at the cost of analyzing a more limited, but still potentially interesting slice of the task. However, we admit that these tasks are still somewhat contrived/templatic. For truly open-ended tasks, we believe that methods to generate 'concept vectors' at scale, such as the ones given in Luo et al. (2024); Zou et al. (2023), may be a promising direction for obtaining a starting point for 'ground-truth' feature dictionaries.

---

[9]However, note that the sufficiency and necessity evaluations are independent of the choice of attributes, as they evaluate SAE reconstructions as a whole.

ACKNOWLEDGEMENTS

We thank Sonia Joseph and Can Rager for feedback on earlier versions of this manuscript. We thank the SERI MATS program for making this research possible. A.M. and G.L. were supported by grants from AI Safety Support and Effective Ventures.

AUTHOR CONTRIBUTIONS

AM designed and ran the algorithms for computing supervised feature dictionaries and observed their theoretical properties; trained task-specific SAEs; designed the tests for sufficiency/necessity of reconstructions, sparse controllability and causal faithfulness of feature descriptions; ran these tests on full-distribution and task-specific SAEs; observed the occlusion/oversplitting phenomena in task-specific SAEs and developed the toy models for them; wrote all sections of the paper unless otherwise specified.

GL trained full-distribution SAEs; designed and ran the automatic feature scoring algorithms for them, and ran most of the interpretability evaluations for full-distribution SAEs; contributed to the sections on interpretability of full-distribution SAE features, created Figures 1 and 15.

NN provided guidance on research goals and prioritization, as well as writing, throughout the entire project; proposed the initial idea of finding supervised feature dictionaries for the IOI task, and using them to evaluate (unsupervised) feature dictionaries; refined the search for algorithms to compute the supervised dictionaries.

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

# 7 APPENDIX

## 7.1 GENERALIZING OUR METHODS TO OTHER MODELS, DISTRIBUTIONS, AND TASKS

In this section, we detail two new tasks and datasets we used to showcase the generalization of our methods.

### 7.1.1 THE GREATER-THAN TASK

First, we ran a proof-of-concept for the "greater-than" task (Hanna et al., 2024) to edit the attribute denoted "YY" in the outputs of 7 attn heads in the greater-than circuit.

**Task, dataset, and model.** We use the greater-than task as-is, with the template

'The $NOUN lasted from the year 17$YY to the year 17'

where YY is a two-digit number that concatenates with 17 to form a year, and we restrict years to be between 1701 and 1720. We intervene on the outputs of the attention heads L5H1, L5H5, L6H1, L6H9, L7H10, L8H8, and L8H11 in order to change the attribute YY. The LLM used is the same as in the main paper, the GPT-2 Small model. We sample several thousand prompts for training and testing.

**SAE training.** We train vanilla task SAEs with an $\ell_1$ regularization coefficient of 5.0; we follow the methodology of Bricken et al. (2023); Conerly et al. (2024) for training the SAEs.

**Sparse control results.** Exchanging as few as 2 SAE latents matched next-token predictions 90% of the time when patching attention heads with counterfactual activations (compared to 38% for no intervention; this is a binary classification task). Results are in Figure 5.

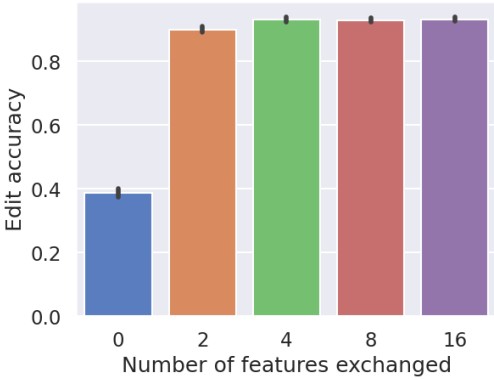

Figure 5: Results for sparse control on the greater-than task. We show the accuracy of next-token predictions (against counterfactual labels) after patching attention heads with SAE latents, compared to patching with ground-truth counterfactual activations.

### 7.1.2 THE 'BOTH' TASK

To showcase the flexibility of our methods, we have evaluated them in a setup that departs in several major ways from the main results reported in the paper:

- We evaluate on a new task in-context learning task we created for this paper. The task is based on setting up a prompt where the model can apply induction (i.e., predict that the next token is the same as some earlier token in the context) in one of two ways; our experiments are centered on disentangling the two different ways the model can apply induction.
- We evaluate on a new model and pre-training dataset, the 4-layer 33M-parameter model trained on the Tiny Stories dataset (Eldan & Li, 2023).
- Whereas for most of the paper we have focused on task-specific SAEs, here we evaluate topK SAEs trained on the same pre-training dataset as the model (Tiny Stories).

This section is a proof of concept that our methods can be straightforwardly generalized to new models and datasets.

**Task.** The task uses a template of the form

Once upon a time, there live a kid named $NAME. $NAME really loved $ANI-MALS. $NAME also really loved $SPORT. So $NAME really loved both

where a correct next-token prediction is either '$ANIMALS' or '$SPORT'; we refer to these as the two **attributes** for the task.

In our experiments, we sample a set of 1000 prompts where we randomly and independently sample from a set of 25 names, 6 animal species, and 9 sports. We observe the following key properties of the task:

- While at temperature 0 the model will always prefer to start its completion with one of the two attributes (e.g., generate 'cats and football'), we can still measure that at the 'both' token the model places higher probability on the correct attribute values compared to incorrect ones.

- Specifically, we define the **restricted logit accuracy** with respect to an attribute, which is the fraction of examples where the correct attribute value has the highest logit among all values for this attribute. For example, for the 'animal' attribute, an example where $ANIMALS='cats' will be considered accurate if the logit for 'cats' is higher than the logit for each of the other 5 animal species in our dataset.

- We find that in our dataset, the restricted logit accuracy for sport is $85.5\%$, and for animal is $75.9\%$. Moreover, we find that the residual stream representation of the 'both' token at the last (4th) layer of the model is causal w.r.t. this metric. Specifically, we can activation-patch this location from a prompt where we keep one attribute fixed but change the other. We observe that the accuracy w.r.t. the attribute value being patched-in is $77.1\%$ for animal and $85.6\%$ for sport, while the accuracy w.r.t. the attribute that is fixed remains mostly unchanged (at most $1 - 2\%$ difference from clean).

**Supervised feature dictionaries.** We create supervised features for the two attributes like in the main paper by using conditional expectation over examples in the dataset having a given value of the attribute.

We observe that the supervised features are able to recover the causal effect of the attribute value being patched-in on the restricted logit accuracy, suggesting that we have found meaningful features for the task. Specifically, when we edit the animal attribute, the restricted logit accuracy w.r.t. the animal being edited-in is $73\%$, while the accuracy w.r.t. the (fixed) sport attribute remains high at $82.8\%$. Similar results are found when editing the sport attribute.

**Sparse control with TopK SAEs.** We train topK SAEs with $k \in \{32, 64\}$ and number of latents in $\{8192, 16384\}$. We find that these SAEs achieve a good value of variance explained in the activations.

We perform a variant of the sparse control evaluation from Section 5.4 where we edit each attribute in isolation, using counterfactual prompts where only this attribute has changed. This is analogous to the intervention we use to evaluate our supervised feature dictionaries.

We explore changing a number of SAE latents in $\{2, 4, 6, 8\}$, and observe that this intervention is not sufficient to recover the full causal effect of our supervised feature dictionaries. Results are shown in Figure 6.

## 7.2 QUALITATIVE PHENOMENA IN SAE LEARNING

### 7.2.1 FEATURE OCCLUSION

Our experiments suggest that when two (causally relevant) attributes are represented in the same activation, but one attribute's (supervised) features have overall higher magnitude, SAEs have a tendency to robustly learn more interpretable features for the attribute with higher magnitude. We observe this in the queries of the L10H0 name mover head at the END token in the IOI circuit, where the **IO** and **S** attributes are both represented and causally relevant (ablating either leads to a change of $\approx 0.5$ in logit difference). We find that our task SAEs consistently find features with high $F_1$ score for individual **IO** names, but fail to find a significant number of features for individual **S** names. In Figure 7 (left), we show interpretability results from training SAEs over a wide grid of hyperparameters that confirm this observation; more details in Appendix 7.16.

*Hypothesis: feature magnitude is a driver of occlusion.* We noticed that the supervised features for **IO** and **S** names in the L10H0 queries have a small but significant difference in norm (see Appendix

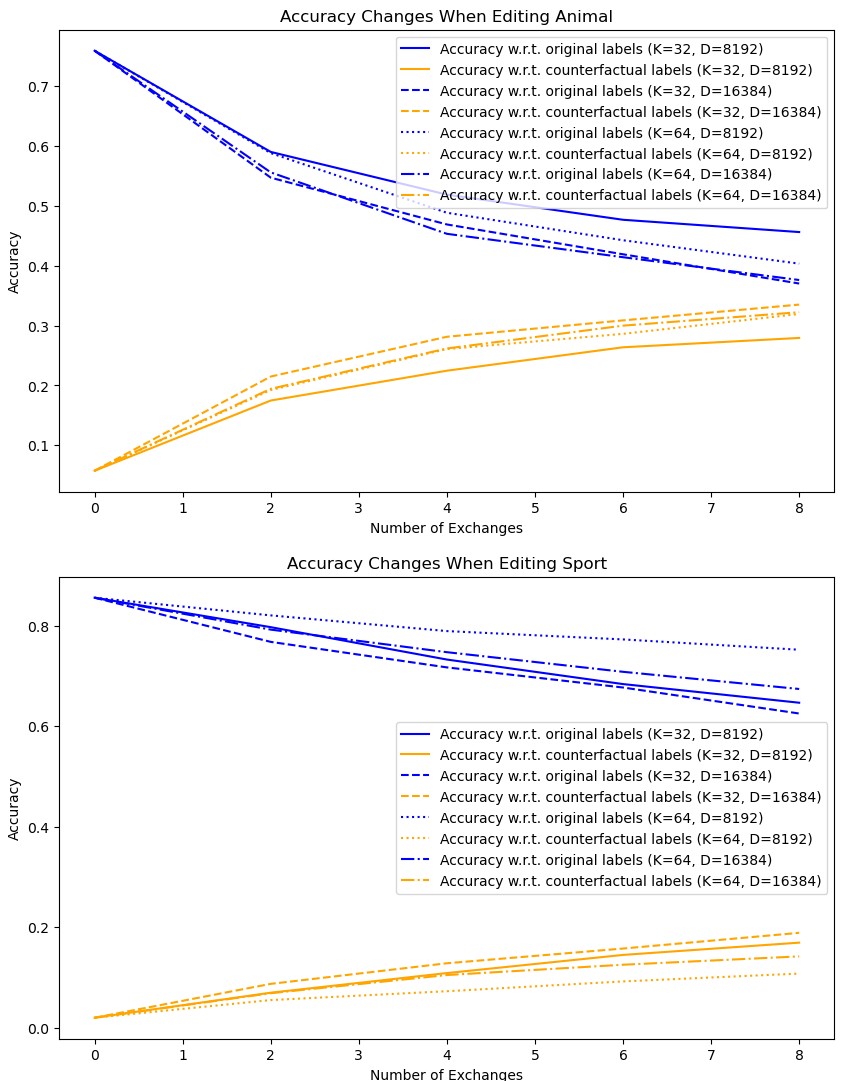

Figure 6: Sparse control results for the new task. We plot how the restricted logit accuracy w.r.t. the original (blue) and counterfactual (orange) attribute values changes as we edit more SAE latents. The accuracy w.r.t. the other attribute is not shown as it remains high and nearly unchanged.

Figure 13 (left)). We then hypothesized that feature magnitude is a factor in this phenomenon. To check this, we surgically reduce the magnitude of **IO** features in the activations using our supervised feature dictionaries, and observe that the number of **S** features discovered in these modified activations monotonically increases as we remove larger fractions of the **IO** features (Figure 7 (right); see Appendix 7.16 for methodology). We furthermore constructed a simple toy model based on i.i.d. isotropic random features that mimic the norms of supervised features in the L10H0 queries, and find that a similar phenomenon occurs for the distribution of features with high $F_1$ score (e.g. higher than 0.9; see Appendix 7.16).

### 7.2.2 FEATURE OVER-SPLITTING

We also found in our experiments that task SAEs have a tendency to split a single binary attribute into multiple features, even when the number of features available could in principle be spent on other attributes. Note that, while this behavior may be counter-intuitive from a human standpoint, it does not necessarily mean that the SAE failed; it may be that the binary attribute is not part of an optimal sparse description of the model's internal states. We observe this phenomenon with the **Pos**

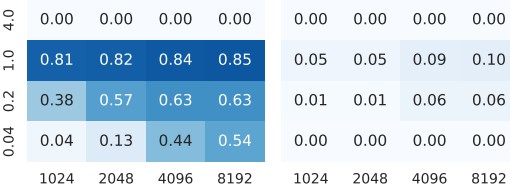 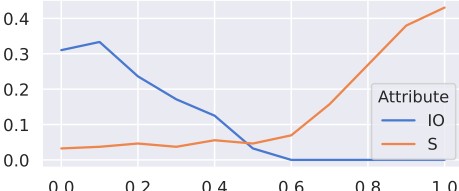

Figure 7: **Left**: Fraction of **IO** (left subplot) and **S** (right subplot) names in our dataset for which a feature with $F_1$ score $\geq 0.5$ is found, as a function of dictionary size (x-axis) and effective $\ell_1$ regularization coefficient (y-axis), over a wide hyperparameter sweep for the queries of L10H0. **Right**: fraction of **IO** and **S** names in our dataset for which a feature with $F_1$ score $\geq 0.5$ is found, as a function of $\alpha$ (x-axis), the fraction of supervised **IO** features we subtract from the activations.

attribute in the IOI task (again in the queries of the L10H0 name mover), which is robustly split into many (e.g. $\geq 10$) features by our SAEs that activate for small, mostly non-overlapping subsets of examples sharing the same **Pos** value that have no clear semantic interpretation.

*Is over-splitting a form of over-fitting?* To investigate whether this is due to overfitting, we compared **Pos** features between (1) different random seeds for the same training dataset and (2) different training datasets. In both cases, we found that the **Pos** features discovered are similar above chance levels, suggesting that the over-splitting is not due solely to overfitting to randomness in the training algorithm and/or dataset.

*Reproducing the over-splitting phenomenon in a simple toy model.* On the other hand, we show empirical evidence that in a toy model where activations are a uniform mixture of two isotropic Gaussian random variables, an appropriately *randomly initialized* SAE with enough hidden features will achieve lower total loss than an ideal SAE with just two features corresponding to the two components of the mixture. Such randomized constructions exist for *any* $\ell_1$ regularization coefficient, even in the limit of infinite training data. Details are given in Appendix 7.17.

## 7.3 ADDITIONAL DETAILS ON THE IOI CIRCUIT

**Circuit structure.** To refer to individual token positions within the sentence, we use the notation of Wang et al. (2022): IO denotes the position of the **IO** name, S1 and S2 denote respectively the positions of the first and second occurrences of the **S** name (with S1+1 being the token position after S1), and END denotes the last token in the sentence (at the word 'to').

Wang et al. (2022) suggest the model uses the algorithm 'Find the two names in the sentence, detect the repeated name, and predict the non-repeated name' to do this task. Specifically, they discover several classes of heads in the model, each of which performs a specific subtask of this overall algorithm. A simplified version of the circuit involves the following three classes of heads and proceeds as follows:

- **Duplicate token heads**: these heads detect the repeated name in the sentence (the **S** name) and output information about both its position and identity to the residual stream[10]

- **S-Inhibition heads**: these heads read the identity and position of the **S** name from the residual stream, and output a signal to the effect of 'do not attend to this position / this token identity' to the residual stream

- **Name Mover heads**: these are heads that attend to names in the sentence. Because the signal from the S-Inhibition heads effectively removes the **S** name from the attention of these heads, they read the identity of the **IO** name from the input prompt, and copy it to the last token position in the residual stream.

---

[10]We follow the conventions of Elhage et al. (2021) when describing internals of transformer models. The residual stream at layer $k$ is the sum of the output of all layers up to $k-1$, and is the input into layer $k$.

In reality, the circuit is more nuanced, with several other classes of heads participating: previous token heads, induction heads (Olsson et al., 2022), backup name mover heads, and negative name mover heads. In particular, the circuit exhibits *backup behavior* (McGrath et al., 2023) which poses challenges for interpretability methods that intervene only on single model components at a time. We refer the reader to Figure 2 for a schematic of the full circuit, and to Wang et al. (2022) for a more complete discussion.

## 7.4  $F_1$ SCORE NOTES

An $F_1$-score of $\alpha$ guarantees that both precision and recall are at least $\frac{\alpha}{2-\alpha}$. For example, when $\alpha = 0.8$ (the value we use in most evaluations), both precision and recall are at least $0.8/1.2 \approx 0.67$. Requiring a sufficiently high $F_1$ value is important in order to avoid labeling a trivial feature as meaningful for attributes where $|A|$ is large, because then a feature active for all examples can have a high $F_1$-score.

The $F_1$ score has some limitations in the context of our work:

- it does not take into account the magnitude of the feature activations; for instance, a feature that is active for all examples in $S$ but only has high activation values on the examples in $A$ may have a low $F_1$ score, even though it is in some sense highly informative for the attribute $A$.

- it is a very conservative metric, in that it requires both high precision and high recall to be high. For example, a feature with precision $0.5$ but recall $0.02$ will have an $F_1$ score of $\sim 0.04$, heavily skewed towards the lower of the two metrics, even though it is in some sense informative for the attribute $A$.

We hope to address these limitations in future work.

## 7.5  ADDITIONAL DETAILS FOR SECTION 4.3

**Computing and evaluating supervised feature dictionaries.** For each parametrization and each method to compute feature dictionaries, we use 20,000 prompts sampled from our IOI distribution (see Appendix 7.6) to compute feature dictionaries for the query, key, value, and attention output (i.e., attention-weighted values) of the relevant token positions of all 26 heads identified in Wang et al. (2022) (recall Figure 2). We use another sample of 5,000 prompts to validate the quality of the feature dictionaries.

**Cross-sections of the circuit.** Based on the understanding of the IOI circuit from Wang et al. (2022), we identify several cross-sections of the computational graph of the IOI circuit where feature editing is expected to have effects meaningful for the task:

- *outputs of (backup) name mover heads at END ((B)NM out)*: these activations encode the **IO** name and write it to the END token of the residual stream. We expect that editing the **IO** name in these activations will directly affect the model's prediction, while editing other attributes will not have a significant effect.

- *queries+keys of (backup) name movers at END ((B)NM qk)*: the queries represent the **S** name and **Pos** information, but they are mainly used as *inhibitory* signals for the model, decreasing the attention to the **S** token[11]. The keys represent information about the **IO** and **S** names: in particular, the **S** information combines with the query to inhibit attention to the **S** token.

  We expect that editing the **S** and **Pos** attributes in *both* the keys and queries will not significantly hurt model performance, because as a result attention to the **S** token will again be inhibited. By contrast, it is unclear what editing the **IO** name is expected to do, since its role in the attention computation is not fully described in Wang et al. (2022).

- *outputs of S-Inhibition heads at END (S-I out), values of S-Inhibition heads at S2 (S-I v), and outputs of duplicate token and induction heads at S2 (Ind+DT out)*: these activations

---

[11]In addition, we later find that the queries of the L10H0 name mover head also represent the **IO** attribute, and serve an *inhibitory* role for it as well, decreasing the attention to the **IO** token.

transmit the inhibitory signal to the name mover heads through the residual stream. We expect that editing **S** and **Pos** in these activations will lower the model's logit difference by disrupting the inhibitory signal, while editing **IO** will have no effect.

**Evaluating necessity of feature reconstructions**: When we intervene on the model by removing reconstructions from activations in cross-sections of the circuit, model performance on the IOI task (as measured by the logit difference) goes down from the clean value $\text{logitdiff}_{\textbf{clean}}$ to a lower value $\text{logitdiff}_{\textbf{intervention}}$. As we describe in the main text, the ground-truth intervention for removing the features from the activations is mean ablation of the corresponding cross-section, which also results in a lower value of the logit difference, $\text{logitdiff}_{\textbf{mean ablation}}$. We want to measure the degree to which $\text{logitdiff}_{\textbf{intervention}}$ approximates $\text{logitdiff}_{\textbf{mean ablation}}$, in a way that normalizes for different values of $\text{logitdiff}_{\textbf{mean ablation}}$ across cross-sections of the circuit. We use the following metric to do this:

$$\text{necessity score} = 1 - \frac{|\text{logitdiff}_{\textbf{mean ablation}} - \text{logitdiff}_{\textbf{intervention}}|}{|\text{logitdiff}_{\textbf{mean ablation}} - \text{logitdiff}_{\textbf{clean}}|}.$$

## 7.6  DATASET, MODEL AND EVALUATION DETAILS FOR THE IOI TASK

We use GPT2-Small for the IOI task, with a dataset that spans 216 single-token names, 144 single-token objects and 75 single-token places, which are split $1 : 1$ across a training and test set. Every example in the data distribution includes (i) an initial clause introducing the indirect object (**IO**, here 'Mary') and the subject (**S**, here 'John'), and (ii) a main clause that refers to the subject a second time. Beyond that, the dataset varies in the two names, the initial clause content, and the main clause content. Specifically, use three templates as shown below:

> Then, [ ] and [ ] had a long and really crazy argument. Afterwards, [ ] said to
> Then, [ ] and [ ] had lots of fun at the [place]. Afterwards, [ ] gave a [object] to
> Then, [ ] and [ ] were working at the [place]. [ ] decided to give a [object] to

and we use the first two in training and the last in the test set. Thus, the test set relies on unseen templates, names, objects and places. We used fewer templates than the IOI paper Wang et al. (2022) in order to simplify tokenization (so that the token positions of our names always align), but our results also hold with shifted templates like in the IOI paper.

On the test partition of this dataset, GPT2-Small achieves an accuracy of $\approx 91\%$. The average difference of logits between the correct and incorrect name is $\approx 3.3$, and the logit of the correct name is greater than that of the incorrect name in $\approx 99\%$ of examples. Note that, while the logit difference is closely related to the model's correctness, it being $> 0$ does not imply that the model makes the correct prediction, because there could be a third token with a greater logit than both names.

## 7.7  PROPERTIES OF MEAN FEATURE DICTIONARIES

Mean feature dictionaries enjoy several convenient properties:

- The vectors $\mathbf{u}_{iv}$ for an attribute $a_i$ do not depend on which other attributes $a_l \neq a_i$ we have chosen to describe the prompt $p$ with.

- If an attribute $i$ is not linearly represented in the activations, the mean code features $\mathbf{v}_{iv} \to 0$ in the limit of infinite data (see below). In particular, this also holds if the attribute is not represented *at all* in the activations.

This suggests that mean feature dictionaries are robust to the inclusion of irrelevant or non(-linearly)-represented attributes, which is a desirable property in real settings where we may not know the exact attributes present in each activation. However, mean feature dictionaries are *not* robust to the inclusion of redundant attributes, as the lack of interaction between the attributes means that redundant attributes cannot 'coordinate' to reduce the reconstruction error $\|\mathbf{a} - \widehat{\mathbf{a}}\|_2^2$.

### 7.7.1 MEAN FEATURES ARE ZERO FOR NON-LINEARLY-REPRESENTED ATTRIBUTES.

Suppose we have a random vector $\mathbf{x}$ for a $k$-way classification task with one-hot labels $\mathbf{z} \in \mathcal{Z} = \{\mathbf{z} \in \{0,1\}^k \text{ s.t. } \|\mathbf{z}\|_1 = 1\}$. In Section 3 of Belrose et al. (2023), it is shown that the following are equivalent:

- the expected cross-entropy loss of a linear predictor $\widehat{\mathbf{z}} = \mathbf{w}^\top \mathbf{x} + \mathbf{b}$ for $\mathbf{z}$ is minimized at a *constant* linear predictor. In other words, the optimal logistic regression classifier (in the limit of infinite data) is no better than the optimal constant predictor (which, at best, always predicts the majority class).
- the class-conditional mean vectors $\mathbb{E}[\mathbf{x}|\mathbf{z} = e_i]$ are all equal to the overall mean $\mathbb{E}[\mathbf{x}]$ of the data.

If we translate this to the context of mean feature dictionaries, we have that logistic regression for the value of an attribute $a_i$ will degenerate to the majority class predictor if and only if the mean feature dictionaries for all values of this attribute are zero. In the finite data regime, this gives us some theoretical grounds to expect that the mean feature dictionaries will be significantly away from zero if and only if the attribute's values can be non-trivially recovered by a (logistic) linear probe. As a special case, if an attribute is not represented in the data at all, we expect the mean feature dictionaries for this attribute to be zero.

## 7.8 DEFINITION AND PROPERTIES OF MSE FEATURE DICTIONARIES

The main alternative method we considered to compute supervised feature dictionaries from attributes was **MSE feature dictionaries**, which use a least-squares linear regression to predict the activations from the attribute values. We note that mean feature dictionaries work well in our setting because the attributes we choose in the IOI task are probabilistically independent w.r.t. the IOI distribution we work with. We recommend using MSE dictionaries when attributes are not probabilistically independent.

MSE feature dictionaries compute $\mathbf{u}_{a_i(\cdot)=v}$ by directly minimizing the $\ell_2$ reconstruction error over the centered activations:

$$\{\mathbf{u}_{a_i(\cdot)=v}\}_{i \in I, v \in S_i} = \underset{\mathbf{u}_{a_i(\cdot)=v}}{\arg\min} \frac{1}{N} \sum_{k=1}^{N} \left\| (\mathbf{a}(p_k) - \overline{\mathbf{a}}) - \sum_{i \in I} \mathbf{u}_{a_i(\cdot)=a_i(p_k)} \right\|_2^2 \tag{6}$$

This objective is convex, and is equivalent to a least-squares regression problem; in fact, the optimal solutions take a form very similar to the mean feature dictionaries (see below). Furthermore, this objective closely mimics the SAE objective: here, the sparsity is hard-coded, leaving only the $\ell_2$ objective.

We next discuss some properties of the MSE feature dictionaries. For brevity, in the remainder of this section we write $\mathbf{u}_{iv}$ instead of $\mathbf{u}_{a_i(\cdot)=v}$.

### 7.8.1 MSE FEATURE DICTIONARIES AS A MULTIVARIATE LEAST-SQUARES REGRESSION PROBLEM.

Let $S = \sum_{i=1}^{N_A} |S_i|$ be the total number of possible values for all attributes. For each attribute $i$, consider the characteristic matrix $C_i \in \mathbb{R}^{N \times S_i}$ of the dataset for this attribute, where

$$C_{kj} = \begin{cases} 1 & \text{if } a_i(p^{(k)}) = v_j \\ 0 & \text{otherwise} \end{cases}$$

for some ordering $(v_1, \ldots, v_{|S_i|})$ of the values in $S_i$, and let $C = \begin{bmatrix} C_1 & C_2 & \cdots & C_{N_A} \end{bmatrix} \in \mathbb{R}^{N \times S}$ be the concatenation of all characteristic matrices. Also, let $A \in \mathbb{R}^{N \times d}$ be the matrix of activations with rows $\mathbf{a}^{(k)}$. Then the objective function for the MSE feature dictionaries can be written as the multivariate least-squares regression problem

$$\min_{U \in \mathbb{R}^{S \times d}} \frac{1}{N} \|A - CU\|_F^2$$

where the rows of $U$ are the vectors $\mathbf{u}_{iv}$ across all $i$ and $v \in S_i$, with the optimal solution given by

$$U^* = \left(C^\top C\right)^+ C^\top A \tag{7}$$

### 7.8.2 MSE FEATURE DICTIONARIES AS AVERAGING OVER EXAMPLES.

Using the special structure of the objective, we can also derive some information about the optimal solutions $\mathbf{u}_{iv}^*$. Namely, at optimality we should not be able to decrease the value of the objective by changing a given $\mathbf{u}_{iv}^*$ away from its optimal value. The terms containing $\mathbf{u}_{iv}^*$ in the objective are

$$\frac{1}{N} \sum_{k \in P_{iv}} \left\| \mathbf{a}^{(k)} - \sum_{l \neq i} \mathbf{u}_{lv_l^{(k)}}^* - \mathbf{u}_{iv}^* \right\|_2^2 = \frac{1}{N} \sum_{k \in P_{iv}} \left\| \left( \mathbf{a}^{(k)} - \sum_{l \neq i} \mathbf{u}_{lv_l^{(k)}}^* \right) - \mathbf{u}_{iv}^* \right\|_2^2$$

$$= \frac{1}{N} \sum_{k \in P_{iv}} \left\| \bar{\mathbf{a}}^{(k)} - \mathbf{u}_{iv}^* \right\|_2^2$$

where recall that $P_{iv} = \{k \,|\, a_i(p^{(k)}) = v\}$, and $\bar{\mathbf{a}}^{(k)}$ is the residual of $\mathbf{a}^{(k)}$ after subtracting the reconstruction using all other attributes $l \neq i$. Since this value cannot be decreased by changing $\mathbf{u}_{iv}^*$, we have that it equals the minimizer of this term (holding $\bar{a}^{(k)}$ fixed). In other words, if we define

$$f(\mathbf{u}) = \frac{1}{N} \sum_{k \in P_{iv}} \left\| \bar{\mathbf{a}}^{(k)} - \mathbf{u} \right\|_2^2$$

we have that $\mathbf{u}_{iv}^* = \arg\min_{\mathbf{u}} f(\mathbf{u})$. Since $f$ is a sum of convex functions, it is itself convex, and so the first-order optimality condition is also sufficient for optimality. We have

$$\nabla f(\mathbf{u}) \propto \sum_{k \in P_{iv}} \left( \bar{\mathbf{a}}^{(k)} - \mathbf{u} \right) \propto \frac{1}{|P_{iv}|} \sum_{k \in P_{iv}} \bar{\mathbf{a}}^{(k)} - \mathbf{u}$$

and so

$$\mathbf{u}_{iv}^* = \frac{1}{|P_{iv}|} \sum_{k \in P_{iv}} \bar{\mathbf{a}}^{(k)} \tag{8}$$

Note that this is very similar to the definition of mean feature dictionaries, but also importantly different, because the optimal $\mathbf{u}_{iv}^*$ depends on the optimal values of the feature dictionaries for the other attributes.

### 7.8.3 MSE FEATURE DICTIONARIES WITH INDEPENDENT ATTRIBUTES.

Finally, we can prove that, under certain conditions, attributes for which $\mathbb{E}[\mathbf{a}|a_i(p) = v_i] = \mathbb{E}[\mathbf{a}]$, i.e. the conditional mean of activations over values of the attribute is the same as the overall mean (assuming both means exist), will have (approximately) constant MSE feature dictionaries $\mathbf{u}_{iv} = \mathbf{u}_i \forall v$. This is a counterpart to the result from Appendix 7.7 for MSE feature dictionaries:

**Lemma 7.1.** *Suppose that all conditional means $\mathbb{E}_{p \sim \mathcal{D}}[\mathbf{a}|a_i(p) = v]$ exist for all $i, v \in S_i$. Let $a_i$ be an attribute such its values appear independently from the values of all other attributes, i.e.*

$$\mathbb{P}_{p \sim \mathcal{D}}[a_i(p) = v_i, a_l(p) = v_l] = \mathbb{P}_{p \sim \mathcal{D}}[a_i(p) = v_i] \, \mathbb{P}_{p \sim \mathcal{D}}[a_l(p) = v_l] \quad \forall v_i \in S_i, v_l \in S_l, l \neq i$$

*Then, in the limit of infinite training data, the conditional means $\mathbb{E}[\mathbf{a}|a_i(p) = v]$ are all equal to the overall mean $\mathbb{E}[\mathbf{a}]$ if and only if the optimal MSE feature dictionaries $\mathbf{u}_{iv}^*$ for this attribute are constant with respect to the value $v$ of the attribute, i.e. $\mathbf{u}_{iv}^* = \mathbf{u}_i$ for all $v \in S_i$.*

*Proof.* From Equation 8, we have

$$\mathbf{u}_{iv}^* = \frac{1}{|P_{iv}|} \sum_{k \in P_{iv}} \bar{\mathbf{a}}^{(k)} = \frac{1}{|P_{iv}|} \sum_{k \in P_{iv}} \left( \mathbf{a}^{(k)} - \sum_{l \neq i} \mathbf{u}_{lv_l^{(k)}}^* \right)$$

$$= \frac{1}{|P_{iv}|} \sum_{k \in P_{iv}} \mathbf{a}^{(k)} - \frac{1}{|P_{iv}|} \sum_{k \in P_{iv}} \sum_{l \neq i} \mathbf{u}_{lv_l^{(k)}}^*$$

The first term converges to $\mathbb{E}[\mathbf{a}|a_i(p) = v]$. The second term is a sum of terms of the form

$$\frac{1}{|P_{iv}|} \sum_{k \in P_{iv}} \mathbf{u}_{lv_l^{(k)}}^* = \frac{1}{|P_{iv}|} \sum_{v_l \in S_l} \mathbf{u}_{lv_l}^* \left| \{k \text{ s.t. } a_i(p_k) = v, a_l(p_k) = v_l\} \right| \tag{9}$$

for $l \neq i$. Since we are assuming $a_i$ is uncorrelated with $a_l$, in the limit of the size $N$ of the dataset $\mathbf{a}^{(1)}, \mathbf{a}^{(2)}, \ldots, \mathbf{a}^{(N)}$ going to infinity, $|\{k \text{ s.t. } a_i(p_k) = v, a_l(p_k) = v_l\}|$ will approach $|P_{iv}| \mathbb{E}_{p \sim \mathcal{D}} \left[ \mathbf{1}_{a_l(p) = v_l} \right]$. Moreover, note that in the closed-form solution $U^* = \left( C^\top C \right)^+ C^\top A = \left( \frac{C^\top C}{N} \right)^+ \frac{C^\top}{N} A$ from Equation 7, the matrix $\frac{1}{N} C^T C$ converges to some limit $\Sigma \in \mathbb{R}^{S \times S}$ as $N \to \infty$, and the matrix $\frac{1}{N} C^\top A$ similarly converges to some limit $M \in \mathbb{R}^{S \times d}$ by the assumption that all conditional means for all attributes exist. Thus, the optimal feature dictionaries $\mathbf{u}_{iv}^*$ will also converge as $N \to \infty$. So we see that the sum in Equation 9 will converge to a value that is independent of the value $v$ for the attribute $a_i$.

Thus, if the conditional means $\mathbb{E}\left[\mathbf{a}|a_i(p) = v\right]$ are all equal to the overall mean $\mathbb{E}\left[\mathbf{a}\right]$, we get that $\mathbf{u}_{iv}^*$ is independent of $v$; conversely, if $\mathbf{u}_{iv}^*$ is independent of $v$, we get that the conditional means are all equal to the overall mean. This completes the proof.

$\square$

## 7.9 Feature-level mechanistic analyses for Section 4.3

Since each activation is approximated as the sum of several vectors from a finite set, it becomes possible to decompose the model's internal operations in terms of elementary interactions between the learned vectors themselves. In the current paper, we are particularly interested in attention heads, as they are the building blocks of the IOI circuit. We consider the following subspace-level analyses:

- **Attention scores**: The attention mechanism is considered to be a crucial reason for the success of LLMs (Vaswani et al., 2017), but a subspace-level understanding of it is mostly lacking (but see Lieberum et al. (2023)). How do the features in the keys and queries of attention heads combine to produce the attention scores? Which feature pairs are most important for the head's behavior?

- **Head composition**: If we are to understand a circuit on the subspace level, we need to develop a subspace-level account of how the outputs of one attention head compose with the queries, keys and values of a downstream head in the circuit. Each head adds its output to the residual stream, and downstream heads' query/key/value matrices read from the residual stream. We can thus examine the contribution, or *direct effect*, of a head's output to another head's queries/keys/values. We can decompose this direct effect in terms of the features of the source head to calculate contributions of each feature to the direct effect.

Implementation details for these analyses follow.

**Attention scores.** Given feature dictionary reconstructions for the keys and queries of an attention head at certain positions

$$\mathbf{k} \approx \sum_{i \in I} \mathbf{u}_{a_i(\cdot) = a_i(p)}, \quad \mathbf{q} \approx \sum_{i \in I} \mathbf{v}_{a_i(\cdot) = a_i(p)}$$

we can decompose the attention scores as a sum of pairwise dot products between the dictionary features

$$\mathbf{q}^T \mathbf{k} / \sqrt{d_{head}} \approx \sum_{i,j \in I} \mathbf{v}_{a_i(\cdot) = a_i(p)}^T \mathbf{u}_{a_j(\cdot) = a_j(p)} / \sqrt{d_{head}}$$

where $d_{head}$ is the dimension of the attention head. This allows us to examine which feature combinations are most important for the head's attention according to the learned dictionaries. Variants of this decomposition can also be applied to e.g. the difference in attention scores at two different token positions.

**Head composition.** Following the terminology and results from Elhage et al. (2021), the residual stream $\mathbf{r}_{l,t}$ of a transformer at a given layer $l$ and token position $t$ is the sum of the input embedding and the outputs of all earlier MLP and attention layers at this position. The residual stream is in turn the input to the next attention layer; so, for example, we can write the query vector for the $h$-th head at layer $l$ and token $t$ as

$$\mathbf{q}_{l,t,h} = W_{l,h}^Q \operatorname{LayerNorm}\left(\mathbf{r}_{l,t}\right) = W_{l,h}^Q \operatorname{LayerNorm}\left(\bar{\mathbf{r}}_{l,t} + W_{l',h'}^O \mathbf{z}_{l',t,h'}\right)$$

where $\mathbf{z}_{l',t,h'}$ is the attention-weighted sum of values of the $h'$-th head at layer $l' < l$ and token $t$, $\bar{\mathbf{r}}_{l,t}$ is the remainder of the residual stream after removing the contribution of this head, and LayerNorm is the model's layer normalization operation (Ba et al., 2016) before the attention block in layer $l$. By treating the layer normalization as an approximately linear operation (taking the scale from an average over the dataset[12]), we can derive an approximation of the *(counterfactual) direct effect* of the output of the $h'$-th head at layer $l'$ and token $t$ on the query vector of the $h$-th head at layer $l$ and token $t$:

$$\mathbf{q}_{l,t,h} \approx W^Q_{l,h} \left( \gamma_l \odot \frac{\mathbf{r}_{l,t} - \mu_{l,t}}{\sqrt{\widehat{\sigma}^2_l + \epsilon}} + \beta_l \right)$$

where $\gamma_l, \beta_l$ are the learned scale and shift parameters of the LN operation, $\mu_{l,t}$ is the average of the vector $\mathbf{r}_{l,t}$ over its coordinates, and $\widehat{\sigma}_l$ is an average over the dataset of the standard deviation of the residual stream at this position. Alternatively, we can use the exact layernorm scale from the forward pass over a large sample to compute the statistics of the exact direct effect over observed data.

With either approach, we obtain a decomposition

$$\mathbf{q}_{l,t,h} \approx \sum_{l' < l, h'} \mathbf{u}_{t,(l',h') \to (l,h)} + \bar{r}_{l,t}$$

of direct contributions from the outputs of earlier heads at this position, plus some residual terms $\bar{r}_{l,t}$ (which are the contributions of all previous MLP layers and the input embedding to the query vector). We can then further decompose $\mathbf{u}_{t,(l',h') \to (l,h)}$ by replacing it with its reconstruction from our feature dictionary.

For either way to treat the layer normalization, we can use the learned feature dictionaries for the outputs, keys, queries and values of attention heads in a number of ways to decompose the direct effect further. Here, we consider **head-and-feature attribution**: fixing the head $(l, h)$, we can vary the head $(l', h')$ and break down the direct effects (projected on the query vector) by feature.

**Results.** An interesting location to examine is the attention of the name mover heads from END to the IO and S1 positions, where (according to the analysis in Wang et al. (2022)) the signal from the S-Inhibition heads effectively removes the **S** name from the attention of these heads.

We show the results for the head L10H0 in Figure 8. Crucially, we observe that most interactions are tightly clustered around zero, which suggests that these feature dictionaries provide a sparse and interpretable account of the attention mechanism. The only significantly nonzero interactions are (1) between the **S** features in the query and the key at the S1 position; (2) between the **Pos** features in the query and the key at both positions; and (3) between the **IO** features in the query and the key at the IO position. The first two interactions are expected given the findings of Wang et al. (2022). More interesting is the third interaction, which is negative, suggesting that the L10H0 head inhibits both the **S** and **IO** name tokens, and effectively relies only on the **Pos** attribute to distinguish between the two names. Notably, this is in contrast with the other two name movers L9H6 and L9H9 (for which analogous plots are shown in Figure 18 and 19), where the inhibition of the **IO** attribute is absent. We found that using other methods to compute feature dictionaries results in less sparse and interpretable patterns.

We further investigated the queries of the L10H0 head, by looking at which features from upstream head outputs at the END token have a large direct effect on these queries. Following the methodology detailed in Appendix 7.9, we plot the direct effect from the outputs of the S-Inhibition heads, as well as the two name mover heads L9H6 and L9H9 in layer 9 in Figure 9. We find that the S-Inhibition heads' **IO** features have no significant contribution to the queries, but the **IO** features from the two name mover heads in layer 9 have a significant direct effect (aligned with the overall centered query vector). This suggests that, having already computed a representation of the **IO** attribute, these heads transmit it to the next layer, where it gets picked up by the L10H0 head's query. Conversely, the S-Inhibition heads contribute significantly with their **Pos** and **S** features, whereas the name mover heads in layer 9 do not.

---

[12]This is justified by the empirical observation that the layer normalization scales across the dataset are well concentrated around their mean.

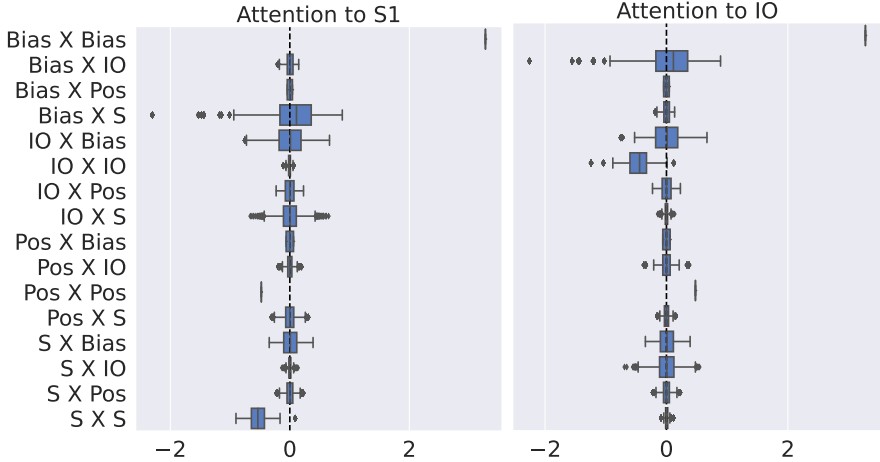

Figure 8: Decomposing the attention scores of the name mover head L10H0 from END to the S1 (left) and IO (right) positions. The y-axis ranges over the combinations of features from the query (first element) and the key (second element). The boxplots show the distribution of dot products between the corresponding feature vectors. The interaction between the bias terms (i.e., the means of the respective queries/keys) provides a sense of the scale of the effects.

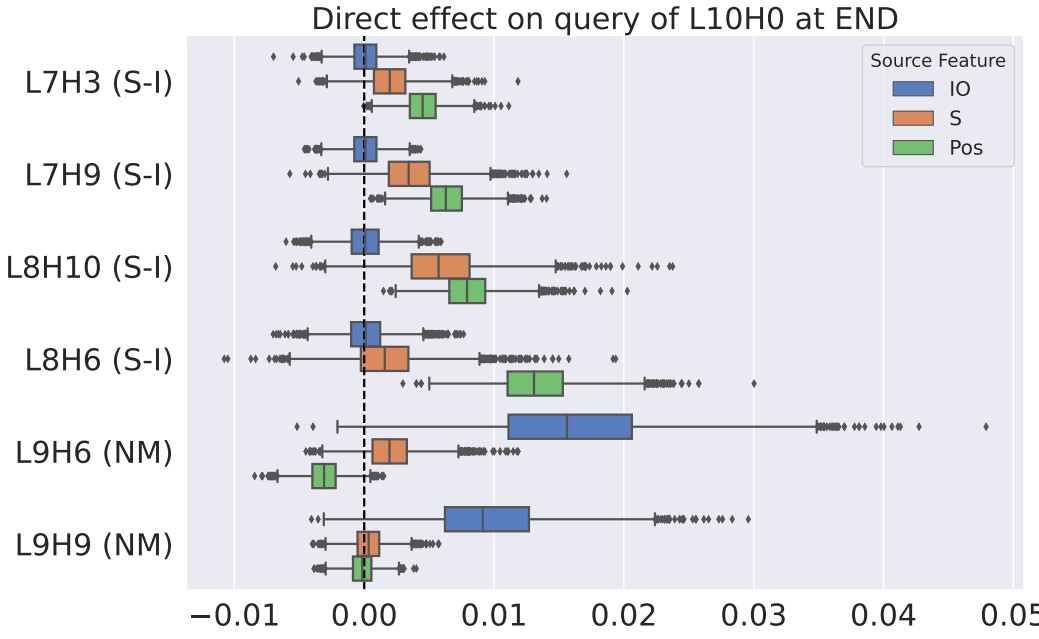

Figure 9: Direct effect of supervised features in the output of S-Inhibition heads, and Name Mover heads in layer 9, on the queries of the L10H0 name mover head at the END token.

## 7.10 ALTERNATIVE PARAMETRIZATIONS FOR THE IOI TASK

We mostly experimented with two possible parametrizations of prompts via attributes:

- **independent parametrization**: we use the three independently varying attributes – **S**, **IO** and **Pos** – to describe each prompt. This is the parametrization used in the main text.

- **coupled parametrization**: we couple position with name, and use the two attributes (**S**, **Pos**) and (**IO**, **Pos**) to describe each prompt. This parametrization is more expressive than

the independent one, as it allows for different features for the same name depending on whether it comes first or second in the sentence. At the same time, the coupled parametrization can express the independent one as a special case (Appendix 7.11).

We find that these parametrizations arrive at highly similar activation reconstructions $\widehat{\mathbf{a}}$. In fact, we find an even stronger property: the coupled parametrization essentially simulates the features in the independent one; details are given in Appendix 7.11.1.

Finally, we note that the fact that we find parametrizations that result in good approximation is not trivial. Not every 'natural-seeming' parametrization will lead to a good approximation of model behavior; we show an example of this with a 'names' parametrization in Appendix Figure 17, where we instead use an attribute for the 1st, 2nd and 3rd name in the sentence.

**What about other parametrizations?**  Note that there are combinatorially many possible ways to pick attributes to disentangle the activations into, and a priori any specific choice is arbitrary. We justify our choice of parametrizations in several ways: (1) they pass our tests for model approximation, control and interpretability given later in this section; (2) they correspond to the internal states of the IOI circuit identified in Wang et al. (2022); (3) we experimented and/or considered other parametrizations, but found they either perform the same or worse on our tests. In Appendix 7.12, we provide a more detailed discussion of different possible parametrizations in the IOI task and their relative strengths and weaknesses.

### 7.11    COMPARING THE COUPLED AND INDEPENDENT PARAMETRIZATIONS

#### 7.11.1    THE COUPLED PARAMETRIZATION CAPTURES THE INDEPENDENT ONE

**Idealized model.** We first note that the coupled parametrization can express all reconstructions expressible by the independent parametrization. Suppose we have an IOI distribution using a set of available names $S_{names}$, and let $\mathbf{pos}_{ABB}, \mathbf{pos}_{BAB}, \{\mathbf{io}_a\}_{a \in S_{names}}, \{\mathbf{s}_a\}_{a \in S_{names}}$ be feature dictionaries for the independent parametrization at some model activation. Then, we can define the following feature dictionaries for the coupled parametrization:

$$\mathbf{io}_{a,ABB} = \mathbf{io}_a + \frac{1}{2}\mathbf{pos}_{ABB}, \quad \mathbf{io}_{a,BAB} = \mathbf{io}_a + \frac{1}{2}\mathbf{pos}_{BAB},$$

$$\mathbf{s}_{a,ABB} = \mathbf{s}_a + \frac{1}{2}\mathbf{pos}_{ABB}, \quad \mathbf{s}_{a,BAB} = \mathbf{s}_a + \frac{1}{2}\mathbf{pos}_{BAB}$$

Then for a prompt $p$ of the form ABB (the BAB case is analogous), with the **IO** name being $a$ and the **S** name being $b$, we have that the reconstruction of an activation $\mathbf{a}$ using the independent parametrization is

$$\widehat{\mathbf{a}}_{independent} = \mathbf{io}_a + \mathbf{s}_b + \mathbf{pos}_{ABB}$$

and the reconstruction using our coupled parametrization is

$$\widehat{\mathbf{a}}_{coupled} = \mathbf{io}_{a,ABB} + \mathbf{s}_{b,ABB} = \mathbf{io}_a + \mathbf{s}_b + \frac{1}{2}\mathbf{pos}_{ABB} + \frac{1}{2}\mathbf{pos}_{ABB}$$

$$= \mathbf{io}_a + \mathbf{s}_b + \mathbf{pos}_{ABB} = \widehat{\mathbf{a}}_{independent}$$

**Empirical evaluation.** We evaluated whether this occurs empirically with the MSE features for the two parametrizations. First, we plot the fraction of variance explained by the reconstructions using the independent parametrization in the reconstructions using the coupled parametrization. We find very high agreement (Figure 10); results in the other direction are almost identical and are not shown here for brevity. Next, we check if the coupled parametrization essentially simulates the independent one as described analytically above. We do this by measuring the cosine similarity between the vector $\mathbf{pos}_{ABB} - \mathbf{pos}_{BAB}$ from the independent parametrization and vectors $\mathbf{io}_{a,ABB} - \mathbf{io}_{a,BAB}$ and $\mathbf{s}_{a,ABB} - \mathbf{s}_{a,BAB}$ from the coupled parametrization. In our idealized simulation of the independent parametrization using the coupled one, these values would be exactly 1 for all names. We find that in all circuit locations that represent both **IO** and **Pos**, the similarities w.r.t the **IO** differences are significant; similarly, in all circuit locations that represent both **S** and **Pos**, the similarities w.r.t the **S** differences are significant (Figure 11). For reference, in a space of this dimensionality (64), the expected magnitude of the cosine similarity between two random vectors is $1/8$.

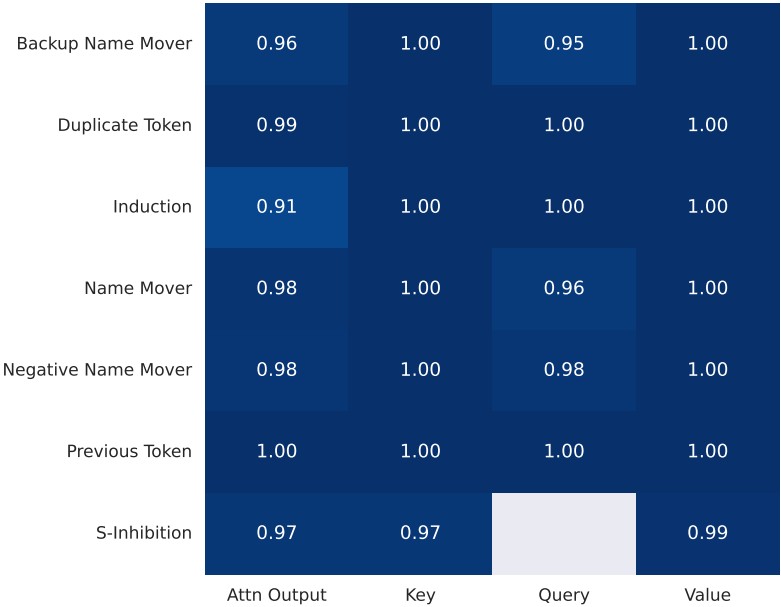

Figure 10: Variance explained by the reconstructions using the independent parametrization, with respect to the reconstructions using the coupled parametrization, averaged over combinations of class of heads in the circuit and activation types.

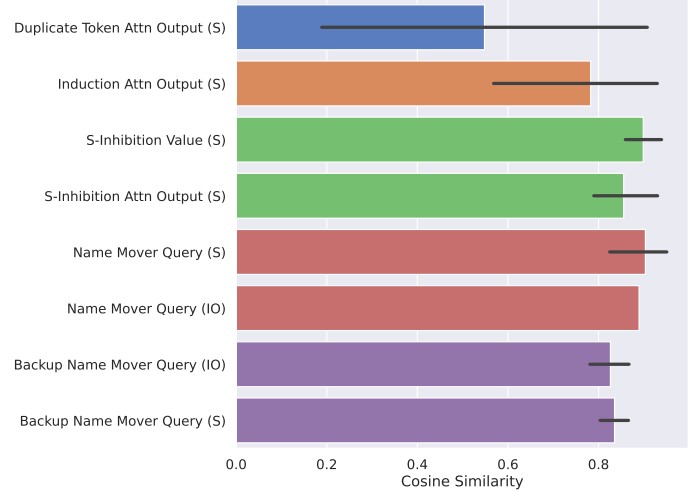

Figure 11: Cosine similarity between the vector $\mathbf{pos}_{ABB} - \mathbf{pos}_{BAB}$ from the independent parametrization and vectors $\mathbf{io}_{a,ABB} - \mathbf{io}_{a,BAB}$ and $\mathbf{s}_{a,ABB} - \mathbf{s}_{a,BAB}$ from the coupled parametrization, averaged over several classes of circuit locations. When evaluating similarity for the $\mathbf{io}_{a,ABB} - \mathbf{io}_{BAB}$ vectors, we only include locations where both the **IO** and **Pos** attributes are represented; and similarly for the **s**-vectors.

### 7.11.2 OTHER PARAMETRIZATIONS EXPRESSIBLE BY THE COUPLED PARAMETRIZATION.

Consider the parametrization with attributes (**S**, **Pos**) and **IO**. Again, let $\{\mathbf{io}_a\}_{a \in S_{names}}$, $\{\mathbf{s}_{a,ABB}\}_{a \in S_{names}}$, $\{\mathbf{s}_{a,BAB}\}_{a \in S_{names}}$ be feature dictionaries for this parametrization at some model activation. Then, we can define the following feature dictionaries for the coupled parametrization:

$$\mathbf{io}'_{a,ABB} = \mathbf{io}'_{a,BAB} = \mathbf{io}_a, \quad \mathbf{s}'_{a,ABB} = \mathbf{s}_{a,ABB}, \quad \mathbf{s}'_{a,BAB} = \mathbf{s}_{a,BAB}$$

Then for a prompt $p$ of the form ABB (the BAB case is analogous), with the **IO** name being $a$ and the **S** name being $b$, we have that the reconstruction of an activation **a** using the (**S**, **Pos**) + **IO** parametrization is

$$\widehat{\mathbf{a}}_{(\mathbf{S},\mathbf{Pos})+\mathbf{IO}} = \mathbf{io}_a + \mathbf{s}_{b,ABB} = \mathbf{io}'_{a,ABB} + \mathbf{s}'_{b,ABB} = \widehat{\mathbf{a}}_{coupled}$$

and so the coupled parametrization can express all reconstructions expressible by the (**S**, **Pos**) + **IO** parametrization. Analogously, it can express all reconstructions expressible by the (**IO**, **Pos**) + **S** parametrization.

### 7.11.3 Editing methodology with the coupled parametrization.

For each activation, we may choose to edit one or several of the **IO**, **S** and **Pos** properties of the prompt. With the independent parametrization, this is straightforward, since the attributes match these properties. With the coupled parametrization, suppose we are given a prompt of the form ABB (the BAB case is analogous) with the **IO** name being $a$ and the **S** name being $b$. Given an activation **a** with corresponding reconstruction under the coupled parametrization

$$\widehat{\mathbf{a}} = \mathbf{io}_{a,ABB} + \mathbf{s}_{b,BAB}$$

we can perform edits as follows:

- to change the **IO** name from $a$ to $a'$: $\mathbf{a}_{edit} = \mathbf{a} - \mathbf{io}_{a,ABB} + \mathbf{io}_{a',ABB}$
- to change the **S** name from $b$ to $b'$: $\mathbf{a}_{edit} = \mathbf{a} - \mathbf{s}_{b,ABB} + \mathbf{s}_{b',ABB}$
- to change the **Pos** property from ABB to BAB: $\mathbf{a}_{edit} = \mathbf{a} - \mathbf{io}_{a,ABB} - \mathbf{s}_{b,ABB} + \mathbf{io}_{a,BAB} + \mathbf{s}_{b,BAB}$

### 7.12 On Possible Feature Dictionaries for the IOI Task

In this section, we compare the properties of several *a priori* possible ways in which the activations of the IOI circuit could be disentangled via sparse feature dictionaries. The main goal is to illustrate that different feature dictionaries can have similar usefulness in terms of model control and interpretability, even if they fail natural tests that directly look for similar features in the two dictionaries. This motivates evaluations that are agnostic to the specific features in a dictionary, as long as the features can parsimoniously disentangle the model's internal computations.

**The independent parametrization from the main text.** It is worth explicitly describing the properties of the supervised feature decomposition we constructed in Section 4.2, which uses the **IO**, **S** and **Pos** attributes to describe the prompts; it serves as an idealized example against which to compare other possible decompositions. In this decomposition, we can approximate an activation **a** for a prompt $p$ where the **IO** name is $a$ and the **S** name is $b$, with the **IO** name appearing first, as follows:

$$\mathbf{a} \approx \mathbf{io}_a + \mathbf{s}_b + \mathbf{pos}_{ABB}$$

where the vector $\mathbf{io}_a$ is the feature for the **IO** name $a$, the vector $\mathbf{s}_b$ is the feature for the **S** name $b$, and the vector $\mathbf{pos}_{ABB}$ is the feature for the **Pos** attribute when the **IO** name appears first in the sentence (and analogously for $\mathbf{pos}_{BAB}$).

Imagine now that we are given an 'unlabeled' feature dictionary that corresponds to this decomposition (i.e., we don't know which attribute each feature corresponds to). We want to evaluate the usefulness of this decomposition for model control and interpretability relative to the 'human-legible' attributes **IO**, **S** and **Pos**. This will be tautologically successful:

- the reconstructions are a faithful and complete representation of the model's internal computations;
- the dictionary can (by definition) express edits to the **IO**, **S** and **Pos** attributes very efficiently, as we only need to change a single feature vector to change the corresponding attribute's value.
- the features are fairly interpretable: we can understand the meaning of each feature in terms of the attribute it represents.

- moreover, using metrics such as recall and precision (following the evaluation methodology of Bricken et al. (2023)) will readily surface the features that are most important for each attribute.

**Using per-gender vectors to describe names.** Another possible decomposition is

$$\mathbf{a} \approx \mathbf{io}_a + \mathbf{io\_gender}_{g(a)} + \mathbf{s}_b + \mathbf{s\_gender}_{g(b)} + \mathbf{pos}_{ABB}$$

where $g : \mathbf{Names} \to \{M, F\}$ is some labeling function that roughly classifies names according to how they are typically gendered[13]. Here, we hypothesize that the model may use features of high norm that sort names into genders (which may be useful to the model for various reasons), and then add a small-norm per-name correction to obtain a name-specific representation. In particular we imagine that $\mathbf{io}_a + \mathbf{io\_gender}_{g(a)}$ in this representation would correspond to $\mathbf{io}_a$ in the supervised decomposition, and similarly for the $\mathbf{S}$ name.

- This decomposition is also fairly sparse and interpretable, and it can express edits almost as parsimoniously as the supervised decomposition (we only need to change *two* feature vectors to edit a name, and one to edit **Pos**).
- Moreover, metrics such as recall and precision will pick up on the per-name features;
- If we compare the prompts for which a feature activates for this dictionary and our 'independent parametrization' feature dictionary (discussed above), we will easily discover the per-name features that correspond to the **IO** and **S** names.
- **However**, if we instead directly use cosine similarity to the supervised features $\mathbf{io}_a, \mathbf{s}_b$ as a metric, we may be misled, because if the per-gender features have sufficiently higher norm than the per-name corrections, the cosine similarity may be low.

**Features for small, overlapping subsets of names.** Going further, we can imagine a decomposition where we have features that correspond to pairs of names, such that each name is in exactly two pairs (this can be achieved by partitioning all names into pairs along a cycle). We can express a name as a sum of the features for the two pairs it is in, with some superposition. Note that more sophisticated constructions with more features per name / more names per feature are possible by e.g. picking subsets at random or using expander graphs (Hoory et al., 2006), as they will 'spread out' the superposition more evenly.

- This decomposition is *somewhat* sparse and interpretable, and can likely be used for feature editing in a reasonable way, as long as the sets of features associated with each name are not too large. Even though we would need to change several feature vectors to edit a name, there should also be a fair amount of disentanglement so that we don't also need to throw away all the features active in an example to change a single attribute.
- **However**, comparing our supervised decomposition against this one using geometric metrics such as cosine similarity may be misleading, because while a sum of a few feature vectors associated with the same name may point in the same direction as our supervised feature, any individual feature may not.
- **Furthermore**, it also has significantly reduced precision for the features, because each of the few features associated with a name will also activate for several other names. This can make directly looking for features whose activation patterns resemble the ones in our supervised decomposition misleading.

Our experiments suggest that both task-specific and full-distribution SAEs trained on IOI circuit activations learn a decomposition resembling this abstract construction more than any other decomposition discussed here.

**Overfitting dictionaries.** Finally, a worst-case decomposition would be to have a single feature for each possible set of values of the **S**, **IO** and **Pos** attributes.

- This decomposition is not interpretable, and it is not editable in any non-trivial way: to change a single attribute, the entire decomposition must be replaced;
- Features of this form will have maximum precision, but very low recall for the attributes.

---

[13]We experimented with this decomposition in our supervised framework, but did not find it to confer additional benefits for the purposes of our tests.

### 7.13 Details for training Sparse Autoencoders

#### 7.13.1 Vanilla Task SAEs

We followed the methodology of Bricken et al. (2023), with the exception that our neuron re-initialization method is not as sophisticated as theirs: we simply re-initialize the encoder bias, encoder weights and decoder weights for the dead neurons every 500 training epochs.

Training SAEs on the IOI distribution alone allows us to do a more extensive hyperparameter search. Importantly, we normalized SAE inputs across attention heads so that activations have an $\ell_2$ norm of 1 on average in order to make it easier for the same set of hyperparameters to work well across different heads. In our main experiments, we use SAEs that were trained with a $16\times$ hidden expansion factor, (effective) $\ell_1$ regularization coefficient in $(0.01, 0.05, 0.1, 0.3)$, batch size of 1024, and learning rate of 0.001.

We evaluated two key test-set metrics across training epochs: the average number of active features per example (i.e. the average $\ell_0$ norm of activations), and the fraction of the logit difference recovered when using the SAE reconstructions at the given model location instead of the original activations, scaled against a mean-ablation baseline (which is more stringent than the zero-ablation baseline employed in most other work). We chose the regularization coefficient and training checkpoint for each node that provided a good trade-off between these two metrics; in particular, for almost all nodes, we recover logit difference to within $20\%$ with respect to the mean-ablation baseline, and there are $< 25$ active features per example. We provide the results of this sweep in Figure 20. While most SAEs seemed adequate, some still have poor approximation as measured by the reconstructed logit difference.

We use a training set of 20,000 examples and an evaluation set of 8,000 examples (for the purposes of automatic interpretability scoring, we need a large enough evaluation sample so that each property in the distribution appears a significant number of times). Since our training regime is significantly distinct from that of Bricken et al. (2023) (we use a much smaller dataset), we first experimented extensively with different hyperparameters, focusing on training SAEs on the queries of the name mover heads. We observed that the most important hyperparameters are the dictionary size and the effective $\ell_1$ regularization coefficient. We found that the batch size did not influence the eventual quality of the learned features, only the speed of convergence, and that a learning rate of $10^{-3}$ (as in Bricken et al. (2023)) was a good choice throughout. The runs reported here used a dictionary size of 1024 (a $16\times$ increase over the dimensionality of attention head activations in GPT-2 Small), an effective $\ell_1$ regularization between 0.05 and 0.3, and a batch size of 1024.

We normalized activations across the circuit to make it easier for the same range of hyperparameters to give good results, and ran a sweep over $\ell_1$ regularization coefficients in $(0.01, 0.05, 0.1, 0.3)$.

**Results.** While performance varied strongly across circuit locations, most full-distribution SAEs had an $\ell_0$-loss between 2 and 12 and a recovered loss fraction (against a mean ablation baseline) between 0.4 and 0.9 (both measured on OPENWEBTEXT). Similarly, most task-SAEs had an $\ell_0$-loss below 25 and a recovered logit difference fraction against mean ablation $> 0.8$ (both measured on the IOI dataset) [14].

#### 7.13.2 Additional Task SAE Variants: Gated and topK

**Training schedule.** We incorporate several training tactics from recent literature. For all SAE variants considered, we used the same (small) learning rate of $3 \times 10^{-4}$, trained for 2000 epochs in total, and applied resampling followed by a learning rate warmup over 100 epochs (roughly following Rajamanoharan et al. (2024)) at epochs 501 and 1001. Our resampling methodology closely follows that of Bricken et al. (2023). In addition, we decay the learning rate linearly to zero over the last $25\%$ of training (following Conerly et al. (2024)). For topK SAEs, we initialize the encoder to the transpose of the decoder, as suggested by Gao et al. (2024); however, we do not use

---

[14]Importantly, we did not perform exhaustive hyperparameter tuning to train these SAEs, as our main goal was to evaluate the methodology and how it can distinguish between different classes of feature dictionaries, rather than to achieve state-of-the-art performance. Thus it is possible that significantly better performance could be achieved with more tuning. Indeed, it is our hope that the methods we present here will be useful for tuning SAEs in the future.

the auxiliary loss term suggested in section 2.4 ('Preventing dead latents') from that work (which is the only difference of this paper from the implementation of Gao et al. (2024)).

We checkpoint all models at 14 epochs: $(1, 2, 4, 8, 16, 32, 64, 128, 500, 750, 1000, 1250, 1500, 2000)$. This checkpoint schedule is chosen to ensure that we have a dense enough sampling of the early stages of training, while also capturing the state of the model right before resampling, and after the learning rate warmup that is done post-resampling is sufficiently in the past.

**Preprocessing.** We normalize all IOI circuit activations prior to passing them through our SAEs, following the scaling methodology in Conerly et al. (2024), so that they on average have $\ell_2$ norm of $\sqrt{d_{head}}$. This helps us share hyperparameters across sites of the circuit, and reduces the range of hyperparameters to search over.

**Hyperparameters.** We sweep over values $\lambda \in (0.5, 1.0, 2.5, 5.0)$ for the $\ell_1$ regularization penalty for vanilla and gated SAEs, and over values $k \in (3, 6, 12, 24)$ for topK SAEs. This is a reasonable range: we found that the highest value of $\lambda$ leads to about 3-4 active features per example on average; the supervised dictionaries have 3. Conversely, the lowest value of $\lambda$ lead to very good $\ell_2$ loss and recover close to $100\%$ of the logit difference, but have too many (about 20-30) active features per example on average. The range for $k$ is chosen to include values equal and close to our expectation for the 'true' number of necessary features (3), while still allowing significantly more features to be active.

### 7.13.3 FULL-DISTRIBUTION SAEs

We trained full-distribution SAEs on every IOI component using OPENWEBTEXT as training data. We mostly followed the method outlined in Bricken et al. (2023). We added a standardization procedure to be able to train SAEs on components with different activation scales using the same l1-coefficient. Before training, we calculated the mean and the mean l2-norm over 10 million activations. These values were then frozen and used to standardize all activations as a preprocessing step and to rescale the SAE reconstructions to match the original scale as a post-processing step. We generated the training dataset by extracting a buffer of 10 million activation vectors from the shuffled OPENWEBTEXT dataset at a time with a maximal context window of 512 tokens. We then trained the SAEs for 250 million activation vectors and resampled dead neurons after 50000 steps (around 100 million activations) as outlined in Bricken et al. (2023). We used a batch size of 2048 and 8192 features per SAE. Post-training, we excluded dead and ultra-low frequency neurons that we define as neurons who activate less than once per million activations. The amount of dead neurons varies across SAEs between 20 and 90%. We plot the fraction of dead neurons versus $\ell_0$ loss in Figure 22, and the loss recovered versus $\ell_0$ loss in Figure 23.

We used an $\ell_1$ coefficient of 0.006 initially for all SAEs, and retrained SAEs with a different $\ell_1$ coefficient for cross-sections whose SAE metrics were undesired (e.g. very low $\ell_0$ / bad reconstruction or very high $\ell_0$). For the name mover outputs, we used 0.025, and for S-Inhibition keys we used 0.005. The test $\ell_0$ and loss-recovered metrics were calculated on 81920 unseen activation vectors.

We trained fewer SAEs on the full pre-training distribution compared to the IOI distribution, as the computational cost is higher. We observe that SAEs with a lower number of active features per example generally perform better for IOI-related tests, even if their other metrics (such as loss recovered on OPENWEBTEXT) are worse.

**SAE training loss metrics.** The most important loss metrics to track during SAE training are the $\ell_0$ loss (measuring the average number of active features per activation) and the language model loss recovered when using the learned features to reconstruct the model's logits (Bricken et al., 2023). To turn the loss recovered into a meaningful quantity, it is rescaled against a baseline; both zero ablation and mean ablation have been used for this purpose in the literature (Bricken et al., 2023; Kissane et al., 2024). In this work, we used mean ablation, as it is a more strict test of the quality of approximation.

### 7.14 ADDITIONAL NOTES ON METHODOLOGY FOR SAE INTERPRETABILITY

**Interpretations considered.** Let **Names** be the set of names in our IOI dataset. We consider the following binary predicates over prompts as possible interpretations for SAE features in the activations at the S2 and END tokens:

- **IO is 2nd name**: the **Pos** attribute having value corresponding to BAB-type prompts;
- **IO is 1st name**: the **Pos** attribute having value corresponding to ABB-type prompts;
- **S is <name>**: the **S** attribute has a certain value in **Names**;
- **S is <name> and at 1st position**: same as above, but also the **S** name is at 1st position in the prompt (i.e., this is a BAB-type prompt);
- **S is <name> and at 2nd position**: same as above, but for ABB-type prompts;
- **IO is <name>**: the **IO** attribute has a certain value in **Names**;
- **IO is <name> and at 1st position**: same as above, but also the **IO** name is at 1st position in the prompt (i.e., this is a ABB-type prompt);
- **IO is <name> and at 2nd position**: same as above, but for BAB-type prompts;
- **S is male**: the **S** name is labeled as a male name under our labeling of **Names** provided by GPT-4;
- **S is female**: same as above for female names;
- **<name> is in sentence**: a certain name in **Names**;
- **<name> is at 1st position**: same as above, but the name is the first name in the sentence;
- **<name> is at 2nd position**: same as above, but the name is the second name in the sentence;

The next several interpretations are only defined for the keys and values of the name mover heads. We collect together activations for the keys according to their role in the IOI circuit as opposed to absolute position: we group together all activations at the IO token position (these are the **IO** keys/values), even though they come from different absolute positions across IOI prompts, because in ABB prompts the **IO** name comes first, while in BAB prompts it comes second. The same applies for gathering the **S** keys/values.

The key/value activations described above have not yet 'seen' the repeated name in the sentence, so there is no meaningful concept of **IO** and **S** for them. Instead, the only potentially task-relevant information contained in them is about the name(s) seen so far in the sentence, and the position (1st name or 2nd name) where the activation is taken from. Accordingly, the applicable interpretations for features contained in these activations are different:

- **current token is <name>**: the token from which the activations are taken holds a certain value in **Names**.
- **token is <name> and at 1st position**: the activation was taken from a token with a certain value in **Names**, and in addition it comes from the first name in the sentence;
- **token is <name> and at 2nd position**: same as above, but activation is from the second name in the sentence;
- **current token is at 1st position**: the activation is from the first name in the sentence;
- **current token is at 2nd position**: same as above, but from second name;
- **current token is female**: the token the activation is from is female under our labeling of **Names**.

*Unions of interpretations*. In addition, for each type of predicate that has a free parameter in **Names**, we considered unions of up to 10 such predicates (recall that we have a total of 216 names in our dataset). We ordered the individual predicates according to their $F_1$ score, and chose the union of the first $k \leq 10$ predicates with the highest $F_1$ score as a possible interpretation. Note that the $F_1$ score is not in general a monotone function of $k$ in this setup; indeed, we find that for many features the highest-$F_1$-score explanation uses $k < 10$ features.

**Sufficiency/necessity of interpretable features.** We take the interpretations of the features described above and their respective $F_1$ scores, and for each threshold $t \in [0, 1]$ over $F_1$ scores consider two interventions:

- **sufficiency**: we subtract from the respective activation all active features with $F_1$ score $< t$;
- **necessity**: we subtract from the respective activation all active features with $F_1$ score $\geq t$;

## 7.15 ADDITIONAL OBSERVATIONS ON SAE LATENT INTERPRETATIONS

*Correlational evaluation.* Full-distribution SAEs must capture variation in activations across a large set of text, of which IOI-like prompts are only a small subset. Consistent with this, we found that only a subset of full-distribution SAE latents activates on IOI prompts, with the number of latents that fire on IOI prompts varying strongly between components. We scored the latents that do fire on IOI prompts and found a significant amount of latent descriptions with high $F_1$-score. We summarize the number of high-$F_1$-score latents per type in Figure 15. Remarkably, we find that the interpretable latents in the full-distribution SAEs and the task-specific SAEs are qualitatively similar; corresponding task-specific results are given in Appendix Figures 29 (showing the most interpretable SAEs at each node of the IOI circuit) and 30 (showing the SAEs chosen to optimize the tradeoff between the $\ell_0$ loss and the logit difference reconstruction, which we use throughout the main body of the paper).

In practice, we want to use latent explanations to get insight into the more general computation of a component. Thus, we investigated whether the latents found are consistent with the previously established function of the heads from Wang et al. (2022). We found that this was true for all heads and that simply looking at the number of latents with a given interpretation draws a clear picture; examples are provided in Appendix 7.15.

Our results also suggest several details about the IOI circuit that weren't reported previously, which we summarize in Appendix 7.15. We were also curious about how the detected latents behave on arbitrary text of the model's training distribution. As creating a rigorous test for this is difficult, we report some anecdotal evidence in support of latent generalization in Appendix 7.15.

*Causal evaluation: sufficiency/necessity of interpretable latents.* Results here are encouraging: keeping/removing the latents with $F_1$ score $\geq 0.6$ often goes a long way towards preserving/degrading the model's performance on the task. Appendix Figures 33 and 34 show the results of these experiments for task SAEs.

*Causal evaluation: sparse control via interpretable latents.* Here, results are also moderately encouraging. We find that, for the task-specific SAEs, editing using the high-$F_1$-score latents w.r.t. a given attribute as a guide performs not much worse than the interpretation-agnostic editing method. Results are provided in Appendix Figure 31. However, for full-distribution SAEs, we again need to edit a high number of latents to achieve a noticeable effect (results in Appendix Figure 32).

**Interpretable latents agree with head roles identified in the IOI circuit by Wang et al. (2022).** For example, duplicate token heads attend to a previous occurrence of the previous token and write information about this to the residual stream. Consistent with this, we found that SAEs trained on them contain latents that indicate the duplicated name, the subject in case of IOI. This information is then used by the induction heads that determine whether the subject is the first or second name. The subject position is subsequently copied to the END position by the S-Inhibition heads, where it will query the name movers to *not* attend to the subject, and to copy and predict the IO name. Indeed, we find latents of the outputs of induction heads, in the outputs and values of the S-I heads, and in the queries of the name movers that inform about the position of the indirect object. Lastly, the name movers attend to the IO position and copy the name to predict it. As anticipated, the name mover values and outputs contain latents that specify the concrete IO name. In summary, the type of latents detected informs well about the function of the head on a certain task.

**New insights from the latent dictionaries.**

- The first layer DT-heads almost exclusively contain **S** latents but the third layer duplicate token head also has positional latents, suggesting more sophisticated text processing happening there.
- Induction heads encompass different positional latents. L6H9 latents inform about what name is at the second position, while L5H5 and L5H8 activate when the **IO** is at the first position. L5H9 is comprised of different latents including positional latents that don't inform about the role (**IO** vs **S**) of the name.
- L7H3 is the only head that contains a significant amount of gender latents.
- S-Inhibition heads include primarily latents that are a combination of names and **S**, while the name mover queries seem to only contain **Pos** latents.

- The keys and values of name movers both inform about the token at the current position, but there is an important difference in the type of latents: while the values primarily contain latents that contain the name (that later gets moved to the END position), the keys consist of positional latents and combinations of position and name. This hints at an important mechanism where the name mover query contains positional information about the **S** name that gets matched with the corresponding key, effectively shifting the attention towards the IO token such that the value with the **IO** gets copied to the END position.

**Editing interpretable latents**

While the latent descriptions generated through our automatic scoring predict well when a latent is active, it is still unclear whether they also have an interpretable causal role, i.e. whether activating or deactivating a certain latent leads to a change in output logits that would be expected from the latent's description. To test this, we propose two experiments to judge the faithfulness and completeness of our interpreted latents that involve patching activations from a counterfactual prompt and calculating the effect on the model's output:

- **Estimating Faithfulness:** To estimate faithfulness, we construct SAE activations where we fix latents with a test F1-score smaller than a threshold and patch activations of latents with a high F1-score from the counterfactual prompt. We then calculate reconstructions of this new SAE activation vector, patch cross-sections, and record whether the model successfully predicts the correct counterfactual name.
- **Estimating Completeness:** To estimate completeness, we propose a similar experiment where we fix latents with a high test F1-score and patch all remaining latents. This intervention should not change the output logits if our latents are complete.

We run this experiment on cross-sections of name mover outputs and repeat this experiment for different thresholds. We observe that for a threshold F1-score of 0.6, the SAE latents are both faithful and complete to a high degree. We observe that the faithfulness metric significantly decreases for higher F1-scores of 0.7 and 0.8 but remarkably, we also observe that only fixing latents with a very high F1-score of $> 0.8$ while patching all other latents from the counterfactual prompt is sufficient to keep the model predicting the base prompt's output.

**Generalization of latents to the full distribution.** We sample prompts from OpenWebText and visualize prompts that highly activate a latent. We do this for the name mover queries and outputs. For the outputs, we calculate the direct-latent-attribution (DFA) metric first proposed by Krzyzanowski et al. (2024) that calculates per position how large the contribution of its values to activating the latent is. Thus, it informs what position led to the latent being activated.

We find that latents mostly generalize. For example, we investigated a latent that activated if the IO name starts with the letter "E" and we found that on full distribution, this latent fires at tokens preceding words starting with "E". DFA suggests that previous tokens starting with "E" activate this latent, and calculating the unembed $W_{dec}[j]W_O W_U$, denoted with "Positive Logits" and "Negative Logits" in Figure 14 shows that activating the latent increases logits for words starting with "E". Together, this hints at a general name moving mechanisms to predict words that previously occurred in the context that drives in-context learning where the head's QK-circuit drives attention to the position of the word to predict, and the OV circuit copies the token from the previous to the current position to predict it.

### 7.16 FEATURE OCCLUSION DETAILS

**Training an exhaustive set of SAEs.** Focusing on the L10H0 queries, we found that our SAEs consistently find a single feature for almost all **IO** names, but fail to find a significant number of features for individual **S** names.

To investigate further whether this is a result of poor hyperparameter choices, we trained SAEs on the queries of L10H0 over a wide grid of hyperparameters, so that performance deteriorates/plateaus at the edges of the grid. We pushed the dictionary size, training dataset size, $\ell_1$ regularization coefficient and number of training epochs significantly beyond the values used in our other experiments. Specifically, we used a training set of 100,000 examples (this is more datapoints than all $\sim 93,000$ possible combinations of **S**, **IO** and **Pos** in our data); we trained dictionaries of up to $128*64 = 8192$

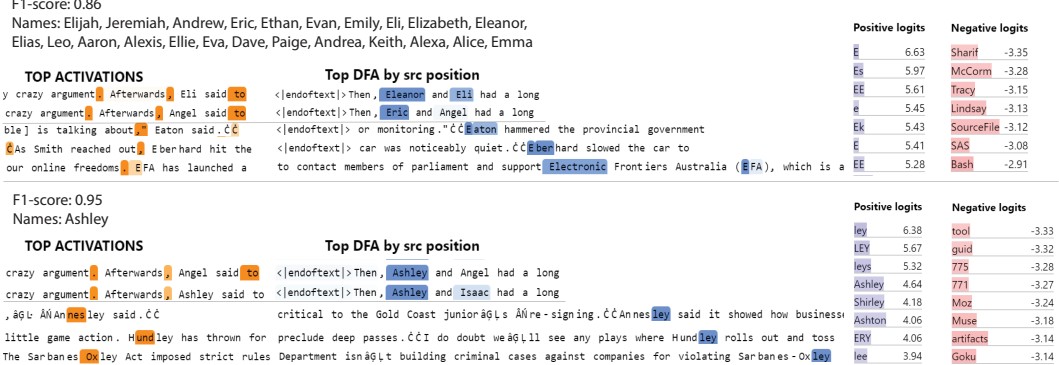

Figure 12: Two representative features discovered in the output name mover SAE L9H9 to illustrate the features behavior or webtext. Both features are IO name features, the upper one containing 23 names, the lower one only a single name. Left: Feature activation per position; Middle: Direct Feature Attribution (DFA) that tracks the position whose values contribute most to activating the given feature; Right: The output tokens whose logits get increased (positive) or decreased (negative) when the feature is active, calculated by $\mathbf{v}\mathbf{W}_O\mathbf{W}_U$ with $\mathbf{v}$ being the decoder weight vector of the feature of interest, $\mathbf{W}_O$ the output weight of the head and $\mathbf{W}_U$ the unembed

features (our supervised feature dictionaries contain $\sim 500$ features); we varied the effective $\ell_1$ regularization coefficient across two orders of magnitude; and we trained for up to $6,000$ epochs.

Results on the number of **IO/S** features with $F_1$ score $> 0.5$ are reported in Figure 7 (left). We observe that across hyperparameters, we often find as many **IO** features as there are names in our dataset; however, the number of **S** features is consistently low, never exceeding 22.

**Magnitudes of the IO and S features.** As a first step, we investigate the distribution of the norms of the features for **IO** and **S** across the names in the IOI dataset in the L10H0 queries in our supervised feature dictionaries from Section 6; results are shown in Figure 13 (left). We observe that the norms of the **IO** features are significantly (but not by much) higher than those of the **S** features.

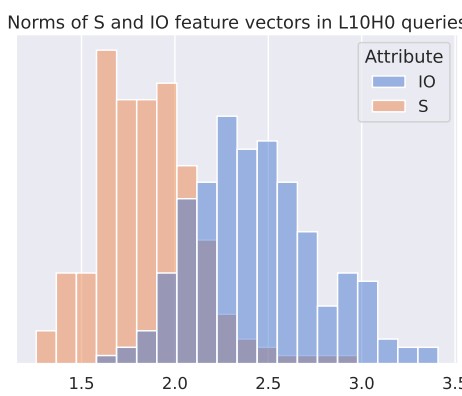
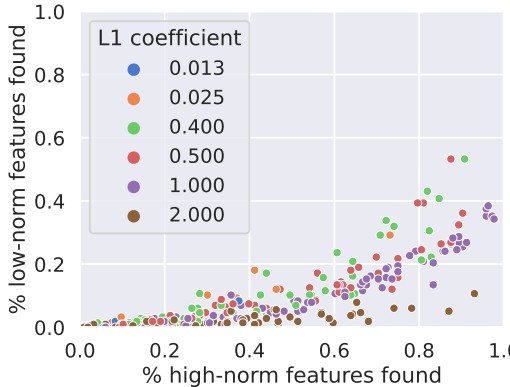

Figure 13: **Left**: distributions of the $\ell_2$ norms of the feature vectors for the **IO** and **S** attributes from our supervised feature dictionary for the queries of the L10H0 name mover. **Right**: the results of the toy model experiment, where we investigate whether a disparity in feature magnitudes alone can lead to the occlusion phenomenon. The $x$-axis shows the fraction of high-magnitude ground-truth features for which we find an SAE feature with $F_1$ score $> 0.9$; the $y$-axis shows the same for the low-magnitude ground-truth features.

**Surgically reducing the magnitude of IO features.** To examine the role of feature magnitude in the occlusion phenomenon, we continuously reduce the magnitude of **IO** features. Namely, given our supervised feature dictionary with features $\{\mathbf{io}_a\}_{a\in\text{Names}}$, $\{\mathbf{s}_b\}_{b\in\text{Names}}$, $\{\mathbf{pos}_v\}_{v\in\{\text{ABB,BAB}\}}$ and

an activation $\mathbf{a}(p)$ for a prompt $p$ where the **IO** name is $a$, we construct a new activation

$$\mathbf{a}_\alpha (p) = \mathbf{a}(p) - \alpha \mathbf{io}_a$$

for $\alpha \in [0, 1]$. We find that, applying this intervention without any hyperparameter tuning (with a modest dictionary size of 1024, and $\ell_1$ regularization coefficient of 0.2), increasing $\alpha$ from 0 to 1 gradually makes the number of **IO** features with $F_1$ score $> 0.5$ to decrease, while the number of **S** features with $F_1$ score $> 0.5$ increases; results are shown in Figure 7 (right).

**Reproducing the occlusion phenomenon in a toy model.** Finally, we wanted to know if a disparity in feature magnitudes alone could lead to the occlusion phenomenon. We constructed a simple toy model closely based on the empirical setup in the queries of the L10H0 head to test this hypothesis.

We form synthetic activations $\mathbf{a} = \mathbf{u}_i + \mathbf{v}_j$ for random pairs $i, j \in \{1, \ldots, |\text{Names}|\}$, where $\mathbf{u}_i, \mathbf{v}_j \in \mathbb{R}^{d_{head}}$ are sampled independently from a standard normal distribution centered at zero, and then $\mathbf{u}_i$ are rescaled so that their mean norm matches the mean norm of **IO** features in the L10H0 queries, and similarly for $\mathbf{v}_j$ and **S** features. We train SAEs on these activations over a wide grid of hyperparmeters: dictionary sizes in $(512, 1024, 2048)$, $\ell_1$ regularization in $(0.0125, 0.025, 0.4, 0.5, 1.0, 2.0)$, batch size in $(256, 1024)$ and learning rate in $(0.001, 0.003, 0.0003)$. We trained for 1000 epochs, saving checkpoints in a geometric progression of epochs. Results for the number of high- and low-magnitude features with $F_1$ score $\geq 0.9$ discovered are shown in Figure 13 (right); we observe that we easily find one SAE feature per each high-magnitude ground-truth feature, but it is more difficult to find an SAE feature for each low-magnitude feature.

However, we note that with lower $F_1$ thresholds, this effect is less pronounced and eventually disappears.

### 7.17  FEATURE OVER-SPLITTING IN A MIXTURE OF GAUSSIANS TOY MODEL

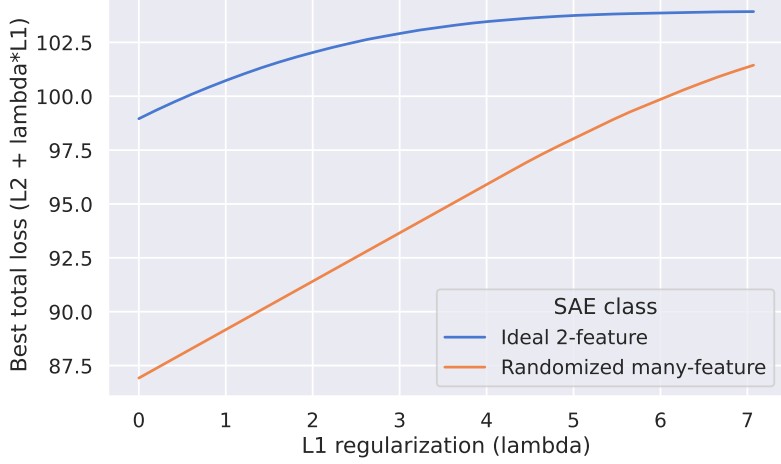

Figure 14: Main experiment for our toy model of feature oversplitting. The data distribution is a uniform mixture of two standard multivariate Gaussian random variables in $100$ dimensions. **Blue**: the (approximate) best possible total loss (in the infinite data limit) achieved by a class of 'ideal' SAEs that use two features pointing towards the means of the two components of the mixture. **Orange**: an approximate upper bound on the best possible total loss achieved by an SAE with $m = 1,000$ hidden features (in the infinite data limit). The $x$-axis is the $\ell_1$ regularization coefficient $\lambda$. The cutoff on the $x$-axis is chosen so that the idealized solution activates only for a vanishing fraction ($< 2\%$) of the examples in the mixture.

Our goal in this section is to demonstrate that there *exist* setups where an SAE with a large number of hidden features $m \gg 2$, when trained on a uniform mixture of two isotropic gaussian variables, will prefer a solution with $\gg 2$ features (as opposed to the 'ideal' solution with only two features, one per component of the mixture), for *any* value of the $\ell_1$ regularization coefficient $\lambda$ and *any* amount of training data.

**Setup.** We consider a simple toy model where activations are distributed according to a uniform mixture $\mathcal{D}_{toy}$ of two isotropic gaussians $\pm\boldsymbol{\mu} + \mathcal{N}(0, \mathbf{I}_d)$ in $\mathbb{R}^d$ (i.e., we first flip a fair coin to determine the sign of $\boldsymbol{\mu}$, and then sample an extra additive term from $\mathcal{N}(0, \mathbf{I}_d)$). We sample an i.i.d. dataset from this mixture. We used $d = 100$, $\|\boldsymbol{\mu}\|_2 = 2$ in the experiments below; this choice guarantees that the two components of the mixture are separable with high ($> 95\%$) probability.

**The 'ideal' solution with 2 features.** We might hope that, with the right $\ell_1$ regularization, an SAE trained on such a distribution will discover only two interpretable features, one for each component in the mixture, with the encoder/decoder vectors aligned with $\boldsymbol{\mu}$ and $-\boldsymbol{\mu}$ respectively, and each feature activating for (approximately) the examples in its corresponding component.

We find this is the case empirically when we train an SAE with only two hidden features on this toy distribution. Specifically, there is a range of $\lambda$ values $1 \le \lambda \le 3$ where the SAE reliably approximately recovers this ideal solution, with the encoder bias controlling the trade-off between the two loss terms: when $\lambda$ increases, the encoder bias changes so that fewer examples in each component of the mixture are activated (and the active examples have a lower $\ell_1$ loss). Beyond this range, the SAE often fails to activate any feature on any examples ($\lambda > 3 + \varepsilon$), or activates both features on almost all examples ($\lambda < 1 - \varepsilon$).

**Analyzing the ideal solution across $\ell_1$ coefficients.** To study the properties of the 'ideal' solution analytically, we make the following assumptions (borne out empirically with 2-feature SAEs) using only symmetries of the data distribution:

- the decoder vectors are $\pm\boldsymbol{\mu}$, normalized to have unit $\ell_2$ norm (by symmetry of each component around its mean);
- the respective encoder vectors are $\pm k\boldsymbol{\mu}$ for some $k > 0$ (again by symmetry of each component around its mean);
- the decoder bias is zero (by symmetry of the mixture around zero);
- both encoder biases are set to $-\gamma$ for some $\gamma > 0$ (again by symmetry of the mixture around zero).

This leaves only two parameters to tune: the encoder bias $\gamma$ and the encoder scale $k$. We can thus use the following strategy to analytically approximate the best loss of this class of solutions for a given $\lambda$:

- approximate the expected $\ell_1$ and $\ell_2$ losses over a fine grid of values for $\gamma$ and $k$, for a large dataset of samples from the mixture;
- given a $\lambda$ value, find the point in the grid that minimizes the total loss $\ell_2 + \lambda\ell_1$.

We implemented this using $10^5$ samples, with a grid of 100 values for $\gamma$ in $[0, 5]$ and 20 values for $k$ in $[0, 2]$, over a grid of 100 values of $\lambda$ in $[0, 20]$. We verify that the best values chosen for each $\lambda$ are not on the edges of the grid; the resulting curve of best total loss values versus $\lambda$ is shown in Figure 14 (blue), cut off at $\lambda \approx 7$, beyond which the selected SAE activates for $< 2\%$ of examples in the components of the mixture.

**SAEs with $m \gg 2$ features prefer other solutions even with infinite data.** Next, we want to show that with enough features, SAEs will prefer solutions different from the class of 2-feature solutions described above. How can we give an *empirical* argument that applies to *any* amount of training data and any $\lambda$? After all, a *trained* SAE is a function of the data it is trained on, so no experiment on datasets of bounded size can establish properties of SAEs trained on arbitrarily large datasets.

We get around this by defining a class of SAEs that is competitive with the class of ideal 2-feature SAEs *upfront*, independent of the training sample, by using a randomized construction that works w.h.p., and then estimating the expected loss of this SAE in the infinite data limit empirically. To give evidence of our result for arbitrary $\lambda$, we consider a fine enough grid of $\lambda$ values, and for each $\lambda$ we construct an SAE that is competitive with the best ideal 2-feature SAE.

Our randomized SAE construction proceeds as follows:

- Sample $m$ encoder vectors $W_{enc} \in \mathbb{R}^{m \times d}$ from $\beta * \mathcal{D}_{toy}$ (i.e. a version of $\mathcal{D}_{toy}$ scaled by $\beta$) where $\beta > 0$ is a hyperparameter that we will tune;

- Set the decoder vectors $W_{dec} \in \mathbb{R}^{d \times m}$ to be the same as the encoder vectors $W_{enc}^{\top}$, but normalized so that each column of $W_{dec}$ has unit $\ell_2$ norm;

- Set all encoder biases to $-\gamma$ (where $\gamma > 0$ is a hyperparameter that we will tune), and the decoder bias to $\mathbf{0} \in \mathbb{R}^d$.

We used $m = 1,000$, a grid of 100 values for $\beta$ in $[0, 0.1]$ to search for the best $\beta$ for each $\lambda$, and fixed $\gamma \approx 2.35$.

**Empirical confirmation.** Finally, we actually trained SAEs with many hidden features on $\mathcal{D}_{toy}$, and observed that these SAEs reliably learned solutions with many active features across $\lambda$ values.

## 7.18   ADDITIONAL DETAILS FOR SECTION 5

**Feature weights are mostly in the interval** $[0, 1]$**.** Recall that given a reconstruction $\widehat{\mathbf{a}} = \sum_i \mathbf{u}_i$, we defined the feature weight for the $i$-th feature as

$$\text{weight}\,(i) = \mathbf{u}_i^{\top} \widehat{\mathbf{a}} / \|\widehat{\mathbf{a}}\|_2^2\,.$$

For our supervised feature dictionaries, we find that $10\%$ of all weights are negative, and that the average value of all negative weights across all nodes in the IOI circuit and all three attributes is $-0.037$. Similarly, for our task-specific SAE feature dictionaries, even though $31\%$ of all weights are negative, the average value of all negative weights is $-0.002$. The number and magnitude of weights higher than 1 are even smaller.

**Causal evaluation using interpretability.** While the feature descriptions generated through our automatic scoring predict well when a feature is active, it is still unclear whether they also have an interpretable causal role, i.e. whether activating or deactivating a certain feature leads to a change in output logits that would be expected from the feature's description. To test this, we propose two experiments to judge the sufficiency and necessity of our interpreted features that involve patching activations from a counterfactual prompt and calculating the effect on the model's output:

- **Estimating sufficiency:** To estimate sufficiency, we construct SAE activations where we fix features with a test F1-score smaller than a threshold and patch activations of features with a high F1-score from the counterfactual prompt. We then calculate reconstructions of this new SAE activation vector, patch cross-sections, and record whether the model successfully predicts the correct counterfactual name.

- **Estimating necessity:** To estimate necessity, we propose a similar experiment where we fix features with a high test F1-score and patch all remaining features. This intervention should not change the output logits if our features are complete.

We run this experiment on cross-sections of name mover outputs and repeat this experiment for different thresholds. We observe that for a threshold F1-score of 0.6, the SAE features are both faithful and complete to a high degree. We observe that the faithfulness metric significantly decreases for higher F1-scores of 0.7 and 0.8 but remarkably, we also observe that only fixing features with a very high F1-score of ¿0.8 while patching all other features from the counterfactual prompt is sufficient to keep the model predicting the base prompt's output.

## 7.19   ADDITIONAL FIGURES

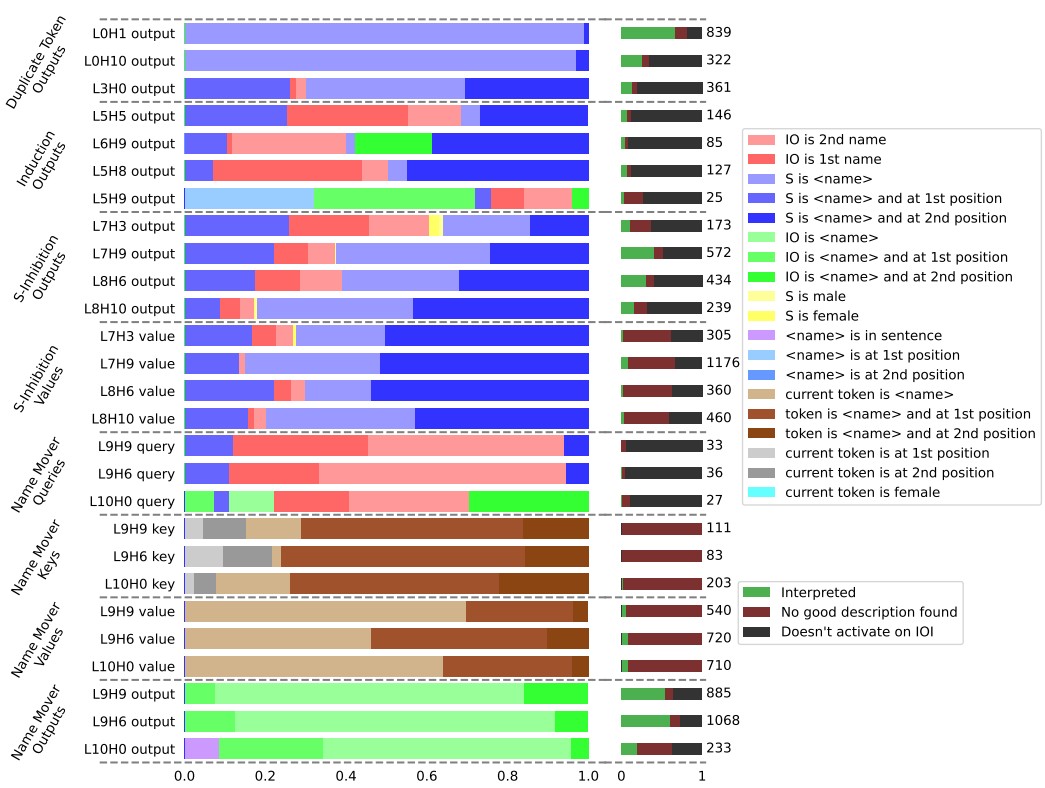

Figure 15: Interpreting the IOI features learned by SAEs trained on OPENWEBTEXT. For each node in the IOI circuit, we show the distribution of interpretations for the features which have any interpretation with $F_1$ score above a threshold. The numbers in the right column indicate the number of features with an assigned interpretation by our method, and the color bars show the overall distribution of the SAE features (conditioned on the feature not being dead on the SAE training distribution).

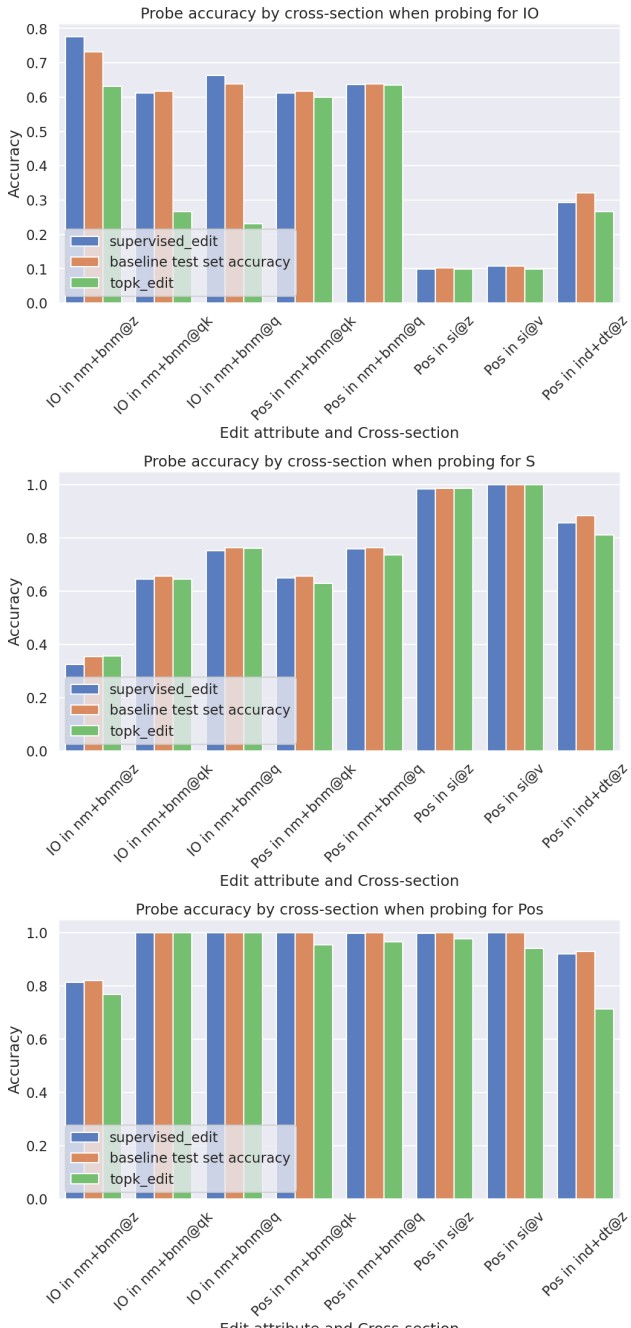

Figure 16: Accuracy for probes for the **IO**, **S** and **Pos** attributes in nodes of the IOI circuit, as we apply our feature editing methods via either supervised (blue) or task-specific topK SAEs (green). For each cross-section, we train a probe on each location in this cross-section, and report average accuracies. The 'baseline test set accuracy' is the accuracy of each probe on a test set of activations w.r.t. the one used to train the probe. We always evaluate probe accuracy w.r.t. the attribute value we expect to observe in an edit. This means that, for example, if we are editing the **IO** attribute, we evaluate the **IO** probes against the name we are editing to, but the **S** and **Pos** probes against the original values of these attributes.

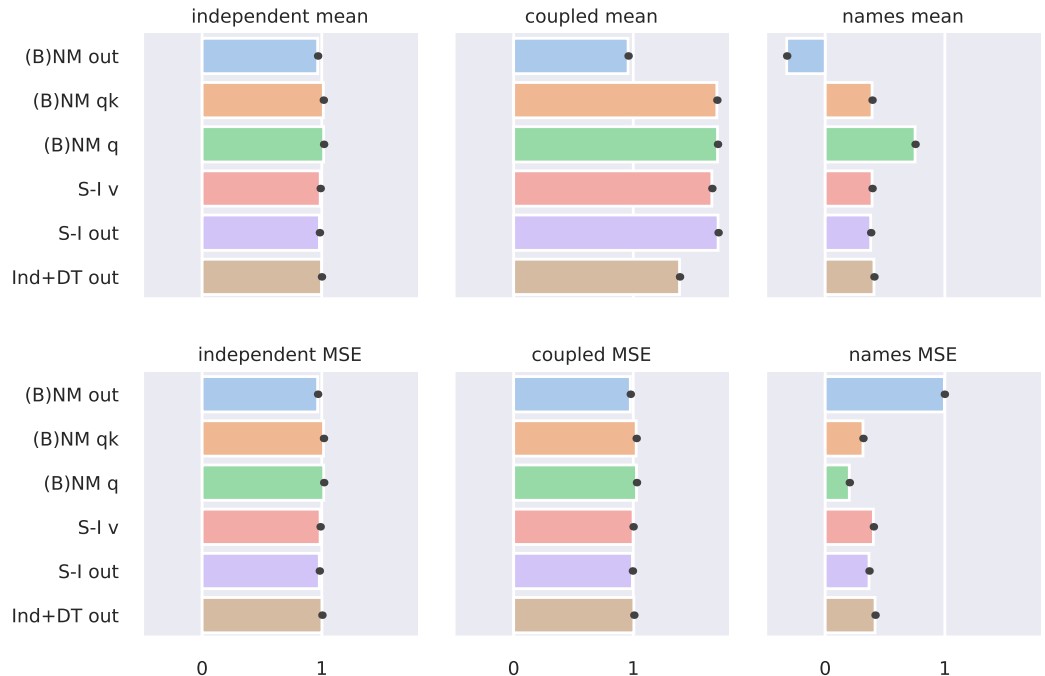

Figure 17: Fraction of recovered logit difference for several different methods to compute feature dictionaries, across cross-sections of the circuit. For a definition of the 'names' parametrization, see Appendix 7.12.

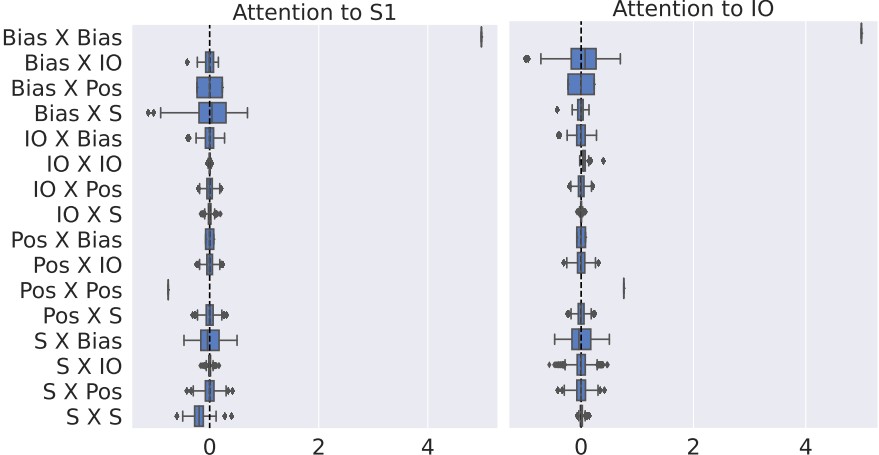

Figure 18: Attention score decomposition for the L9H6 name mover (see Figure 8 for explanation). Notice that, in contrast with L10H0 attention socres, there is no significant (inhibitory) interaction between the IO features in the query and key.

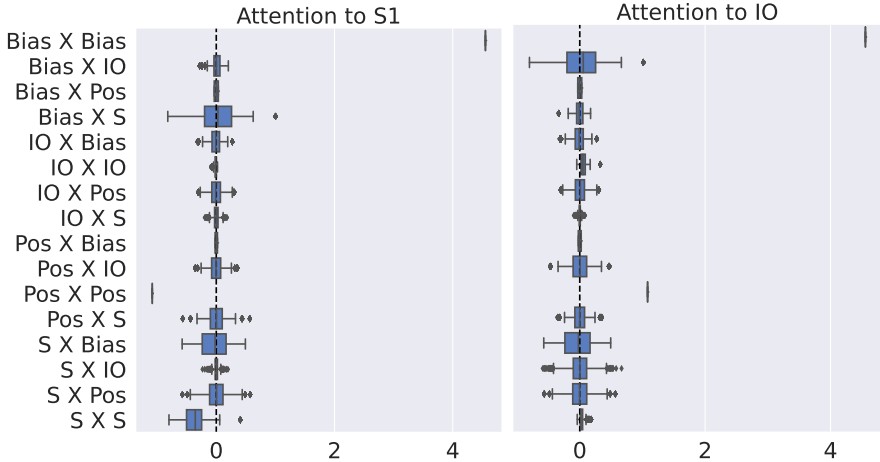

Figure 19: Attention score decomposition for the L9H9 name mover (see Figure 8 for explanation).

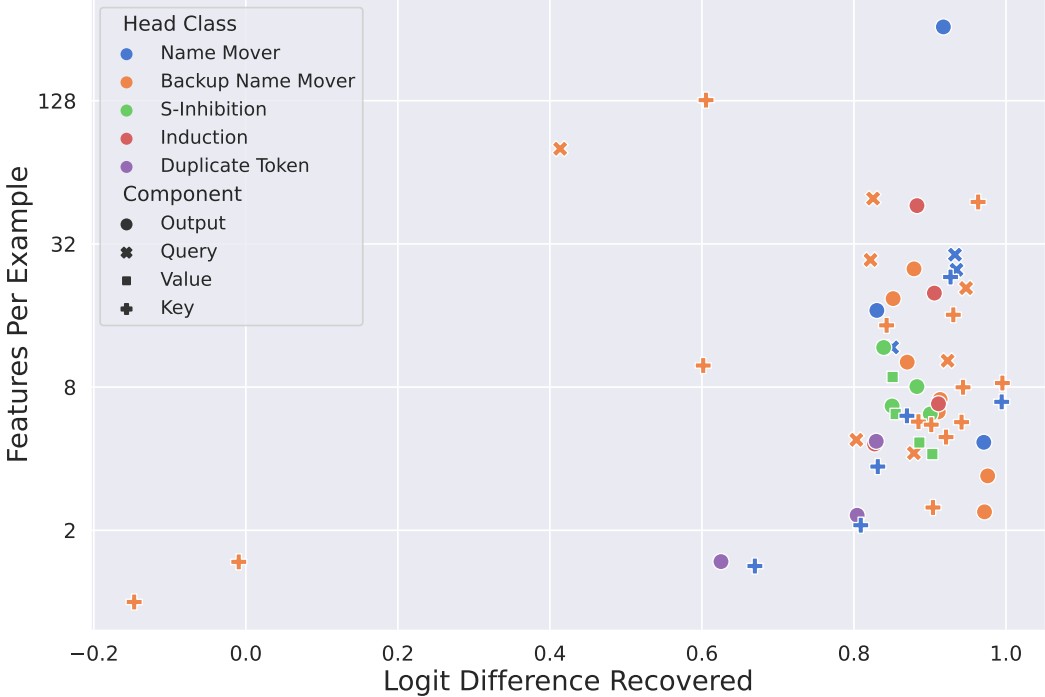

Figure 20: Metrics for our chosen task-specific SAEs for each relevant node in the IOI circuit. The x-axis shows the absolute value of the difference in logit differences between a clean run of the model, and a run where the activations at the given node are replaced by the SAE's reconstructions, normalized by the difference between the clean logit difference and the logit difference when the node is mean-ablated instead. The y-axis shows the average number of features active per prompt.

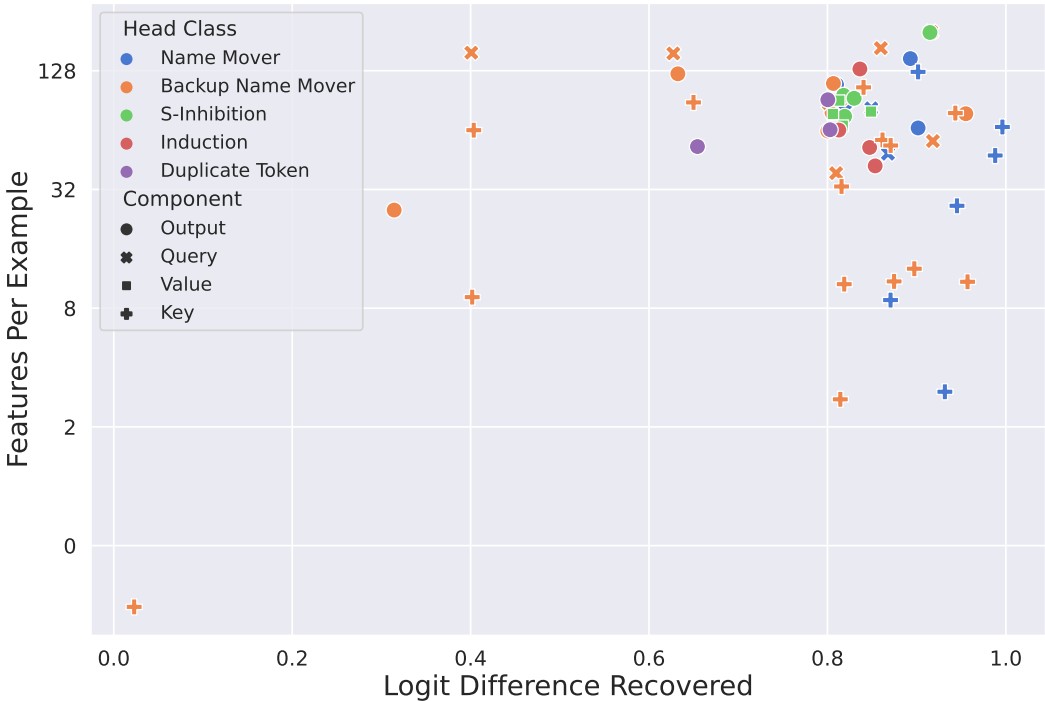

Figure 21: Counterpart to Figure 20 where the decoder vectors are frozen during SAE training.

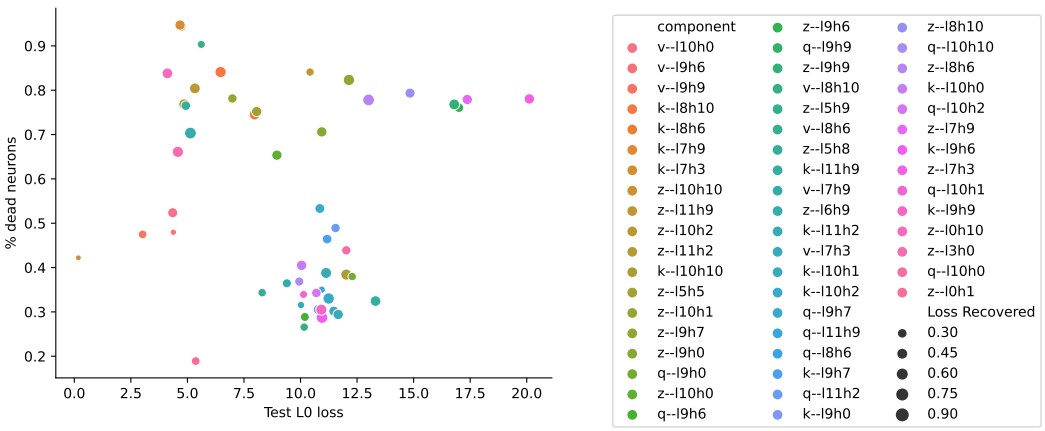

Figure 22: $\ell_0$ loss versus fraction of dead neurons for our full-distribution SAEs.

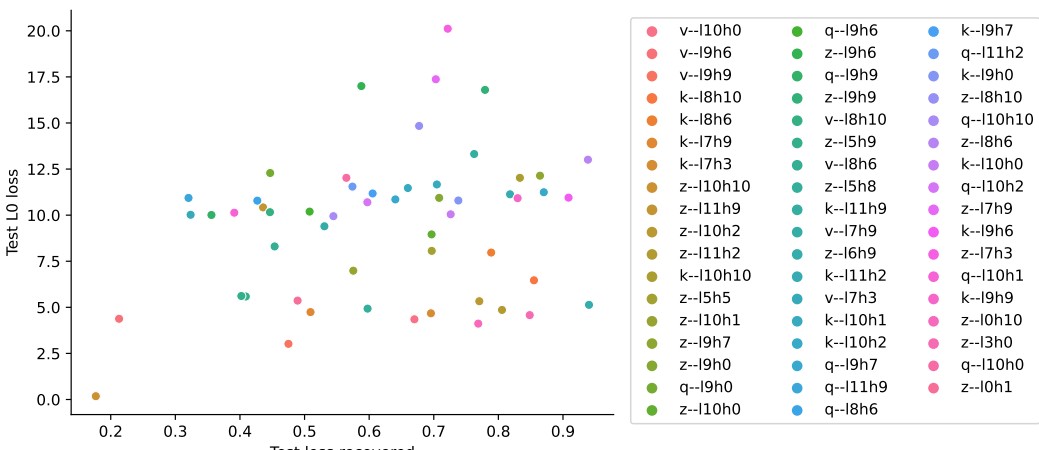

Figure 23: $\ell_0$ loss versus loss recovered (against a mean ablation) for our full-distribution SAEs.

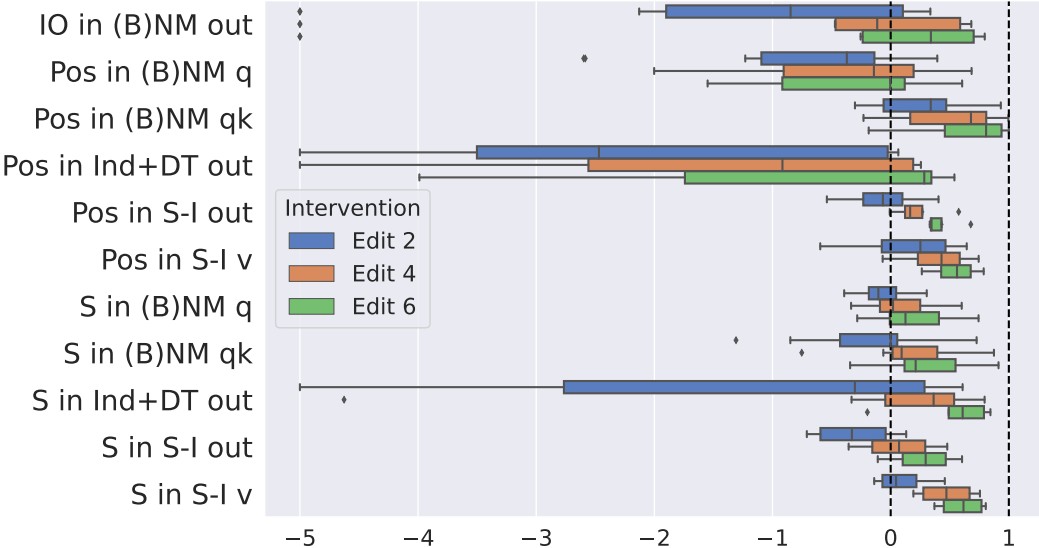

Figure 24: Distribution of the average feature weight removed by our interpretation-agnostic edits using SAE feature dictionaries, over all locations in a given cross-section, normalized by the corresponding weight for the supervised feature dictionaries. Weight removed values are transformed linearly so that a value of 0 indicates that the weight removed by the edit equals the weight removed by the corresponding 'ground truth' supervised edit; and a value of 1 indicates that the edit removed a total weight of 1, meaning that the edit essentially overwrites all SAE features present in the activation. Negative values are clipped at −5 to preserve readability.

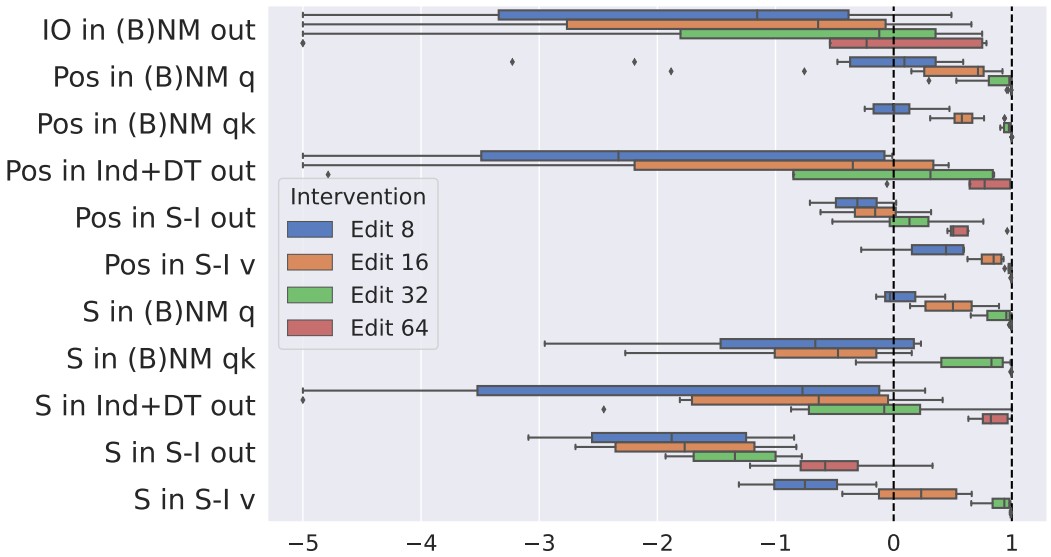

Figure 25: Counterpart of Figure 24 for full-distribution SAEs, when editing using features with high F1 score for the attribute

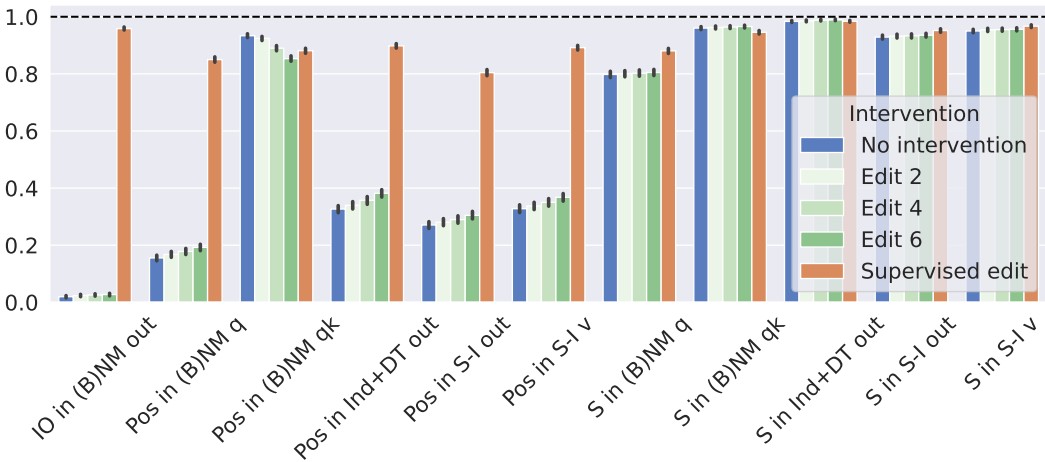

Figure 26: Counterpart to Figure 28, where the decoder vectors are frozen during SAE training.

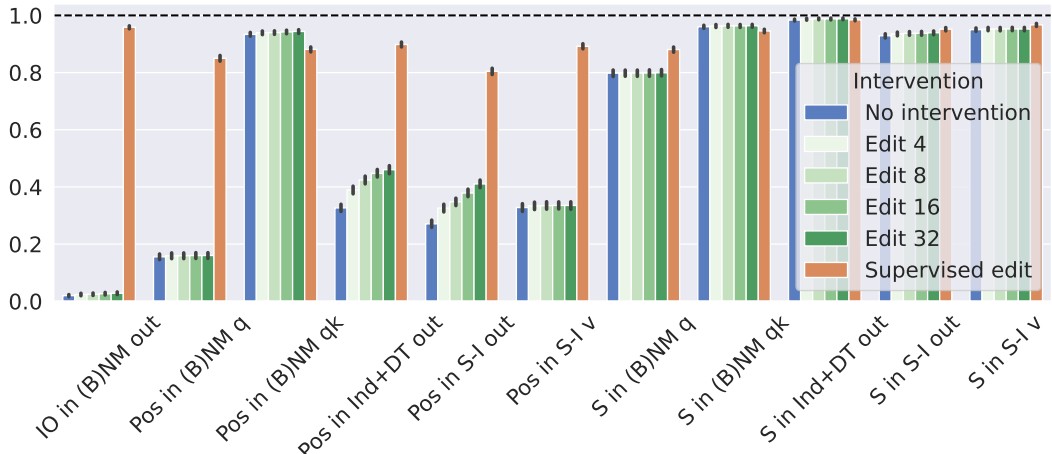

Figure 27: Counterpart to Figure 28, where we use full-distribution SAEs instead. Here, we need to change a much higher number of features in order to have a noticeable effect (and sometimes editing even 32 features fails)

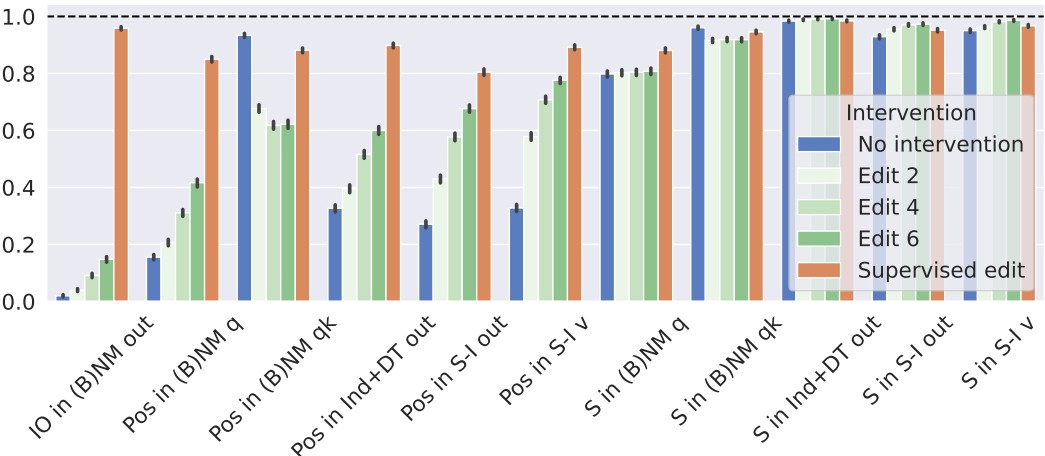

Figure 28: Accuracy when editing **IO**, **S** and **Pos** for circuit cross-sections using our supervised feature dictionaries and task-specific SAEs; the outcome in the absence of intervention is shown in blue for reference. When using task-specific SAEs, we edit either 2, 4 or 6 features (which means we in total add and/or remove up to that many features from activations). For comparison, supervised edits always involve removing 1 feature and adding 1 feature. Accuracy is measured as the proportion of examples for which the model's prediction agrees with the ground-truth prediction for the edit; see Section 4.3 and Appendix 7.5 for details.

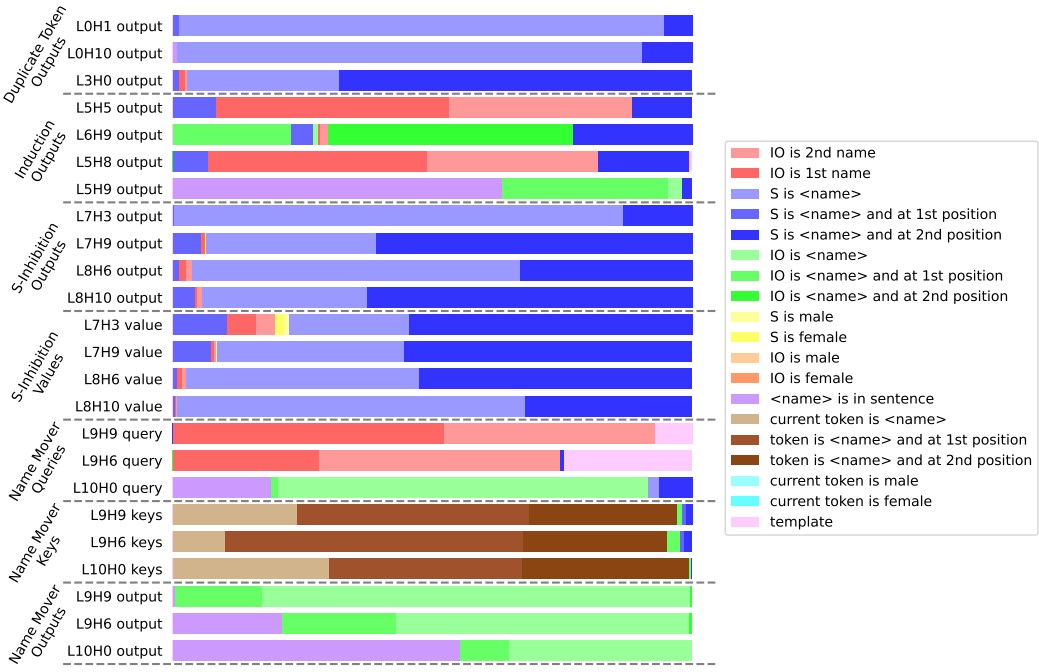

Figure 29: Interpreting the features learned by the task SAEs. For each node in the main IOI circuit (without backup/negative name movers), we show the distribution of the features which have an explanation with $F_1$ score above a threshold. The SAE chosen at each node is the one with the most interpretable features out of all SAEs trained on this node during our hyperparameter sweep.

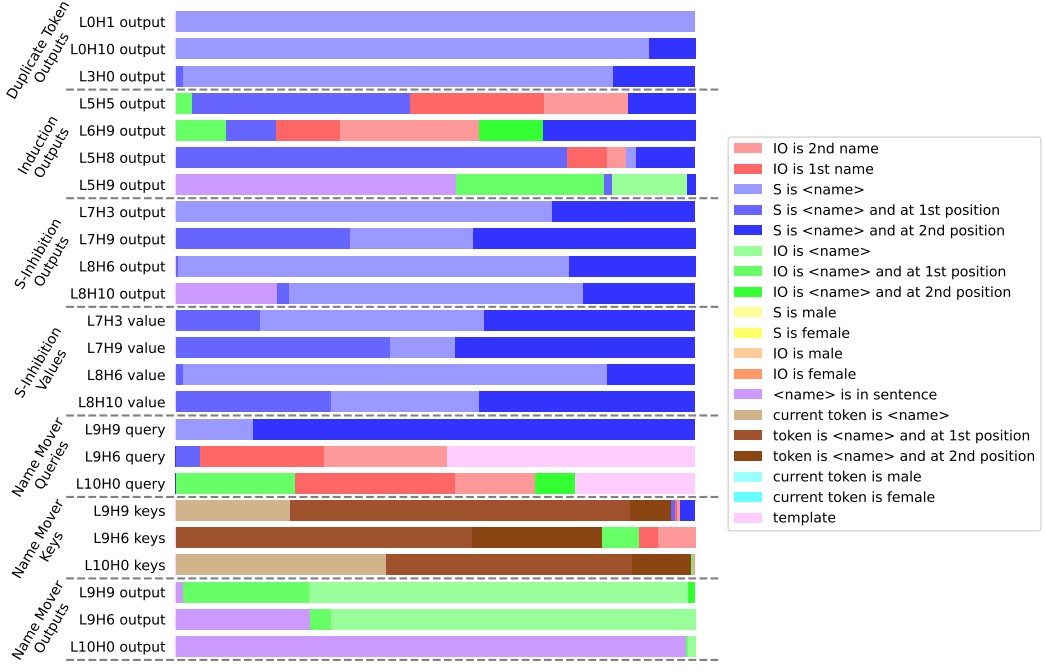

Figure 30: Interpreting the features learned by the task SAEs. This is a counterpart to Figure 29 for the SAEs chosen based on the $\ell_0$ and logit difference recovered metrics.

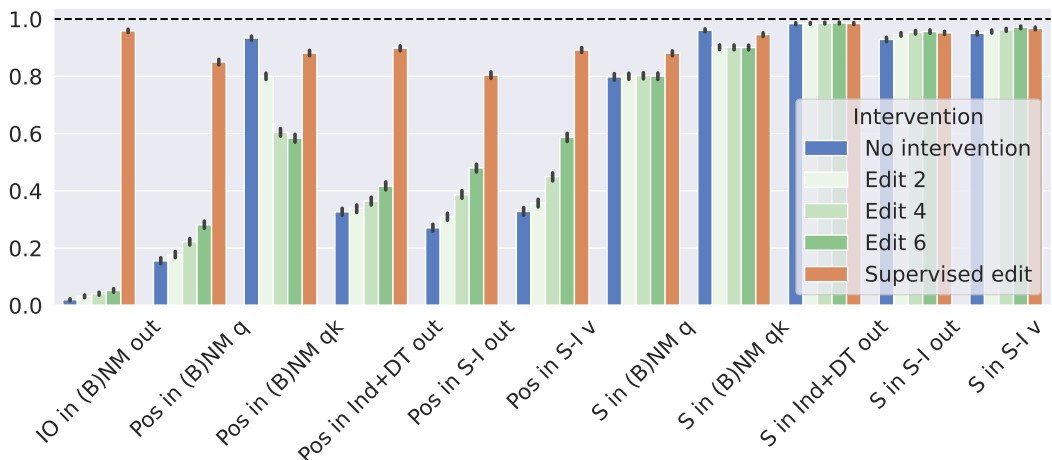

Figure 31: Interpretation-aware sparse control, using task SAE features with the highest $F_1$ score with respect to the given attribute.

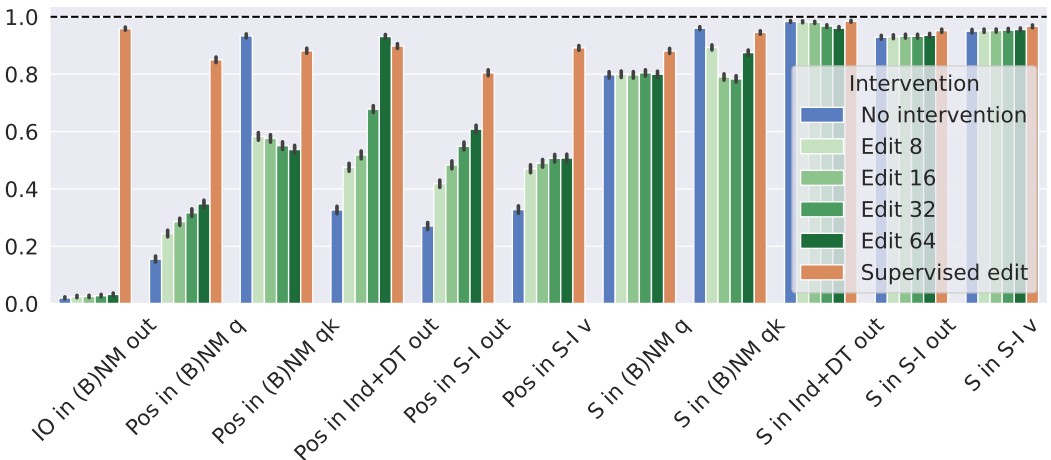

Figure 32: Counterpart of Figure 31 with full-distribution SAEs.

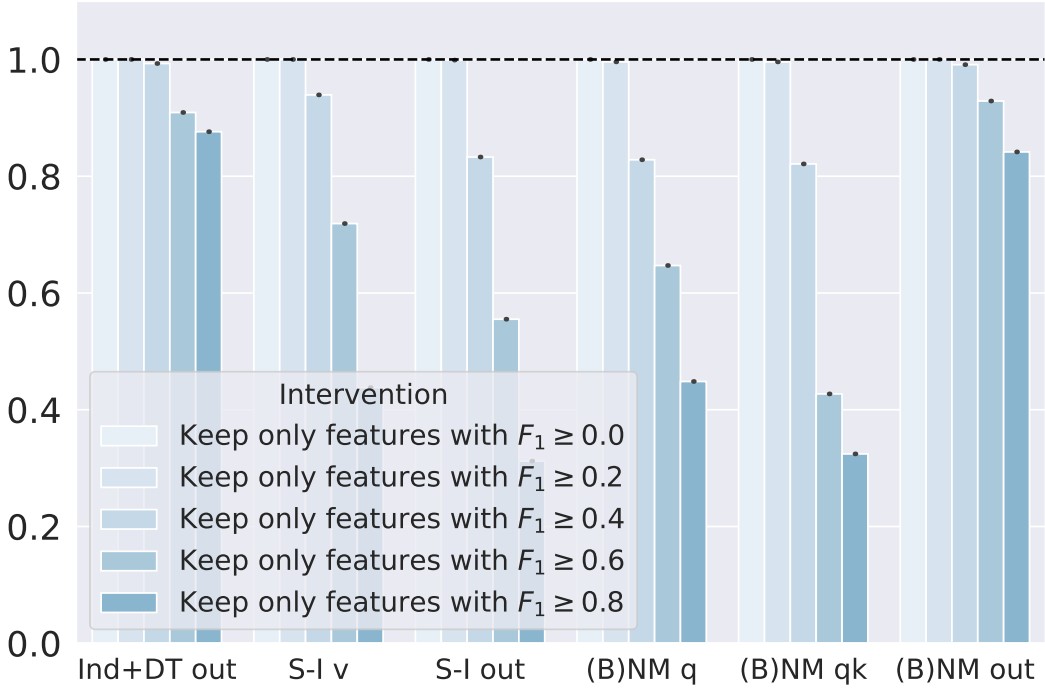

Figure 33: Measuring the **sufficiency** of interpretable features for task SAEs: effect of subtracting features with the lowest $F_1$ score from activations on logit difference. A value of 1 is best.

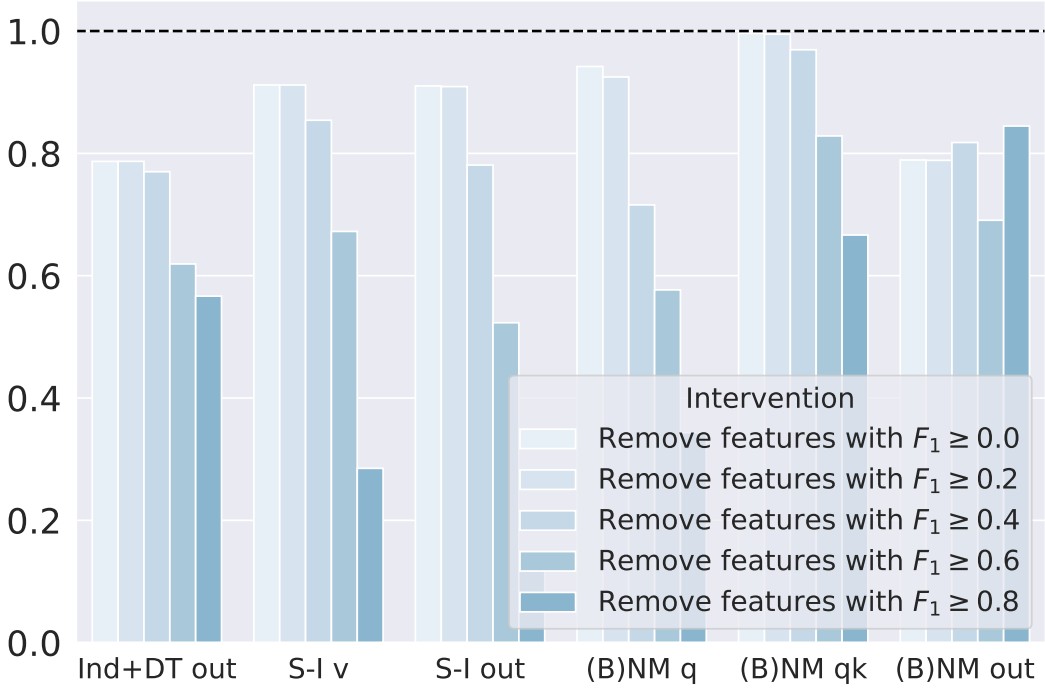

Figure 34: Measuring the **necessity** of interpretable features for task SAEs: effect of removing features with the highest $F_1$ score from activations on the logit difference. Values are rescaled linearly so that a value of 1 corresponds to perfect recovery of the logit difference achieved by mean ablation (i.e., ideal intervention removing all features). A value of 1 is best.

