# OpenReview forum: "Towards Principled Evaluations of Sparse Autoencoders for Interpretability and Control"
_ICLR.cc/2025/Conference — ICLR 2025 Poster_

### Official Review · Reviewer_TRbA · 2024-10-30

**Soundness:** 3
**Presentation:** 3
**Contribution:** 3
**Rating:** 8
**Confidence:** 4

**Summary:**

This paper introduces a way to compute sparse feature dictionaries that disentangle model computations similar to an SAE fashion using a supervised method; as well as introduces several ways to evaluate these feature dictionaries from different perspective such as necessity, sufficiency and controls of the feature dictionaries towards a specific task. The author applies the work to the indirect object identification (IOI) using GPT-2 Small, and compare the feature dictionaries obtained by their method, vs some other recent SAE variants such as Gated SAE adn Top-K SAE.

**Strengths:**

- Thorough study in the case of IOI. The use of IOI in the study mainly because we know the "feature" that more or less directly affect the outputs. This minimizes the possibilities of the cases that "SAE features" may not coincide with the features that human understands. This was also implied in Section 4.2.
- Extensive study on the paper that includes a lot of results in the appendix.
- Some proposed methods such as Necessity and sufficiency, to the control method proposed can be applied to a more general case, as described in Section 4.
- This paper also addresses a lot of the details on small nuances in evaluation of SAEs. This includes the discussion in Session 4.2, and the paragraph on "Preventing..." in Section 4.3.

**Weaknesses:**

**Overall**

- As this paper is a case study to IOI, it is somewhat restrictive. I don't think it is necessarily a weakness of the paper though as I don't think anything can be done to address this "weakness". It could be viewed as a stepping stone for future work as well.
- Presentation and Readability are the main weakness of this work. For example, experiment results in Section 3 includes a lot in section 4 such as description of FullDistribution SAE, Task SAE, TopK, Gated SAE etc. There are crucial formulae and methods that should be in the main text, but instead they are in the Appendix. For instance, the necessity formula in Section 7.4 ~line 878 in the appendix.
- The section on "Interpretability" such as in Section 3.2, Figure 5 and Section 4.4 are out of place. The author even mentioned that "we stress that this experiment is of a rather exploratory and qualitative nature". Putting these sections in the appendix would be much more coherent.
- Section 6 on discussions and limitations is too short. There should be more discussions on the experiment results. Possibilities of applying similar techniques to a general settings (as IOI provides the "known features"). Including a discussion on future work may be useful.

**Detailed issues**

- On Sufficiency in "Figure 3. Left" - as the experiment is to test whether the reconstructions are sufficient, we would hope to compare the logit difference of the original and the reconstruction. The method in Figure 3 shows the ratio of the *average* logit difference with and without the intervention. This may not be the best because the averaging may hide the changes in logit difference with the intervention for each example. A simpler method like the average changes (absolutized) of the logit difference _may_ work. The authors can also opt to include a brief discussion on this so that it does not seem like they were hiding something. For example, showing the distribution of  absolute changes of the logit difference in a histogram, or some statistics on it.
- Necessity. The experiment is to test whether the reconstructions are necessary. This means that we want to show that without the reconstructions, the model cannot do so well, resulting a drop in model performance - i.e. logit difference. Thus there are three quantities
1) The reconstruction $\hat{a}(p)$
2) The proposed quantity; average plus SAE error term
3) The average.

Showing necessity should be showing the difference between 1) and 2) are large. However, in the main text of the paper, it opts to show the difference between 2) and 3) only. A crucial formula and description of the necessity score directly addressing this problem is in the appendix (Section 7.4, around line 878), which in my opinion, should be in the main text.
- "Probing accuracy" in "Control": Seems out of place? Is it referenced somewhere else in the paper? Also no results were shown? In the absence of results, this section on probing accuracy does not seem to achieve the goal of the section: "measures the degree to which the supervised feature dictionaries disentangle the different attributes". This is because the probe is linear, which itself can "disentagle" the attributes. I think it would be better to either remove this section (put this in the appendix), or show some experiment results with discussions on the way to disentangle the attributes.

**Minor**
- Section 4.3 Expressing... line ~361. edit 3 --> (3) or Equation 3.
- Results (line ~424). objective 4 --> (4) or Equation 4.
- missing parenthesis at line 434 (resp. $a(p_t)$ **)** by their...
- broken references in appendix (line 883, 935, 1849)

**Questions:**

- Equation 2. Reason for the "bias" term is $E[a(p)]$? Does this mean $E[u_{IO}] + E[u_S] + E[u_{POS}] \approx 0$ if we take the expected value on $a(p)$ and $\hat{a}(p)$? If it is by design, can we make a comment on this design? What is a brief explanation behind this design?
- What is F and A in F1 Score? In the text, it seems F=the set of examples activating a specific SAE latent "f". A=binary attribute of a prompt. It seems that the F1 Score is applied **on each SAE latent**, as described in Section 4. How do we get "A" in this case? How do we know which "binary" attribute of a prompt that the latent f corresponds to? Can we give a more detailed explanation in the text? It would be nice to include some examples of F and A (specifically A)?

---

> ### Author Response · Authors · 2024-11-24
> **Response to reviewer**
>
> Dear Reviewer,
>
> Thank you for your detailed and thoughtful review. We especially appreciate your careful analysis of the paper's structure and technical content.
>
> **Regarding additional tasks**
> * See our top-level comment on additional tasks, where we generalize to new models, datasets and tasks.
>
> **Regarding presentation and readability:** We have:
> * moved some crucial formulae from the appendix to the main text (e.g., the necessity formula from Section 7.4)
> * put the information important to our paper about the new SAE variants (TopK and Gated) and how they're trained in the main body
> * extended the limitations section and added some directions for future work
> * moved the related work section to directly follow the introduction
> * added pointers from Figure 1's caption to relevant sections of the paper
> * reorganized the interpretability sections for better coherence
> * fixed the minor mistakes you pointed out
> * moved one of the exploratory figures on interpretability to the appendix
>
> **Regarding technical issues and experimental results:**
> * On sufficiency in Figure 3: We appreciate your point about averaging potentially hiding per-example changes. We have added a note of the distribution of changes in logit differences to address this concern; briefly, we observe empirically that logit differences are well-concentrated around their means.
> * On necessity: We agree this important formula should be in the main text and have moved it there. However, we disagree with your characterization that showing necessity necessarily means the difference between 1) and 2) (in your terminology). Specifically, with our setup, a necessity score close to 1 indicates that removing our reconstructions degrades performance similarly to removing all task-relevant information via mean ablation, suggesting our features capture the essential computation. We think this is the meaningful way to evaluate whether the SAE error term contains information relevant for the task.
> * On probing accuracy in Control: We have revised this section by adding results (figures contained in Appendix) that substantiate the claims made in this part of the paper.
>   * You said: "This is because the probe is linear, which itself can "disentagle" the attributes. ". However, this indicates a misunderstanding of the experiment. What we check here is two things:
>      * whether a probe trained to predict e.g. **IO** will "change its mind" when we edit the **IO** attribute in an activation;
>      * Whether probes for the other attributes will keep their original predictions
>   * In other words, we use probes merely as one tool to measure the information encoded in activations. We use this to evaluate whether our interventions have the desired effect on the information encoded in the activations.
>
> **Regarding your questions:**
> * **About the bias term:** yes, this is by design -
>   * It is well known that activations in LLMs may have non-zero average values over a dataset, especially when this dataset is very narrow/specialized compared to the full pre-training distribution. The bias term is a way to account for this, and let us focus on how activations differ between examples on the IOI task.
>   * As another justification, SAEs also have a similar bias term added to reconstructions. In this way, our supervised dictionaries imitate this design choice.
> * **About F1 score**: you are correct; we have added more explanation about this (with examples) to the relevant parts of the paper.
>
> We are very grateful for your many detailed observations about our paper. Please let us know if these revisions address your concerns. We remain available to answer any additional questions during the discussion period.

---

> ### Author Response · Authors · 2024-11-25
>
> Before this phase of the discussion period ends, we wanted to ask the reviewer whether we have addressed your concerns with our work?
>
> We thank you for the valuable feedback, especially regarding presentation, readability, and the structure of this work, and we believe we have now resolved these issues.

---

> ### Comment · Reviewer_TRbA · 2024-11-25
> **Response to author**
>
> Thanks to the authors for addressing the comments. I especially appreciate the inclusion of the necessity formula being moved to the main text, adding some limitations and future work, as well as moving the exploratory figures to the appendix. I think the current presentation is better than before - but there are still places of minor improvements.
>
> - *F1 score*. I still don't think the F1 score description in section 3 is clear enough. If I understand correctly, the F1 score is a measure for interpretability, which I think in the original comment it is out of place. The description of "A" is still not clear. Is it done automatically, or does it depend on the choice of human-defined subsets as in section 5.5? More comments on the sections on interpretability follows.
>
> - *Figure 3*. The supervised/Task SAE/Full distribution SAE are introduced. The Task SAE and Full distribution SAE was slightly introduced in the introduction. I think adding a sentence reminding what Task SAE and full distribution SAE in the beginning of section 4.3 (just like the beginning of 5.1) will do.
>
> - *Sufficiency*: I apologize that I did not raise this last time. I think for clarity, one should describe it in terms of formulae. For each p, we have a logit difference logitdiff(p) given by the difference in model logprobs for IO and S names. Thus, if we perform the intervention, we will have a different logit difference, say $logitdiff_{intervention}(p)$. From the text, it seems like what you were doing was to do $E[logitdiff_{intervention}(p)] / E[logitdiff(p)] \approx E[logitdiff_{intervention}(p)] / 3.3$. (Also, are there absolute values any where?) Also, thanks for addressing the concerns of the per-example change.
>
> - *Necessity*: Thanks for addressing the concern. I think the presentation has improved a lot. It would be nice to say it even clearer, if I understand correctly, as
>   - $a_{intervention}(p)$ = E(..) + error term,
>   - $a_{mean ablation}(p)$ = E(..),
>   - $a_{clean}(p) = a(p)$.
>
> Then the subscript in logitdiff will be the same as the "a"'s. In this case, we also don't have to use the left arrow to indicate assignments. Regarding the "difference between 1 and 2", I agree that the formula is a reasonable measure for necessity.
>
> - *Probing accuracy*: Thanks for addressing the comments. I think I understand what you are doing in this section.
>
> - *Bias term*: Thanks for addressing it in the comments.
>
> Upon a second reading, regarding the interpretability sections including the section at the end of section 4 as well as section 5.5,
>
> - I think the interpretability at the end of section 4 is very interesting - and I think that alone can be a separate paper, with more experiment results. Thus one could remove this section and put it in the future work. The reason I think it is out of place is because a) this section is not related related to any of the other parts of the paper; b) the results are only shown in the appendix; and c) both the methods and the results are interesting on their own.
>
> - Section 5.5 is less interesting (to me) compared to the end of section 4, but it is more related to the other parts of the paper. It is still slightly out of place because the methods and experiments in this section are not clearly described. (They are of course much more clearly described in the Appendix)
>
> - Both sections are labelled "interpretability" under "evaluating Sparse Feature dictionaries" and "evaluating SAE", but they correspond to different methods. This is also confusing.
>
> My original suggestion was to remove both sections - or put it in the appendix. I still think this should be the case. However, in case the authors strongly prefer to keep these sections, I think they should at least rename the "interpretability" section at the end of section 4 to "Feature-level Interactions" instead.
>
> I also appreciate the addition of more discussions on limitations and future work. The removal of these interpretability sections also can benefit a more thorough discussion on the results and the limitations. For example,
> - section 5.3, "We find that vanilla task SAEs are good at sufficiency/necessity, but full-distribution SAEs are not.". Are there any discussions or explanations on why this is the case?
> - Expand on how to extend the proposed methods to datasets other than IOI. Thus, can we compute supervised feature dictionaries without knowing the set {IO, S, Pos}? Can we evaluate sufficiency and necessity using similar methods for SAE features?
>
> The above two questions are just suggestions of discussion items if the interpretability sections are removed.

---

> > ### Author Response · Authors · 2024-11-26
> > **Implementing and discussing suggestions provided by reviewer**
> >
> > Thank you for these great suggestions and the very constructive discussion so far! Upon a re-read, we agree your proposed edits make a lot of sense, and we have implemented ~all of them in the latest revision. A few notes on what we changed, as well as what we didn't change and why:
> >
> > - we added an explicit note in the $F_1$ score's definition that $A$ refers to an interpretation that we wish to evaluate
> > - regarding the reminder about full vs task SAEs in 4.3, we feel this is unwarranted, as this section does not deal with SAEs at all; we feel that pointing the reader to the relevant bars in the figure (referred in the text by color) should be less confusing.
> > - we introduced lightweight notation for the logit difference under an intervention to better explain the necessity formula
> > - we re-used this notation to explain the sufficiency formula, which in turn greatly shortened this section while noticeably improving the presentation. **Thank you very much for this astute observation!**
> > - as suggested, we renamed the part about interpretability of supervised features to "Sparsity of supervised feature interactions"
> > - we included in 5.3. a hypothesis about why task SAEs are better at full-distribution ones at sufficiency and necessity; briefly, we believe it's just because they're trained on a dataset much closer (in fact, precisely matching) to the task distribution under investigation, despite being a much smaller dataset than the full pre-training one.
> > 	- It has become a theme of other related works in the SAE literature that task SAEs may be a valuable addition to the interpretability toolkit, as they may surface domain-relevant features with low sample complexity.
> > - we shortened section 5.5. significantly, while retaining the key elements of our exploratory evaluation framework; we feel these are still valuable additions to the paper (hence we still keep 5.5)
> > - we expanded the limitations section a bit.
> > 	- we note that we had already added notes on how the method can be generalized to other tasks, and how we can target only specific attributes and model locations to reduce the amount of "manual" labor required to find good attributes
> > 	- however, also note that the sufficiency and necessity evaluations are independent of the choice of attributes, because they evaluate SAE reconstructions as a whole. Hence they readily generalize to other tasks (but also tell us nothing about disentanglement)
> >
> > Let us know if this addresses your concerns. We also remain available throughout the discussion period to discuss any further changes that improve your evaluation of the paper.

---

> > > ### Comment · Reviewer_TRbA · 2024-11-27
> > > **Response to author**
> > >
> > > Thanks to the authors for the quick reply. I think it addresses most of the points I raised. I appreciate the changes made. I especially like the change in the notation in both the sufficiency and necessity section. I admit that I missed the "orange bar" in pointer of 4.3.
> > >
> > > Also thanks for shortening section 5.5. I think now the presentation is quite neat.
> > >
> > > For the generalization to other tasks, I think I have not made my point clear. In the cases that we actually do not know the attributes of the tasks, how can we apply this method? The "other tasks" described in this papers are simple "greater than" and "both". But it is possible that, say, a QA dataset and we train an SAE using some inner layer embedding. How can we evaluate that? I am not looking for an answer here - but only as a discussion point.
> > >
> > > Final Comments:
> > > I think the authors did a great job in tackling various problems and running experiments in the field. Due to the page limit, it is not uncommon that to include all the methods and results in a paper. Unfortunately this reduces the readability and the focus of the paper. The reader will find the ideas of the paper all over the place. The latest edit did make the paper much more readable.
> > >
> > > Thanks for the work.

---

> > > > ### Author Response · Authors · 2024-11-27
> > > > **Adding discussion on open-ended tasks**
> > > >
> > > > Thanks again for continuing the constructive discussion in such a timely manner. We're very grateful for your feedback, which has greatly improved the clarity of our paper, and we're glad you find the recent changes a major improvement to the presentation.
> > > >
> > > > Regarding generalization to other tasks: oh, OK - sorry we didn't get that the first time. Yes, generalizing the methods here to open-ended tasks is a very valid question, and qualitatively different from the things currently discussed in the paper. We've uploaded a new revision that adds a short discussion of this at the very end of the paper, pointing to two recent works that seem like promising starting points for crafting supervised feature dictionaries. For the purposes of this paper, we wanted to get the basic parts of the methodology right in a very controlled setting.
> > > >
> > > > In particular, one of the works we added [1] proposes a way to create concept dictionaries that encode single-word concepts by computing vectors with supervision (e.g., by averaging token representations across sentences mentioning the word) and then decomposing activations using these vectors as an overcomplete basis, plus an L1 penalty encouraging sparsity. This is very similar to an SAE - except the decoder vectors are pre-computed and not learned. Some preliminary experiments by one of the authors indicate that these concept vectors are enough to explain a meaningful chunk of activation variance at satisfactory sparsity, but still a lot of variance remains unexplained. Still, this framework gives a scalable way to at least begin to tackle the question of
> > > > "ground-truth" attributes in open-ended settings. We think this is a very interesting area for future work.
> > > >
> > > > [1] Luo, Jinqi, et al. "PaCE: Parsimonious Concept Engineering for Large Language Models." arXiv preprint arXiv:2406.04331 (2024).

---

### Official Review · Reviewer_w15N · 2024-11-02

**Soundness:** 3
**Presentation:** 4
**Contribution:** 2
**Rating:** 6
**Confidence:** 3

**Summary:**

This paper focuses on evaluating sparse autoencoders (SAEs) for their ability to recover known ground-truth features learned by a model. To do so, the authors first train a supervised dictionary on the indirect object identification (IOI) task, for which model computation is already relatively known due to prior interpretability and circuit discovery work. Both IOI task-specific SAEs and full-distribution SAEs are trained and evaluated with respect to the supervised dictionaries to understand if SAEs allow for the same level of approximation, control, and interpretability as the supervised dictionaries. Results reveal that more recent SAE architectures improve these capabilities and task-specific SAEs are much more effective than full-distribution SAEs.

**Strengths:**

- The writing and paper structure are very clear and easy-to-follow.
- The focus of this paper is highly relevant and interesting. The problem of finding a ground truth with which to evaluate interpretability and explainability methods has remained an issue for decades, and this work works towards solving this problem by exploring using human-generated groundtruths that have been backed up by prior work.
- The experiments are well-defined, inutitive, and easy to understand.
- I believe the results are interesting and useful - they reveal that task-specific SAEs are more useful in practice than full-distribution SAEs, hinting that data quality is of utmost importance when training SAEs. Further, this suggests that human priors may be useful when developing interpretability methods/SAEs.

**Weaknesses:**

- I find the “Why use these attributes?” paragraph in Section 3.1 to be confusing. If prior work had not proposed the IO, S, and Pos attributes, how would one go about defining the supervised dictionary? If the evaluation pipeline described in this paper were to be used for a different task, I’m not sure whether this section would be generalizable. In particular, when there are many choices of attributes, what is the manner of choosing between them without using another interpretability method, which would then prevent the need of using an SAE in the first place?
- It would have been significantly more convincing to me if the authors had considered more than one task in their evaluation. At the moment, it’s unclear to me how the proposed methodology and results from this work could be applied to future works that want to evaluate trained SAEs.
- The section on interpretability (section 4.4) is also a bit confusing to me - I would find it very helpful if the authors provided interpretations of the SAE latents, and a visualization of how these features could then be used to explain the LLM’s computation on a single example. Some examples of /possible/ interpretations are provided in Appendix 7.13-7.14, but if I understand correctly these are not the actual labels of the SAE features.
- It is my understanding that the authors wish to propose the use od supervised dictionaries an evaluatory baselines for SAEs. However, in practice, this paper reads more as an exploration of whether SAEs can recover the IOI task. While the authors discuss the limitations of hardcoding the attributes to compare SAEs against and only considering a single task and model, I believe these drawbacks fundamentally limit the work in its general utility.

**Questions:**

- In section 4.3, if I understand correctly, the SAE latents are found by simply optimizing/searching for the features that perform the task (move one latent to the specified counterfactual latent). This seems a bit roundabout to me - wouldn’t this propose that you need to know the features you are looking for in order to label SAE features? How would one do this searching or interpretation without access to the counterfactual activations? Wouldn’t it be more realistic to interpret or label each SAE feature and then use the features that are labelled to be relevant to the task at hand?
- Please see the above weaknesses!

---

> ### Author Response · Authors · 2024-11-24
> **Response to reviewer**
>
> Dear reviewer,
>
> Thank you for your detailed and thoughtful comments.
>
> We note that a central concern in your review is the restricted nature of our evaluation (considering only the IOI task with a single model and dataset). To this end, we have applied our framework with minimal changes to a new setup combining a new model, new pre-training dataset, and new linguistic task, for the purpose of evaluating SAEs trained on the full pre-training dataset of the LLM being analyzed. Please refer to our top-level comment on generalizability of results for more details.
>
> To address your other concerns:
>
> * While we agree that focusing on IOI may seem narrow, we believe it serves as a valuable proxy for evaluating SAEs more broadly. Practitioners often need to evaluate SAEs across hyperparameter sweeps or compare new architectures, and the IOI task better represents realistic downstream applications than standard proxy metrics. Importantly, even though IOI is a special case where prior work identified relevant features, evaluating full-distribution SAEs on this task provides evidence for their effectiveness on tasks where ground truth is unknown.
>
> * Regarding how attributes are chosen: you say that "In particular, when there are many choices of attributes, what is the manner of choosing between them without using another interpretability method, which would then prevent the need of using an SAE in the first place?" - however, this objection runs orthogonal to the goal of our paper.
>    * Specifically, our goal is to, through whatever means, find a situation in which we can have an *independent* ground-truth evaluation of SAE quality. The prior work on IOI gives us (through lots of manual effort) a good understanding of the IOI circuit, which allows us to set up an independent evaluation by comparing features in the IOI circuit (found via non-SAE means) to SAE latents.
>    * In our newly added task, we have shown that we can construct supervised features like we do for IOI (via conditional expectation over attribute values), and we check that the supervised features disentangle two relevant properties for the task. This serves as validation that the supervised features are relevant to the model's computation (similar to IO, S and Pos).
>
> * You say "Some examples of /possible/ interpretations are provided in Appendix 7.13-7.14, but if I understand correctly these are not the actual labels of the SAE features." - indeed, the labels of SAE latents in figure 5 *are* the ones from Appendix 7.13; we are sorry for the misunderstanding!
>
> * You say "How would one do this searching or interpretation without access to the counterfactual activations? Wouldn't it be more realistic to interpret or label each SAE feature and then use the features that are labelled to be relevant to the task at hand?" - we are sympathetic to your objection; some qualifications:
>    * The reason we set things up in this way is that we first ran an extensive validation showing that IO, S and Pos are indeed meaningful for the model's computation on IOI. In this sense, it makes sense from a human perspective to want to see how well SAE latents can control these attributes.
>    * We agree this is somewhat biased, and have tried to be agnostic to whether SAEs represent these features in a 1-to-1 way or some more complex pattern. Still, this does not fully address your concern. We hope that future work can move towards an even more agnostic evaluation.
>
> Please let us know if these responses address your concerns. We remain available to answer questions throughout the discussion period.

---

> > ### Comment · Reviewer_w15N · 2024-11-25
> >
> > Thank you to the authors for their thoughtful rebuttal and additional experiments.
> >
> > * Regarding your second point, if your goal is to find independent ground truth evals, but the development of the ground truth comes at the cost of "lots of manual effort," it is still difficult for me to determine how useful your proposed approach is for evaluation of future models. Wouldn't this same manual effort be required for each task a new model is to be evaluated on? I believe this (the necessary manual labor required to construct the ground truth baselines) should be discussed as a key limitation of the work and made clear in the intro if possible.
> >
> > * Regarding your fourth/last point, I understand why you made this choice and I appreciate that you acknowledge it in the rebuttal, but I believe this too should be noted as a limitation of the work in the main paper.
> >
> > Contingent upon the above, I will raise my score to a 6. Thank you!

---

> > > ### Author Response · Authors · 2024-11-25
> > > **Response on implementing reviewer feedback**
> > >
> > > We thank you for the productive discussion and your willingness to update your score in light of our responses. We agree with your feedback, and have implemented it in the latest revision of the paper. Specifically, we added:
> > > - in the introduction:
> > >
> > > > A
> > > limitation of our approach is that it requires potentially substantial per-task effort to independently
> > > identify the relevant attributes, which is proportional to the complexity of the task and the number
> > > of attributes we want to consider. However, as we show in Appendix 7.1, our framework allows for
> > > the targeted evaluation of a few attributes in a few locations of the model, which can substantially
> > > reduce this effort.
> > >
> > > - in the discussion on limitations:
> > >
> > > > This influences many of our evaluations, in particular
> > > the ‘sparse control evaluation’ from Section 5.4, which relies on these attributes to compute the
> > > counterfactual activations used to evaluate edit accuracy.
> > >
> > > and
> > >
> > > > Finally, a third limitation is that it takes work to identify in a way independent of SAEs what good
> > > attributes for a given task are. We show through two case studies in Appendix 7.1 that, by targeting
> > > only key attributes in a single activation location, we can reduce this effort substantially. This comes
> > > at the cost of analyzing a more limited, but still potentially interesting slice of the task.

---

### Official Review · Reviewer_S53N · 2024-11-03

**Soundness:** 3
**Presentation:** 3
**Contribution:** 3
**Rating:** 6
**Confidence:** 2

**Summary:**

This paper studies the Sparse autoencoders to capture interpretable features for the IOItask and the expeirment results show that the proposed approach achieves the best performance.

**Strengths:**

This paper is well-written and easy to follow.

**Weaknesses:**

There are several weaknesses:

1. The motivation of this section should be enhanced.

2. The English language should be improved.

3. The main idea seems not very novel. This paper should provide a strong motivation.

4. The experiment can be further improved by providing more results and analysis.

**Questions:**

Please see the weakness section.

---

> ### Author Response · Authors · 2024-11-24
> **Response to reviewer**
>
> Dear reviewer,
>
> Thank you for your comments. We appreciate your focus on strengthening the paper's motivation and empirical validation.
> We have made several significant changes to address your concerns:
>
> **Motivation and Novelty**: We have substantially reworked the introduction and related work sections to better motivate our contribution (see top-level comment). The key insight is simple: while SAEs are increasingly used for model interpretability, existing evaluations rely only on indirect metrics. Our work provides the first ground-truth evaluation framework using supervised features of the same mathematical form as SAEs.
>
> **Additional Empirical Results**: We have strengthened the empirical validation by:
> * Adding a complete new case study applying our framework to a different model, dataset, and linguistic task (see the top-level comment on generalizability of results)
> * Demonstrating that our supervised features successfully disentangle relevant task attributes in this new setting
> * Showing how our evaluation methodology transfers with minimal changes
>
> **English Language**: We have carefully revised the manuscript for clarity and readability, with particular attention to:
> * Technical explanations and terminology
> * Flow between sections
> * Motivation of methodological choices
>
> Please let us know if you would like us to clarify or expand on any of these changes. We remain available for questions throughout the discussion period.

---

> > ### Comment · Reviewer_S53N · 2024-11-26
> > **The response**
> >
> > Thank you for your feedbacks, which address my all concenrs. I will keep my previous score.

---

> > > ### Author Response · Authors · 2024-11-26
> > > **Response by authors**
> > >
> > > Thank you for confirming that our response has addressed all your concerns. Given that the concerns that led to your initial score have been resolved, we would greatly appreciate understanding what additional improvements would help you view the paper more favorably.

---

> ### Author Response · Authors · 2024-11-25
>
> Before this phase of the discussion period ends, we wanted to ask the reviewer whether we have addressed your concerns with our work?
>
> We believe we have addressed the concerns you raised regarding motivation, clarity of language, and including sufficient technical detail.

---

### Official Review · Reviewer_jgpT · 2024-11-05

**Soundness:** 3
**Presentation:** 3
**Contribution:** 3
**Rating:** 8
**Confidence:** 2

**Summary:**

The authors propose a (principled) method allowing to create supervised dictionaries for space features, which allow for evaluating the degree of disentanglement of SAEs. The developed method is then applied to SAEs and LLMs, witnessing not only interpretable later variables, but also providing possibility of editing attributes. Metrics of sufficiency, necessity and control are used for this.

**Strengths:**

In general the manuscript is well written and "strong".

The question which is really to ask is how much this finding is relevant for the literature. Here I should say my expertise is perhaps too limited to provide a proper judgment.

**Weaknesses:**

Though this has most probably costed a lot of work, the empirical validation of the proposed methodology is rather scarce. Whether the constructed dictionaries would also function for other tasks/semantics is not clear.

The mathematics are well explained, in clear and simple way.

Some (not that many) parts of the manuscript I had to read several times, e.g., the title under Figure 1 or the paragraph on interpretability at the end of Section 3.2. But in general formulas aid much understanding.

**Questions:**

In the title of Figure 1, could you make a clear connection of the text with precise parts of visuals, to facilitate the understanding?

Taking into account the size of the paragraph on related work, it should be possible to describe the related work without much terminology and thus shifting it closer to the beginning of the manuscript. This would allow the reader to better position the framework with respect to the state of the art.

---

> ### Author Response · Authors · 2024-11-24
> **Response to reviewer**
>
> Dear reviewer,
>
> We warmly thank you for your enthusiastic reception of our paper and your thoughtful comments. Regarding the relevance of our work to the literature (where you noted your expertise might be limited): ours is the first work to provide a thoroughly validated approximation of ground-truth features to evaluate SAEs against. We believe our evaluation framework provides important results for the IOI task, and we have also shown it can be readily generalized to other linguistic tasks. We have reworked the introduction and related work sections to more clearly situate our paper within the field of SAE evaluations, which we hope will help readers better understand our contribution.
>
> For a concise summary of the main changes in the revision, please also consult our two top-level comments:
> * **On novelty/motivation**: we describe briefly and clearly our contribution and how we re-worked the paper to present it better
> * **On generalizability of results**: we give a concrete example with another task where we show that our approach for computing supervised dictionaries and evaluating SAEs works too.
>
> We have also incorporated your helpful suggestions:
> * Moving the related work section to directly follow the introduction
> * Adding pointers from Figure 1's caption to relevant parts of the paper
>
> Thank you again for your constructive feedback. We remain available to answer any questions.

---

### Author Response · Authors · 2024-11-24
**Top-level comment on motivation/novelty**

We thank all reviewers for their feedback. Since the unclear presentation of the paper's motivation and novelty were shared concerns for multiple reviewers, we would like to respond with a top-level comment addressing this concern.

We have (in the attached revision) **reworked the introduction and related work sections of the paper to emphasize the motivation of our work and what sets it apart from other works in this area** (changes marked in red). In brief:
- the research community has invested a lot of effort in the SAE paradigm; the main goal of this paradigm is to recover features the model uses to perform computations on various tasks.
- However, prior evaluations have focused on *proxy metrics* that we only hope correlate with recovery of these features
- The novelty of our work is, by contrast, in being the first to compare SAEs to carefully validated approximations of ground-truth features in a realistic linguistic task (as well as two other tasks we added during the discussion period to showcase the flexibility of our framework; see the top-level comment on generalizability of results). Moreover, these ground-truth features provide a skyline of a supervised dictionary trained to predict the labels of features discovered by prior work, which allows a fair comparison grounded in the limits of the representational power of SAEs, and allows us to contextualize the outputs of our metrics.

While we have added two other tasks to demonstrate broader applicability (see other top comment), we believe our framework provides significant value even with a limited task selection. Practitioners often need to evaluate SAEs across hyperparameter sweeps [1] or compare new architectures [2,3,4], and the IOI task better represents realistic downstream applications than standard proxy metrics (which we discuss in the related work of the paper). Even though IOI is a special case where prior work identified relevant features, evaluating full-distribution SAEs on this task provides evidence for their effectiveness on tasks where ground truth is unknown. The same applies to evaluating SAE training methods via task-specific SAEs.

We thank the reviewers for pointing out the poor readability of the paper's motivation, and hope that our revisions and comments will mitigate this issue.

[1] Lieberum, Tom, et al. "Gemma scope: Open sparse autoencoders everywhere all at once on gemma 2." *arXiv preprint arXiv:2408.05147* (2024).

[2] Rajamanoharan, Senthooran, et al. "Jumping ahead: Improving reconstruction fidelity with jumprelu sparse autoencoders." *arXiv preprint arXiv:2407.14435* (2024).

[3] Gao, Leo, et al. "Scaling and evaluating sparse autoencoders." *arXiv preprint arXiv:2406.04093* (2024).

[4] Rajamanoharan, Senthooran, et al. "Improving dictionary learning with gated sparse autoencoders." *arXiv preprint arXiv:2404.16014* (2024).

---

### Author Response · Authors · 2024-11-24
**Top-level comment on generalizability of results**

We thank all reviewers for their feedback. Since the seemingly restricted nature of the paper's evaluation (which is based only on the IOI task) was a shared concern of most reviewers, we would like to respond with a top-level comment addressing this.

We have (in the attached revision) **added a new appendix section (as well as relevant pointers from the main text) on how our methods can be straightforwardly extended to a new natural language task we introduce on a different model and distribution, to test an SAEs ability to independently represent and disentangle different concepts**. We additionally apply our methods to another task from the mechanistic interpretability literature that has been studied in a way similar to IOI: the greater-than task [1]

In brief:
- We generalize the setup of the paper in multiple ways:
  - **dataset and model**: we use the Tiny Stories dataset and the 4-layer 33M parameter model from the Tiny Stories paper (https://arxiv.org/abs/2305.07759)
  - **full-distribution SAEs**: we use TopK SAEs trained on the full set of Tiny Stories activations over the 4 residual streams of the model (the dataset has on the order of 100s of millions of tokens)
- We consider sentences of the form "NAME really loves ANIMALS. NAME also really loves SPORT. So NAME really loves both"
  - For example, "Lily really loves cats. Lily also really loves basketball. So Lily really loves both" should be followed by "cats and basketball" or "basketball and cats", so the model should be able to put high probability on both cats and basketball coming next. A well functioning SAE should be able to disentangle these, ie we should be able to damage the model's ability to say cats but not basketball, and vice versa
  - We consider several values for each of NAME, ANIMALS and SPORT to sample a dataset of 1k prompts of this form
- Intuitively, at the "both" token, the model "prepares" to output either the ANIMALS or SPORT. We verify experimentally that the next-token probability distribution after "both" prefers the correct SPORT compared to other sports, as well as the correct ANIMALS compared to other animals.
- We turn this into a task to which we apply (key parts of) our framework. Namely:
  - We compute supervised features corresponding to each given SPORT and ANIMALS value by taking conditional expectations (like in the paper)
  - We check that the supervised features work to edit the logit distribution in the expected way
  - We compare this to SAEs via a sparse control test to change the value of one variable at a time via an SAE latent exchange.
  - This setup also allows us to measure the "disentanglement" between the two attributes (sport and animals) afforded by the SAE latents.

Overall, we see that our framework applies with minimal modification to this different setting, and yields interesting results about SAEs.

We thank the reviewers for pointing out this weakness in the presentation of our paper, and hope that this new addition will serve as a convincing example of the generality of our evaluation framework.


[1] Hanna, Michael, Ollie Liu, and Alexandre Variengien. "How does GPT-2 compute greater-than?: Interpreting mathematical abilities in a pre-trained language model." Advances in Neural Information Processing Systems 36 (2024).

---

### Meta-Review · Area_Chair_sqCg · 2024-12-12

**Metareview:**

This paper explores sparse autoencoders on language tasks that have a known ground truth to evaluate whether SAEs can provide similar interpretability and control as supervised feature dictionaries.

Since most language tasks do not have a single ground truth, the authors originally used only one highly specialized experimental setting. Reviewers found this to be lacking, and also gave advice on how to improve the presentation, and better communicate the limitations of the work. After discussion, the authors heavily revised their presentation, included better discussion of limitations, and added additional experimental settings. All these changes were well-received by the reviewers, and the paper now stands as a clear and interesting contribution to studies on interpretability techniques for language models. This is a valuable research direction, and one in desperate need of more investigation by the community.

As such, I am recommending acceptance as a Spotlight.

**Additional Comments On Reviewer Discussion:**

The main points of concern raised by reviewers were: lack of diverse experiments beyond the IOI dataset; poor presentation and writing; lacking discussion of limitations; questions about the experimental methodology; and other minor concerns.

Through the discussion (especially with Reviewer TRbA) the authors heavily edited the paper, including major rework to the presentation, inclusion of limitations, and new experiments including on new datasets. This was well received by reviewers, some of which greatly increased their scores. The major concerns have been resolved.

---

### Decision · Program_Chairs · 2025-01-22

Accept (Poster)